# Enhancing Physics-Informed Neural Networks Through Feature Engineering

**Shaghayegh Fazliani**                                                                    *fazliani@stanford.edu*
*Department of Mathematics, Stanford University, Stanford, CA, USA*

**Zachary Frangella**                                                                       *zfrangella@stanford.edu*
*Department of Management Science & Engineering, Stanford University, Stanford, CA, USA*

**Madeleine Udell**                                                                          *mudell@stanford.edu*
*Department of Management Science & Engineering, Stanford University, Stanford, CA, USA*
*ICME, Stanford University, Stanford, CA, USA*

**Reviewed on OpenReview:** *https://openreview.net/forum?id=J25OqR4pBB*

## Abstract

Physics-Informed Neural Networks (PINNs) seek to solve partial differential equations (PDEs) with deep learning. Mainstream approaches that deploy fully-connected multi-layer deep learning architectures require prolonged training to achieve moderate accuracy, while recent work on feature engineering allows higher accuracy and faster convergence. This paper introduces **SAFE-NET**, a **S**ingle-layered **A**daptive **F**eature **E**ngineering **NET**work that improves errors with far fewer parameters than baseline feature engineering methods. SAFE-NET returns to basic ideas in machine learning, using Fourier features, a simplified single hidden layer network architecture, and an effective optimizer that improves the conditioning of the PINN optimization problem. Numerical results show that SAFE-NET converges faster and typically outperforms deeper networks and more complex architectures. It consistently uses fewer parameters — on average, **53%** fewer than the competing feature engineering methods and **70-100×** fewer than state-of-the-art large-scale architectures — while achieving comparable accuracy in less than **30%** of the training epochs. Moreover, each SAFE-NET epoch is **95%** faster than those of competing feature engineering approaches. These findings challenge the prevailing belief that modern PINNs effectively learn relevant features and highlight the efficiency gains possible through feature engineering.

## 1 Introduction

Partial Differential Equations (PDEs) underpin scientific modeling but remain notoriously challenging to solve. Classical numerical methods struggle with high dimensionality and nonlinearity, while analytical solutions are rare. Traditional numerical methods are supported by software libraries such as FEniCS (Alnaes et al., 2015), deal.II (Arndt et al., 2021) and PETSc (Balay et al., 2025), but still considerable domain expertise is required for effective implementation. More recently, LLM-based approaches have been explored for automating PDE solver generation, combining symbolic reasoning and code synthesis to produce solvers (Li et al., 2025; Soroco et al., 2025; Fazliani & Udell, 2025). Such approaches might struggle when the target PDE lacks prior numerical treatment or is underrepresented in the training data, limiting their generality. Physics-Informed Neural Networks (PINNs) (Raissi et al., 2019a; Karniadakis et al., 2021) are a promising alternative, leveraging neural networks to approximate PDE solutions through residual minimization. They aim to solve PDE systems of the form

$$
\begin{aligned}
D[u(\mathbf{x}), \mathbf{x}] &= 0, & \mathbf{x} \in \Omega \\
B[u(\mathbf{x}), \mathbf{x}] &= 0, & \mathbf{x} \in \partial\Omega
\end{aligned}
\tag{1}
$$

where $D$ represents the differential operator defining the PDE, $B$ represents a general boundary condition operator, and $\Omega \subseteq \mathbb{R}^d$ is the domain of the PDE. By avoiding mesh generation, PINNs offer flexibility for forward/inverse problems and high-dimensional settings.

Despite their potential, PINNs face a fundamental challenge—they are difficult to train (Krishnapriyan et al., 2021; Rathore et al., 2024). Research shows that the differential operator in the residual loss induces ill-conditioning (De Ryck et al., 2023; Rathore et al., 2024), leading to a poor optimization landscape and slow convergence for popular first-order optimizers such as Adam (Kingma & Ba, 2014). Thus, successful PINN training can be time-consuming and fiddly, limiting the use of PINNs.

Recent work has developed several strategies to improve PINN training, including feature engineering. Feature engineering endows the network with additional features that better capture the inductive bias of the learning task. A range of feature engineering approaches, from Fourier features (RFF) (Wang et al., 2020) to radial basis function features (RBF) (Zeng et al., 2024a), have been proposed. This focus on feature engineering represents a departure from the dominant trend in modern deep learning, where end-to-end learning has largely replaced hand-crafted features. However, the scientific computing domain presents unique challenges that may favor explicit feature design. Unlike image recognition or natural language processing, where optimal feature representations are unknown a priori, PDE solutions have well-understood mathematical structures rooted in Fourier analysis, differential geometry, and physics principles. This domain knowledge provides strong theoretical guidance for feature design that is often absent in other ML applications. Recent work in domain-specific architectures—from Graph Neural Networks for relational data Bronstein et al. (2017) to Transformer attention mechanisms for sequences Vaswani et al. (2017)—suggests that incorporating structural knowledge can outperform generic architectures.

However, prior work on feature engineering generally suffers from one or more of the following four limitations: 1) they impose rigid priors (e.g. the features are fixed or random functions), 2) they require hyperparameter tuning (e.g. determining kernel hyperparameters in RBFs), 3) they are often computationally expensive, and 4) they fail to integrate domain knowledge such as boundary or initial conditions. Thus, while existing feature engineering techniques can improve performance under certain conditions, they can be PDE-specific, expensive, and sensitive to hyperparameters.

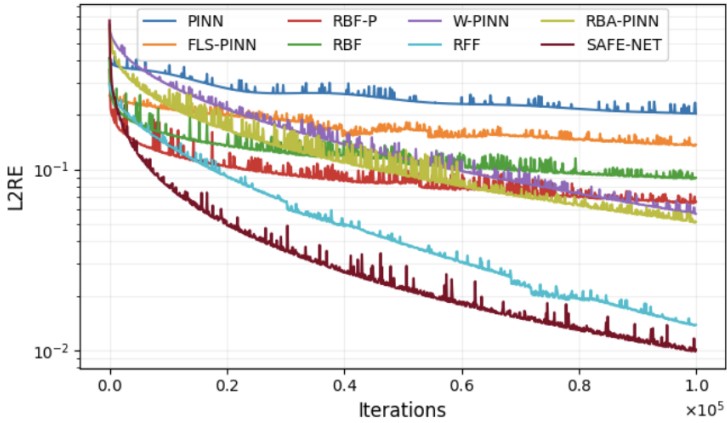

Figure 1: L2RE for the wave PDE with SAFE-NET and baselines methods using Adam. For detailed numerical results see Table 19 in Appendix C.2

To address these shortcomings we introduce **SAFE-NET**, a feature engineering framework for PINNs that combines:

(1) **Well-conditioned adaptive Fourier basis terms** as features with trainable frequencies $\omega_x$, $\lambda_t$ and amplitudes adapt to PDE-specific dominant frequencies while mitigating spectral bias.

(2) **Domain knowledge features** derived from initial and boundary conditions (if available), providing added benefits in boosting accuracy.

(3) **Normalization** of features prior to network input stabilizes quasi-Newton optimizers (e.g., L-BFGS) and other advanced optimizers and prevents divergence. Many other methods introduce sensitive architectural or problem-related hyperparameters or features, causing instability across some of our tested PDEs. In other words, SAFE-NET can safely be optimized by high-performance optimizers.

SAFE-NET does not directly target the optimization landscape like prior feature engineering approaches such as RFFs or RBFs. Instead, it seeks to improve the inductive bias of the PINN by augmenting the initial data representation with well-conditioned Fourier features and domain knowledge. These design choices make SAFE-NET particularly effective for PDEs without shocks or discontinuities, where Fourier bases work best. Interestingly, this inductive bias also provides an implicit preconditioning effect that leads to a better-conditioned optimization problem and facilitates training.

**Contributions.** We highlight the contributions of this paper:

- Our new computationally-efficient feature engineering method, SAFE-NET, offers better inductive bias than existing feature engineering methods.

- We demonstrate empirically that SAFE-NET implicitly preconditions the loss landscape, leading to faster, more stable convergence.

- SAFE-NET achieves runtime performance comparable to non-feature engineering methods with very low parameter count, while other feature engineering approaches are slower (Table 1 and Figure 4).

- Experiments across a wide variety of PDEs, including (i) linear, mildly nonlinear, and highly nonlinear dynamics, (ii) time-dependent and steady-state problems, and (iii) PDEs with predominantly smooth behavior as well as discontinuous shock-forming PDEs, in dimensions $\leq 3$ and baseline methods (Table 2) show that on average, SAFE-NET yields the best or comparable performance.

Deep learning has been seen as a promising tool for solving PDEs. However, our work with SAFE-NET shows that traditional techniques like feature engineering can achieve lower error rates and faster training times than conventional multi-layer networks, particularly suited to PDEs whose solutions are sufficiently smooth. While recent large-scale approaches such as PINNsFormer Zhao et al. (2024), PirateNets Wang et al. (2024), and operator learning approaches such as FNO Li et al. (2020b) and its variations may achieve superior accuracy in certain PDEs through complex architectures, these approaches are extremely expensive as they use very large networks: e.g., 454k parameters for PINNsFormer Zhao et al. (2023), $\geq$ 720k parameters for PirateNets Wang et al. (2024), and approximately 1M parameters for FNO and similar approaches. This paper shows that for many PDE tasks, simple feature engineering can perform as well or better than complex architectures, challenging the preconception that deeper networks effectively learn important problem features.

## 2 Insights into Feature Engineering in PINNs

This section explores the two primary approaches to feature engineering, Fourier-based and non-Fourier feature mappings. To better understand the advantages SAFE-NET offers, we highlight the strengths, limitations, and applicability of these approaches to different PDE classes.

### 2.1 Fourier-Based Feature Engineering

Fourier feature mappings leverage the spectral properties of PDE solutions to enhance high-frequency learning. They aim to address spectral bias — the tendency of neural networks to favor low-frequency functions — by transforming input coordinates into a more expressive representation. The most prominent approach, RFF Wang et al. (2021), uses the feature mapping

$$\gamma(v) = \big[\cos(Bv), \sin(Bv)\big], \tag{2}$$

with fixed Gaussian weights $B \in \mathbb{R}^{m \times d}$ drawn from $\mathcal{N}(0, \sigma^2)$. With both cosine and sine terms, this mapping projects inputs into a high-dimensional space where periodic and high-frequency patterns are more easily

captured. Theoretical insights from Tancik et al. (2020b) show that Fourier features help neural networks learn high-frequency functions in low-dimensional domains, which are typical in PDE applications. However, RFF's reliance on fixed, random frequencies limits its adaptation to PDE-specific spectral properties. Moreover, the Gaussian initialization of $B$ may not align with the dominant frequencies of the solution, leading to reduced performance.

Another baseline method we study, RBA-PINN from Anagnostopoulos et al. (2023), also incorporates Fourier feature embeddings for enforcing periodic boundary conditions in some of its variants. While the main contribution of RBA-PINN is the residual-based attention weighting scheme, their ablation studies demonstrate that Fourier features provide the most significant performance improvement among all components tested. Specifically, for the Allen-Cahn equation, RBA alone achieves $3.16 \times 10^{-3}$ relative $L^2$ error, while RBA+Fourier improves this by two orders of magnitude. The authors explicitly acknowledge that the coupling of the RBA scheme along with the Fourier feature embedding is the most important component to achieving low L2RE. We therefore include RBA-PINN as a feature engineering baseline (for periodic boundary conditions), recognizing that while RBA's core innovation lies in adaptive weighting, its best-performing configuration on PDEs with periodic boundaries relies heavily on Fourier feature engineering for its effectiveness.

## 2.2 Non-Fourier Approaches

For PDEs with sharp gradients or discontinuities, Fourier features may struggle due to the Gibbs phenomenon; see Zeng et al. (2024b). The Burgers PDE, with its sharp discontinuity at $x = 0$, is well-suited to observe this behavior. Comparing numerical results for the Burgers PDE across methods in Table 3 show that RFF performs comparably to methods without feature engineering but underperforms RBF by over an order of magnitude.

Zeng et al. (2024a) addresses the Gibbs phenomenon by using Radial Basis Functions (RBFs) as

$$\phi_{\text{RBF}}(x; c_i, \sigma) = \exp\left(-\frac{\|x - c_i\|^2}{2\sigma^2}\right), \tag{3}$$

where $c_i$ denotes the center and $\sigma$ controls the kernel width. RBF expansions can better approximate local, abrupt changes but are computationally intensive due to kernel regression requirements and less suited for periodic or high-frequency PDE solutions Tancik et al. (2020b). The RBF feature mapping function is defined as:

$$\Phi(x) = \frac{\sum_{i=1}^{m} w_i \, \phi_{\text{RBF}}(x; c_i, \sigma)}{\sum_{i=1}^{m} \phi_{\text{RBF}}(x; c_i, \sigma)}, \tag{4}$$

where $\{c_i\}$'s are the centers of the RBFs (trainable parameters) and $\{w_i\}$'s are weights for the feature mapping layer.

**Polynomial Enhancement.** A key limitation of pure RBF approaches is their difficulty in representing global trends and polynomial components that frequently appear in PDE solutions. To address this, RBF-P (RBF with Polynomials) from Zeng et al. (2024b) augments the RBF feature set with polynomial terms, so the feature mapping function is adjusted to

$$\Phi(\mathbf{x}) = \frac{\sum_{i}^{m} w_i^m \phi_{\text{RBF}}(|\mathbf{x} - c_i|)}{\sum_{i}^{m} \phi_{\text{RBF}}(|\mathbf{x} - c_i|)} + \sum_{j}^{k} w_j^k P(\mathbf{x}),$$

where P is the polynomial function. In the feature mapping layer, it can be represented as

$$\begin{bmatrix} f_1 \\ \vdots \\ f_N \end{bmatrix} = \begin{bmatrix} \phi_{\text{RBF}}(r_1^1) & \cdots & \phi_{\text{RBF}}(r_1^m) & | & 1 & x_1 & x^k \\ \vdots & \ddots & \vdots & | & \vdots & \vdots & \vdots \\ \phi_{\text{RBF}}(r_N^1) & \cdots & \phi_{\text{RBF}}(r_N^m) & | & 1 & x_N & x_N^k \end{bmatrix} \begin{bmatrix} W^m \\ W^k \end{bmatrix}$$

with $r_i = x - c_i$. The polynomial terms capture global structure while RBFs handle local variations. The weights serve as Lagrange multipliers, enabling constraints on the RBF coefficients in the parameter space. This hybrid approach combines the local approximation power of RBFs with the global trend-fitting capability of polynomials, often resulting in improved accuracy over pure RBF methods. We compare against both RBF and RBF-P in our study (see Appendix A.2 for details on the experimental set-ups for the baseline methods).

## 2.3 Comparisons & Practical Considerations

The choice between Fourier and non-Fourier features depends on PDE characteristics. Fourier features are ideal for smooth, periodic solutions but struggle with discontinuities. RBFs handle sharp discontinuities better but are computationally costly. Experiments show that the polynomial terms in RBF-P are also more effective in nonlinear equations like the Burgers Equation and Navier-Stokes; see Table 3 for numerical comparisons. Despite the disadvantage of Fourier features in these tasks, SAFE-NET offers a sensible compromise, allowing trainable frequency parameters and domain-inspired features to improve inductive bias with a unified computationally-efficient design.

## 3 Methodology

We introduce SAFE-NET in this section. We begin with some motivation from Fourier analysis.

### 3.1 Theoretical Background

Let $f(\mathbf{x}) : \mathbb{R}^d \to \mathbb{R}$ be a function defined on a $d$-dimensional domain. Under mild regularity conditions, $f(\mathbf{x})$ can be reconstructed from its Fourier transform using the inverse Fourier transform $f(\mathbf{x}) = \int_{-\infty}^{\infty} \hat{f}(\kappa) e^{2\pi i \kappa \cdot \mathbf{x}} \, d\kappa$, where $\hat{f}(\kappa)$ is the Fourier transform of $f$ at frequency $\kappa$. To approximate $f$, we can focus on the *dominant frequencies* with large $|\hat{f}(\kappa)|$. Summing over these dominant frequencies, we obtain $f(\mathbf{x}) \approx \sum_{\kappa \text{ dominant}} \hat{f}(\kappa) e^{2\pi i \kappa \cdot \mathbf{x}}$. We can express the Fourier transform $\hat{f}(\kappa)$ through its real and imaginary components to rewrite the approximation as

$$f(\mathbf{x}) \approx \sum_{\kappa \text{ dominant}} \left( A_\kappa \cos(2\pi \kappa \cdot \mathbf{x}) + B_\kappa \sin(2\pi \kappa \cdot \mathbf{x}) \right),$$

where $A_\kappa$ and $B_\kappa$ are real-valued coefficients derived from $\hat{f}(\kappa)$. Fourier basis elements are effective as features when they include the dominant frequencies $\kappa$ of $f(\mathbf{x})$.

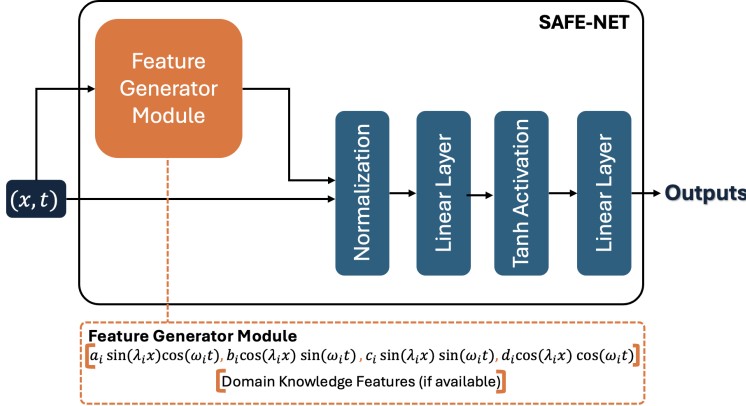

Figure 2: Diagram showing how SAFE-NET works for a 1D time-dependent PDE as an example. The Feature Generator Module has trainable frequencies and coefficients for more effective feature selection.

For a PDE solution $u(x,t) : \mathbb{R}^2 \to \mathbb{R}$, the 2D Fourier transform and its inverse are given by

$$\hat{u}(\omega_x, \lambda_t) = \iint_{\mathbb{R}^2} u(x,t) e^{-2\pi i (\omega_x x + \lambda_t t)} dx dt,$$

$$u(x,t) = \iint_{\mathbb{R}^2} \hat{u}(\omega_x, \lambda_t) e^{2\pi i (\omega_x x + \lambda_t t)} d\omega_x d\lambda_t,$$

where $(\omega_x, \lambda_t)$ are spatial and temporal frequencies. Expanding the complex exponential yields the tensor product basis

$$e^{2\pi i(\omega_x x + \lambda_t t)} = e^{2\pi i \omega_x x} \otimes e^{2\pi i \lambda_t t}$$
$$= \big[\cos(2\pi \omega_x x) + i\sin(2\pi \omega_x x)\big]$$
$$\otimes \big[\cos(2\pi \lambda_t t) + i\sin(2\pi \lambda_t t)\big].$$

For notational simplicity we absorb $2\pi$ into $\omega_x, \lambda_t$ in what follows. This expansion produces four real-valued basis functions per frequency pair $(\omega_x, \lambda_t)$ as

$$\phi_1^{\omega_x, \lambda_t}(x, t) = \cos(\omega_x x)\cos(\lambda_t t) \tag{5}$$

$$\phi_2^{\omega_x, \lambda_t}(x, t) = \sin(\omega_x x)\cos(\lambda_t t) \tag{6}$$

$$\phi_3^{\omega_x, \lambda_t}(x, t) = \cos(\omega_x x)\sin(\lambda_t t) \tag{7}$$

$$\phi_4^{\omega_x, \lambda_t}(x, t) = \sin(\omega_x x)\sin(\lambda_t t). \tag{8}$$

## 3.2 SAFE-NET

Motivated by the considerations of Section 3.1, SAFE-NET implements the parametric basis in equations (5)-(8) through learnable frequencies $\{\omega_x^{(\ell)}, \lambda_t^{(\ell)}\}_{i=1}^N$ and amplitudes $\{a^{(\ell)}, b^{(\ell)}, c^{(\ell)}, d^{(\ell)}\}_{i=1}^N$ to estimate

$$u_\theta(x, t) = \sum_{\ell=1}^N \big[a^{(\ell)} \phi_1^{\omega_x^{(\ell)}, \lambda_t^{(\ell)}}(x, t) + b^{(\ell)} \phi_2^{\omega_x^{(\ell)}, \lambda_t^{(\ell)}}(x, t)$$
$$+ c^{(\ell)} \phi_3^{\omega_x^{(\ell)}, \lambda_t^{(\ell)}}(x, t) + d^{(\ell)} \phi_4^{\omega_x^{(\ell)}, \lambda_t^{(\ell)}}(x, t)\big] \tag{9}$$

where $\theta = \{\omega_x^{(\ell)}, \lambda_t^{(\ell)}, a^{(\ell)}, b^{(\ell)}, c^{(\ell)}, d^{(\ell)}\}$ are trainable parameters. The explicit cross-frequency terms in equation (9) capture the tensor product structure of the 2D Fourier basis.

**Domain Knowledge Features.** The solution $u(x, t)$ to a PDE often inherits structure from the domain geometry, boundary conditions, and physical invariants. SAFE-NET explicitly encodes this domain knowledge through features $\psi(x, t)$ automatically derived from boundary conditions, initial conditions, and known solution patterns for each PDEs in the feature generator module. The systematic feature extraction follows these rules:

- **For Initial Conditions:** Features are extracted by analyzing the functional form of initial conditions $u(x, 0) = u_0(x)$. If $u_0(x)$ contains specific functional components (e.g., trigonometric, polynomial, or exponential terms), corresponding features are included. Therefore, if $u(x, 0) = \sum_k a_k f_k(x)$, then $\psi(x, t) = \{f_k(x)\}$ are included as domain features.

- **For Boundary Conditions:** If homogeneous Dirichlet conditions $u(0, t) = u(L, t) = 0$ are present, features satisfying these conditions naturally (e.g., $\sin(\frac{n\pi x}{L})$) are prioritized by SAFE-NET's frequency initialization (Appendix A.1.4).

Domain Knowledge features for the PDEs considered in this paper appear in Appendix A.1. As shown in Figure 2, depending on the availability of domain information, these features are concatenated with the Fourier basis terms before normalization and linear projection. Ablation studies on the role of these extra added features (see Appendix C.1.6) show that they result in improvements varying across different PDE tasks. Overall, they provide more accuracy boosting when the initial and boundary conditions carry more information of the final solution while being a lightweight supplementary component of SAFE-NET. SAFE-NET uses only one hidden layer: given an input $(\boldsymbol{x}, t) \in \Omega \times \mathbb{R}$, the SAFE-NET network computes

$$f_\theta(\boldsymbol{x}, t) = w_2^T \sigma(W_1 \phi(w_{\text{SAFE-NET}}, (\boldsymbol{x}, t)) + b_1) + b_2, \tag{10}$$

with parameters $\theta = (w_{\text{SAFE-NET}}, W_1, b_1, w_2, b_2)$, nonlinearity $\sigma = \tanh(\cdot)$, and learnable feature mapping $\phi(w_{\text{SAFE-NET}}, (\boldsymbol{x}, t))$. Complete architectural specifications, parameter initialization schemes, and training

procedures for SAFE-NET are provided in Appendix A.1. To ensure robust experimental validation, we conducted comprehensive ablation studies examining key design choices including sensitivity to the number of Fourier features (Figure 9 and 18), activation function selection for the hidden layer (Figure 17 and Table 15), sensitivity to our initialization strategy for Fourier coefficients and frequencies (Tables 17 and 18), number of layers (Figure 19), and normalization (Figure 20). These ablation results, presented Appendix C.1, explaining the choice of each component.

By combining the generality of Fourier features with known solution characteristics, SAFE-NET accelerates convergence. Figure 3 compares SAFE-NET with the best competing PINN architecture for each PDE in Table 2.

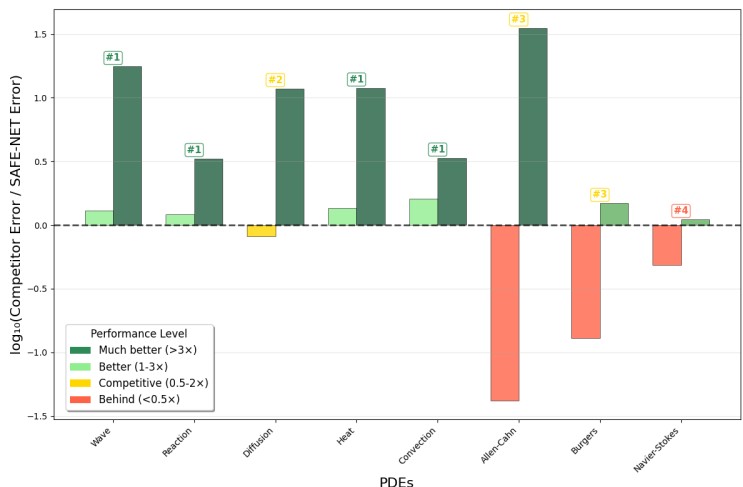

Figure 3: For each PDE, bars show $\log_{10}\left(\frac{\text{Competitor Error}}{\text{SAFE-NET Error}}\right)$ comparing SAFE-NET to the best competitor baseline method (left) and median competitor baseline method (right). Colors represent performance levels, and rank annotations show SAFE-NET's position among all methods. SAFE-NET ranks first on 4/8 PDEs. It achieves top-2 performance on 5/8 PDEs; the PDEs on which SAFE-NET falls behind (3rd or 4th place) are Allen-Cahn, Burgers and Navier-Stokes, which are non-linear and not shock-free.

**Cost.** Table 1 and Figure 4 illustrate the parameter count and runtime (per epoch) for different baseline methods. The setups used for each method in our experiments appear in Appendix C. One hidden layer suffices for SAFE-NET, reducing its parameter count and improving speed.

Table 1: Parameter count comparison between baseline methods and SAFE-NET. Using the same number of features (128), SAFE-NET achieves a significantly lower parameter count than competing feature engineering methods while remaining comparable to non-feature engineering approaches.

| PINN | FLS-PINN | W-PINN | RBA-PINN |
|------|----------|--------|----------|
| 5.3k | 5.3k | 5.3k | 16k |
| **RFF** | **RBF** | **RBF-P** | **SAFE-NET** |
| 14.5k | 14.2k | 14.7k | 6.8k |

**Scalability.** The number of additional features in SAFE-NET increases exponentially with the spatial dimension of the PDE. Hence SAFE-NET offers improves accuracy and computational advantages for lower-dimensional problems, where better coverage of frequency space improves approximation quality, but is not suited to problems with high dimensions. Developing effective feature engineering methods for higher dimensional PINNS is an important challenge for future work.

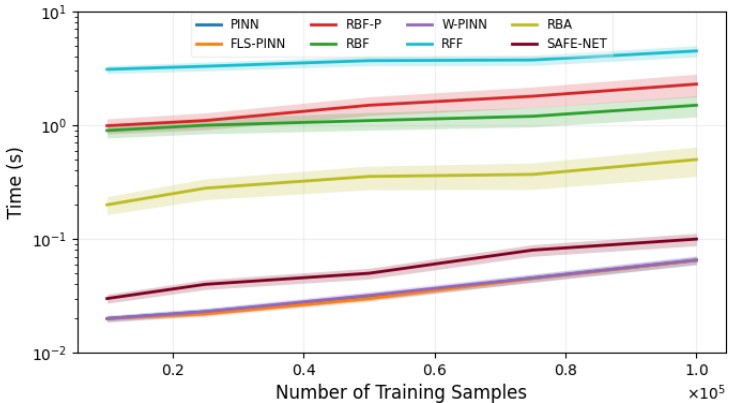

Figure 4: Average runtime comparison across varying numbers of training samples for baseline methods and SAFE-NET using identical computational resources and no concurrent processes. SAFE-NET demonstrates efficiency over competing feature engineering methods, maintaining runtime close to the less computationally expensive non-feature engineering methods.

## 4 Related work

Much work has been done to improve the training of PINNs that take different approaches from feature engineering. Broadly, these approaches can be divided into three categories: architectural modifications, loss-reweighting, and optimizer design. Recent efforts to improve PINN accuracy and speed include:

**Architectural Modifications.** One way to improve the training procedure for PINNs is to modify the network architecture, improving the optimization landscape relative to the basic PINN, making training easier. Examples of this approach include the adaptive activation functions of Jagtap et al. (2020) (A-PINNs), and specialized architectures designed to mitigate spectral bias (Li et al., 2020a).

**Loss Re-weighting.** Another popular technique is loss reweighting. For certain PDEs, the residual loss tends to dominate the boundary loss in that the optimizer focuses too much on minimizing the residual loss, leading to a solution that fails to satisfy the boundary conditions. To address this, techniques like W-PINN Wang et al. (2022) balance loss components through heuristic or learned weights to down-weight the residual loss and better fit the boundary loss.

**Optimizer Design.** Another popular approach to PINN training is to develop more sophisticated optimizers that are more robust to ill-conditioning. Several notable proposals in this area are the natural gradients method of Müller & Zeinhofer (2023), MultiAdam Yao et al. (2023), and NysNewton-CG Rathore et al. (2024). These methods target ill-conditioning directly and use curvature information from the loss or the model to precondition the gradient. This leads to an improved optimization landscape locally, enabling the optimizer to take better steps and progress faster. In contrast, SAFE-NET enjoys an implicit preconditioning effect that is global—by incorporating a trainable feature layer, SAFE-NET changes the PINN objective, globally changing the optimization landscape. The results in Section 5.3 show SAFE-NET enjoys a significantly better-conditioned optimization landscape. Thus, SAFE-NET can be further combined with more sophisticated optimization schemes to obtain further improvements.

## 5 Results

We provide an overview of the experimental setup and present results across multiple benchmarks.

### 5.1 Baselines

We compare two groups of PINN baselines: (i) without feature engineering (PINN, FLS-PINN, W-PINN) and (ii) with feature engineering (RFF, RBF, RBF-P, RBA-PINN). Details on each baseline could be found in Appendix A.2.

As noted earlier in Section 1, large models such as PINNsFormer (454k parameters), PirateNets (720k+ parameters), operator learning methods (1M+ parameters), and several other PINN variants with hundreds of thousands of parameters demonstrate the performance gains possible with large and complex architectures. In contrast, our experiments focus on well-performing baseline methods with simple structures and low parameter counts (at or below 16k; see Table 1), especially emphasizing feature engineering approaches that offer computational efficiency. This scope allows us to demonstrate that simple feature engineering also yields accurate solutions at a fraction of the computational cost of the largest models.

We experiment on eight PDE benchmarks in this section as provided in Table 2. Depending on availability, the datasets for these tasks are either from PDE benchmarks such as PDEBench Takamoto et al. (2024) and PINNacle Hao et al. (2023) or implemented directly if unavailable online. More details on each PDE and its source are provided in Appendix B. An additional experiment on a synthesized non-homogeneous boundary heat task is included in Appendix C.3.

Table 2: Overview of the tested PDEs. Following the conventions of PINNs literature, the dimensions provided in this table are *spatial dimensions*. An extra dimension should be added for time-dependent PDEs to reach the correct number of variables.

| PDE | Dimensions | Type | State |
|---|---|---|---|
| Wave | 1D | Linear | Time-dependent |
| Reaction | 1D | Nonlinear | Time-dependent |
| Diffusion | 1D | Linear | Time-dependent |
| Heat | 2D | Linear | Time-dependent |
| Convection | 1D | Linear | Time-dependent |
| Burgers | 1D | Nonlinear | Time-dependent |
| Allen-Cahn | 1D | Nonlinear | Time-dependent |
| Navier-Stokes | 2D | Nonlinear | Steady-state |

### 5.2 Experiments

Following prior work (such as Rathore et al. (2024) and Zeng et al. (2024b)), we use a two-stage Adam + L-BFGS optimizer for the first set of experiments. Recent loss-landscape studies on PINNs show that first-order methods such as Adam make rapid progress away from saddle points but eventually slow down in the ill-conditioned valleys induced by differential operators, whereas quasi-Newton methods like L-BFGS improve local conditioning yet are attracted to saddles and often stall when started from scratch. Combining them inherits the best of both worlds: an Adam "warm-up" steers the parameters clear of saddles; switching to L-BFGS then preconditions the Hessian and yields much faster local convergence . Nevertheless, pure L-BFGS can terminate early because its strong-Wolfe line search sometimes returns a step size of zero, especially on stiff PDEs . Our schedule therefore runs Adam first until progress slows down significantly and switches to L-BFGS until the maximum number of iterations is reached. A more detailed theoretical justification of the Adam + L-BFGS schedule appears in Rathore et al. (2024).

**Optimization Schedule (1)**. The first set of experiments uses Adam for either 10k, 20k, or 30k iterations, followed by L-BFGS for the remainder of the training run (total 40k). We found that for our tested set of experiments, 30k iterations of Adam followed by 10k iterations of L-BFGS performs best. Adam has an initial learning rate of 0.001, decaying exponentially by 0.9 every 2k iterations, followed by full-batch L-BFGS until convergence (tolerance $10^{-10}$) or if the maximum number of iterations is reached.

Optimizer settings are identical across all PDEs and architectures; each experiment is repeated with five random initializations, and the model performance is assessed using the relative $\ell_2$ error (L2RE), defined as

$$\frac{\|y - \hat{y}\|_2}{\|y\|_2} = \frac{\sqrt{\sum_{i=1}^{n}(y_i - \hat{y}_i)^2}}{\sqrt{\sum_{i=1}^{n} y_i^2}},$$

where $\hat{y}$ is the predicted solution and $y$ is the true solution. Notably, while we run every method for the same number of epochs, SAFE-NET achieves faster per-epoch computation times than competing approaches, reducing its total training time.

Table 3: PDE benchmark results comparing several PINN architectures in relative $L^2$ error. The best results are shown in **Bold**. However, numerically close results (i.e., within 30% of the best result, including the best result in **Bold** itself) are also highlighted in Blue . This accounts for potential variance in implementation and hyperparameter sensitivity. Cells with (*) indicate L-BFGS divergence in our experiments.

| PDE | PINN | FLS-PINN | W-PINN | RFF |
|---|---|---|---|---|
| Wave | 6.62e−2 | 4.37e−2 | 8.79e−3 | 7.39e−3 |
| Reaction | 4.98e−2 | 1.27e−1 | 4.52e−2 | 1.20e−2 |
| Diffusion | 9.65e−3 | 5.89e−2 | 1.43e−3 | 2.56e−3 |
| Heat | 6.34e−3 | 6.98e−3 | 9.98e−3 | 8.56e−3 |
| Convection | 1.96e−2 | 4.12e−2 | 9.77e−3 | * |
| Allen-Cahn | 4.98e−1 | 1.46e−0 | 1.29e−1 | 1.03e−2 |
| Burgers | 1.01e−2 | 6.16e−2 | * | * |
| Navier-Stokes | 7.49e−1 | 7.91e−1 | * | 6.76e−1 |

| PDE | RBA-PINN | RBF | RBF-P | SAFE-NET |
|---|---|---|---|---|
| Wave | 1.57e−3 | 2.48e−2 | 2.13e−2 | **1.21e−3** |
| Reaction | 3.31e−2 | 1.98e−2 | 1.37e−2 | **9.93e−3** |
| Diffusion | 1.29e−4 | 4.02e−4 | **9.91e−5** | 1.21e−4 |
| Heat | 7.35e−4 | 3.65e−3 | 7.21e−4 | **5.31e−4** |
| Convection | 8.32e−3 | 7.72e−2 | 7.02e−3 | **4.37e−3** |
| Allen-Cahn | **8.21e−5** | * | 1.06e−4 | 1.97e−3 |
| Burgers | 4.17e−4 | 3.95e−3 | **3.47e−4** | 2.67e−3 |
| Navier-Stokes | 4.86e−1 | **2.56e−1** | 2.98e−1 | 5.26e−1 |

Even after warming up with Adam, L-BFGS can still cause divergence or instability on some PDEs and baseline methods. Corresponding entries are denoted by "*" in the result table. There could be several reasons why this is possible. For instance, RFF uses fixed random frequencies drawn from a Gaussian distribution, which can lead to very poor conditioning depending on the random initialization. The mismatch between random frequencies and the true solution frequencies can create ill-conditioned optimization landscapes that cause L-BFGS to diverge. Alternatively, in the case of W-PINN and Navier-Stokes, W-PINN's adaptive weighting scheme computes weights based on NTK matrix traces (see Wang et al. (2022) for the details or the computation). Due to the complex flow physics of Navier-Stokes, the NTK matrices could become ill-conditioned near boundary layers or rapid changes in the adaptive weights can destabilize L-BFGS.

SAFE-NET, on the other hand, is compatible with quasi-Newton methods such as L-BFGS: it can *safely* be optimized by them. Table 3 demonstrates that for shock-free PDEs (i.e., not Burgers or Navier-Stokes, and Allen-Cahn to some extent), SAFE-NET outperforms competing methods in L2RE by in 4/5 tasks, following closely behind RBF-P for the diffusion task. (See Figure 3 for visual demonstration). Allen-Cahn is "less nonlinear" than Burgers or Navier-Stokes, but can also develop steep gradients and interfaces, especially in phase separation dynamics. In practice, these sharp transition layers are not true shocks in the hyperbolic PDE sense, but numerically they still present challenges similar to shocks — including Gibbs phenomena for Fourier features in particular. Despite that, SAFE-NET manages to come third best after RBA-PINN and RBF-P for both Allen-Cahn and Burgers tasks. This outcome is not surprising as SAFE-NET is not designed for PDEs with shocks. Particularly, Zeng et al. (2024b) suggests that the RBF kernel handles the discontinuity at $x = 0$ in the Burgers equation more effectively.

**Experiments with Alternative Combinations.** While the Adam + L-BFGS combination in Optimization Schedule (1) demonstrates superior performance compared to individual optimizers (see Appendix A.3), our analysis reveals that L-BFGS consistently stalls before reaching the maximum iteration limit. To further improve SAFE-NET's performance, we developed optimizer schedules that restart L-BFGS after it stalls. The following combination, which we call $(\text{Adam} + \text{L-BFGS})^2$, is a particularly successful choice for optimizing the SAFE-NET architecture. Hyperparameter analysis in Appendix A.3 validates our optimizer choice.

**Optimization Schedule (2)**. The experiments with $(\text{Adam} + \text{L-BFGS})^2$ also run for 40k iterations. First, Adam is used as a warm-up phase (**Adam(1)**), followed by full-batch L-BFGS until it stalls (which is always the case in our experiments) or hits the maximum number of iterations. Then Adam (**Adam(2)**) runs again followed by a final round of L-BFGS. Testing multiple switch points for both Adam(1) and Adam(2) suggest that for $(\text{Adam} + \text{L-BFGS})^2$, running a shorter first warm-up phase of 3k iterations of Adam, followed by L-BFGS until it stalls (which happen within 10k iterations), then reverting to Adam and switching to L-BFGS for the last 3k epochs works best for the tested PDEs. Since Adam (1) runs as a short warm-up phase, we grid-searched the learning rate $\eta \in \{10^{-1}, 10^{-2}, 10^{-3}, 10^{-4}, 10^{-5}\}$ per PDE. However, for Adam(2), the learning rate schedule matches our choice in Optimization Schedule (1), which starts with an initial learning rate of $\frac{\eta}{2}$, decaying exponentially by 0.9 every 2k iterations. Both rounds of L-BFGS in $(\text{Adam} + \text{L-BFGS})^2$ match our choices for L-BFGS in Optimization Schedule (1).

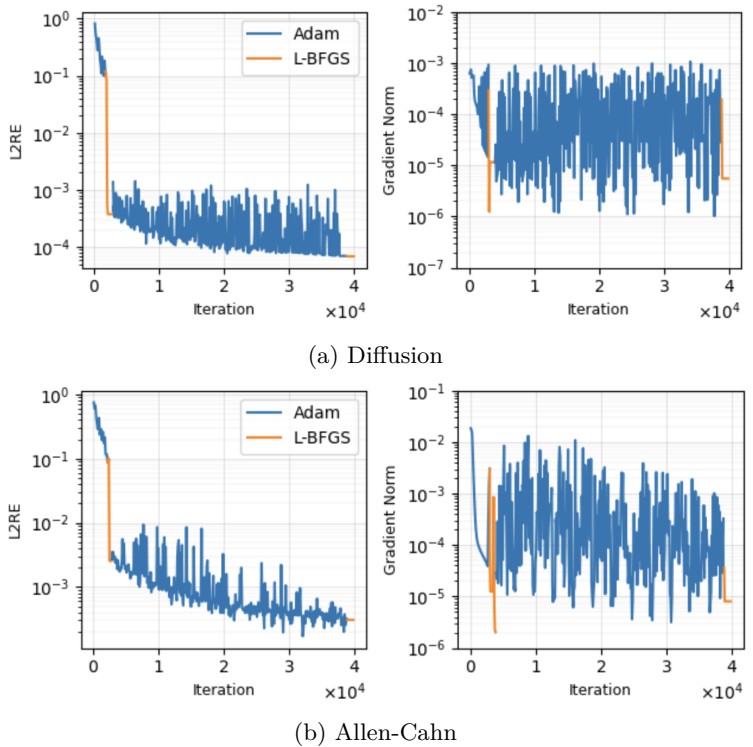

(a) Diffusion

(b) Allen-Cahn

Figure 5: L2RE and gradient norm for the diffusion and Allen-Cahn PDEs using SAFE-NET with $(\text{Adam} + \text{L-BFGS})^2$.

Figure 5 demonstrates the performance of SAFE-NET + $(\text{Adam} + \text{L-BFGS})^2$ on the diffusion and Allen-Cahn PDEs, presenting the gradient norm plots as well. Table 4 summarizes these numerical results, showing significant improvements on more challenging PDEs containing steep interface transitions, shocks, and discontinuities (Allen-Cahn, Burgers, and Navier-Stokes) along with improvements on other PDEs matching the top results of Table 3. These results demonstrate the effectiveness of $(\text{Adam} + \text{L-BFGS})^2$ when paired with SAFE-NET, leading to significant improvements compared with Adam + L-BFGS, especially in the diffusion and Allen-Cahn tasks. With this improved optimizer, SAFE-NET beats all compared architectures

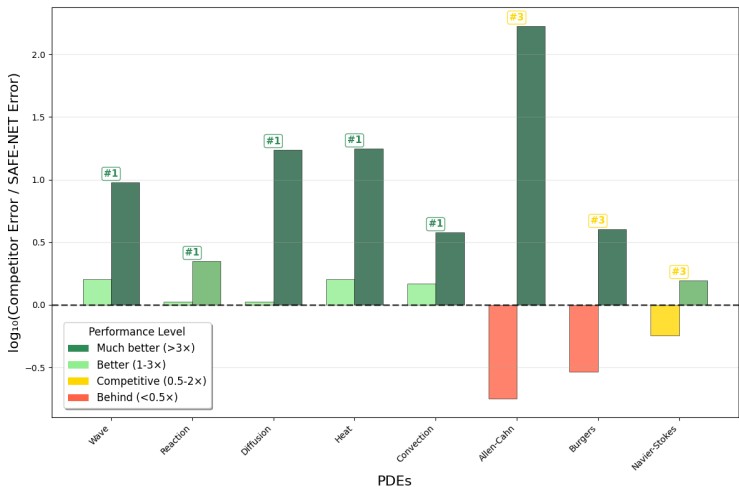

Figure 6: Comparison of SAFE-NET + $(\text{Adam} + \text{L-BFGS})^2$ results for each PDE against the best baseline performance achieved using either $(\text{Adam} + \text{L-BFGS})$ or $(\text{Adam} + \text{L-BFGS})^2$. This comparison selects the optimal result for each competing method across both optimization strategies, highlights SAFE-NET's consistent advantages on shock-free PDEs (now ranking first on 5/8 tasks) and validates the effectiveness of $(\text{Adam} + \text{L-BFGS})^2$ with SAFE-NET. Compare with Figure 3.

on 5/8 of the tested PDEs as well as showing improvements for all the nonlinear PDEs with shocks; see Figure 6 for a visual demonstration.

Table 4: L2RE for SAFE-NET with Optimization Schedule (1) and Optimization Schedule (2)

| PDE | Adam + L-BFGS | $(\text{Adam} + \text{L-BFGS})^2$ |
|---|---|---|
| Wave | 1.21e$-$3 | 9.81e$-$4 |
| Reaction | 9.93e$-$3 | 8.91e$-$3 |
| Diffusion | 1.21e$-$4 | 8.23e$-$5 |
| Heat | 5.31e$-$4 | 3.60e$-$4 |
| Convection | 4.37e$-$3 | 3.84e$-$3 |
| Allen-Cahn | 9.97e$-$4 | 4.13e$-$4 |
| Burgers | 2.67e$-$3 | 9.87e$-$4 |
| NS | 5.26e$-$1 | 3.71e$-$1 |

Tables 5 and 6 appearing in Appendix A.3 contain complete numerical results for Optimization Schedule (1) and (2) along with other combinations tested. Notably, $(\text{Adam} + \text{L-BFGS})^2$ fails to improve (and often degrades) performance for the majority of the baseline methods across different PDEs, making it unsuitable to compare PINN architectures. Figure 8 shows the average percentage of improvement or deterioration (in L2RE) for each method after switching from Optimization Schedule (1) to Optimization Schedule (2). To see an example of the loss curves, Figure 7 compares the performances of the top 3 best performing methods on the diffusion PDE using Adam + L-BFGS (Optimization Schedule (1)) and $(\text{Adam} + \text{L-BFGS})^2$ (Optimization Schedule (2)).

Finally, Figure 9 shows the effects of adding Fourier features on the wave, heat, and Navier-Stokes PDEs; the error gradually decreases as the number of SAFE-NET features increases until it reaches a saturation point (at around 120–140 features for these problems). This observation (and similar observations in Appendix C.1) suggests that there is no advantage to adding additional features after a certain complexity is reached: they do not improve performance, but only increase the number of trainable parameters and computational cost. Detailed numerical results for other PDEs are provided in Table 16.

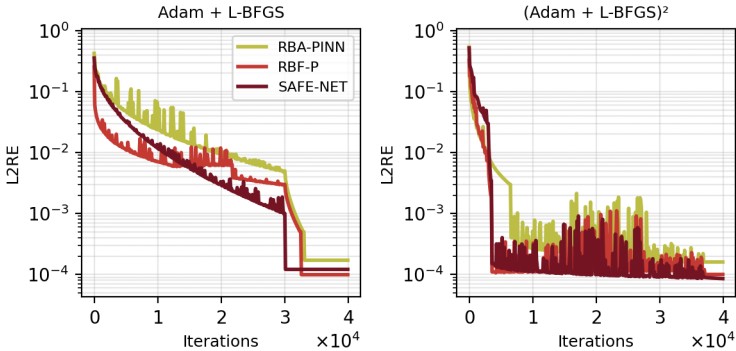

Figure 7: L2RE for RBA-PINN, RBF-P, and SAFE-NET on the diffusion PDE using Adam + L-BFGS (left) and $(\text{Adam} + \text{L-BFGS})^2$ (right). SAFE-NET, previously ranking second in performance on the diffusion task, is now outperforming the other methods using $(\text{Adam} + \text{L-BFGS})^2$.

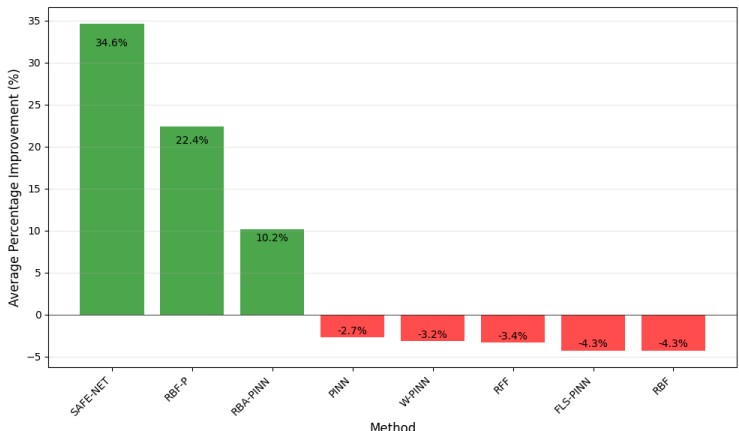

Figure 8: Average percentage of improvement or deterioration (in L2RE) for each method after switching from Optimization Schedule (1) to Optimization Schedule (2). Figure 12 in the appendix shows percentages for every method on every PDE, demonstrating that SAFE-NET is the only method that produces results in consistent improvement on every PDE after switching to $(\text{Adam} + \text{L-BFGS})^2$.

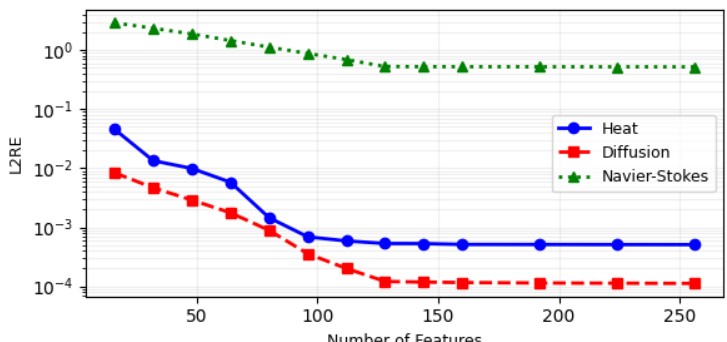

Figure 9: L2RE decreases as number of features increases.

### 5.3 Feature Engineering & Spectral Density Analysis

SAFE-NET significantly improves the conditioning of all tested PDEs. We analyze conditioning both at initialization (3k epochs) and after training for a while (100k iterations) using SAFE-NET and PINN with Adam as the optimizer with an initial learning rate of 0.001 and an exponential decay rate of 0.9 every 2k steps (Experimental setup of Figure 1)

This experiment uses Adam only since L-BFGS has preconditioning effects that would confound our attempt to understand the effect of architecture on problem conditioning. Figure 10 illustrates the spectral density on the Hessian for the wave PDE as an example, plotted from the same experiment as in Figure 1. Analogous plots for other PDEs are provided in Appendix C, alongside spectral density calculation details.

We see that SAFE-NET shifts most eigenvalues values close to 1 at initialization (Figure 10 top). Post-training spectral density plots (Figure 10 bottom) reveal dramatic conditioning improvements: the top eigenvalues for the wave problem decrease by a factor of $10^4$, while those for the heat and Burgers problems decrease by $10^2$ (Figures 25 and 26. Further experiments (Appendix C) show that SAFE-NET reduces both the number and density of large eigenvalues across all problems and all baseline methods (see Table 21)

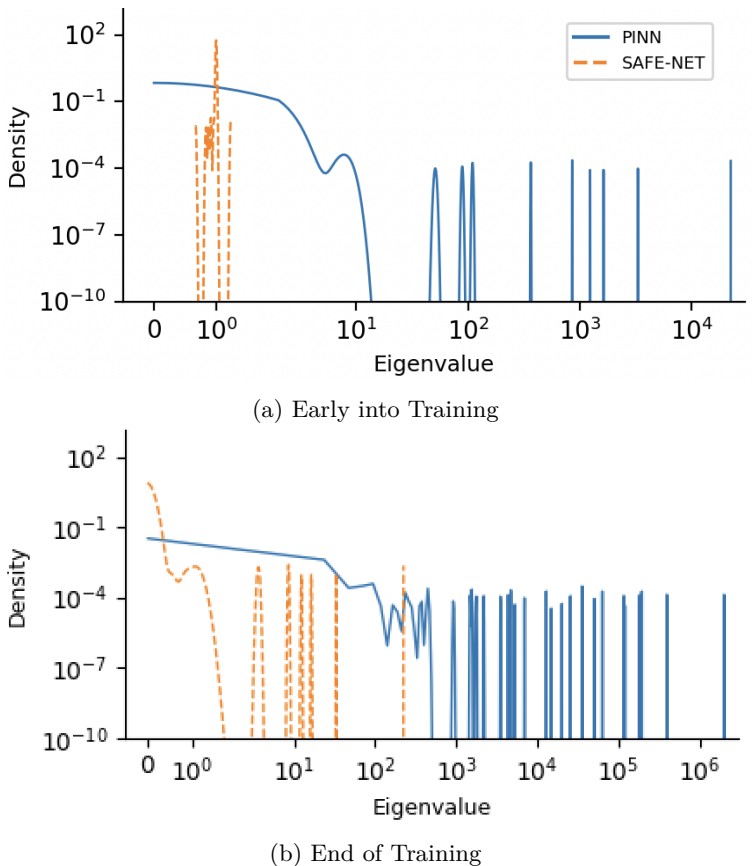

(a) Early into Training

(b) End of Training

Figure 10: Spectral density for the wave PDE using SAFE-NET and PINN at the early stages of training and at the end of training.

This improvement in conditioning is a key factor explaining why SAFE-NET features are easier to optimize. Figure 11 compares SAFE-NET and PINNs performance on different loss components for the diffusion and Allen-Cahn PDEs (from the same experiment as Figure 5) as an example. Rathore et al. (2024) argues the residual loss is the main cause of the ill-conditioned loss landscape of PINNs. SAFE-NET significantly improves the conditioning of each loss component, including the residual loss.

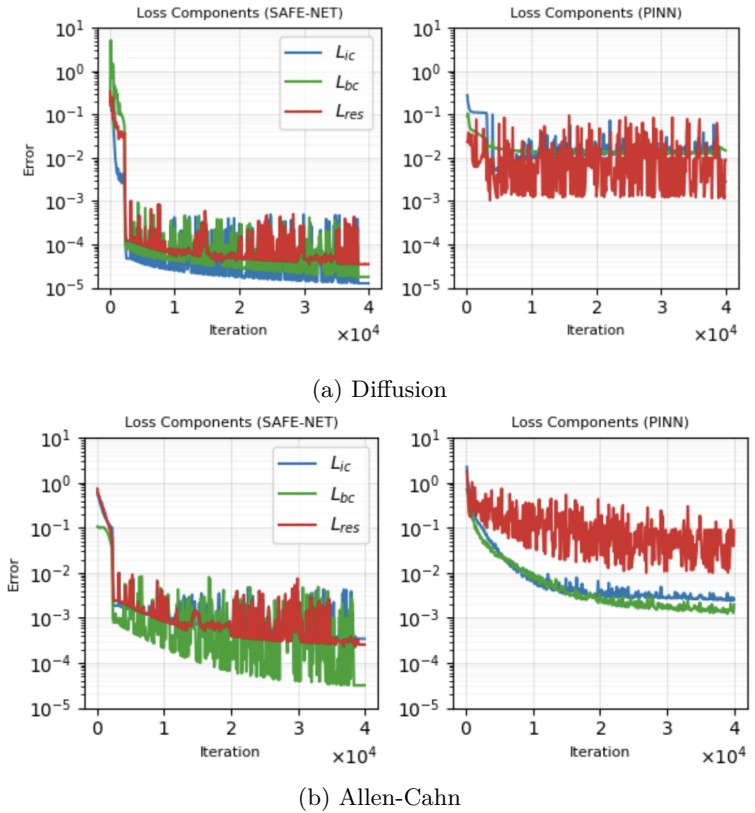

(a) Diffusion

(b) Allen-Cahn

Figure 11: Comparison of loss components of Figure 5 with SAFE-NET and PINN with $(\text{Adam} + \text{L-BFGS})^2$. SAFE-NET significantly improves each loss component for both PDEs.

# 6  Implicit preconditioning in SAFE-NET

Empirical results in Section 5.3 show SAFE-NET improves the conditioning of each problem both at the beginning and end of training. To develop better intuition into this empirical observation for the early stages of training, we consider a simple didactic setting similar to Wang et al. (2020), where the network is given by

$$u(x,t) = w^\top \phi(x,t), \tag{11}$$

here $(x,t) \in \mathbb{R}^2$ and $\phi$ is the feature map.

We begin with a definition:

**Definition 6.1.** For a neural network $f_\theta(x)$ with parameters $\theta \in \mathbb{R}^p$, the tangent kernel $\Theta_f : \Omega \times \Omega \to \mathbb{R}$ is given by

$$\Theta_f(x',x) = \nabla_\theta f(x')^\top \nabla f_\theta(x).$$

Given an input dataset $X \in \mathbb{R}^{n \times d}$, the tangent kernel matrix is the $n \times n$ matrix with entries

$$(\Theta_f(\theta))_{ij} = \nabla_\theta f(x_i)^\top \nabla f_\theta(x_j), \qquad i,j = 1, \ldots, n,$$

where $x_i$ and $x_j$ are $i$th and $j$th rows of $X$.

The *neural tangent kernel* $\Theta_f^\infty \in \mathbb{R}^{n \times n}$ is the fixed kernel defined as

$$\Theta_f^\infty(x',x) = \lim_{p \to \infty} \mathbb{E}[\Theta_f(x',x)],$$

where $p$ is the number of trainable parameters and the expectation is taken over the weights at initialization Jacot et al. (2018); Liu et al. (2020). The neural tangent kernel matrix $\Theta_f^\infty$ is defined analogously to the tangent kernel matrix.

In the limit $p \to \infty$, neural net training with $f_\theta$ is equivalent to kernel regression with the NTK matrix $\Theta_f^\infty$. As training is reduced to a kernel regression problem, the convergence speed of gradient-based optimizers is controlled by the conditioning of the NTK matrix $\Theta_f^\infty$ Jacot et al. (2018); Liu et al. (2022). Thus, a better-conditioned NTK yields a better optimization landscape and faster convergence.

In the context of (11) and under idealized conditions, we continue the following calculations as an example to build intuition into the empirical results suggesting that SAFE-NET features lead to a better conditioned NTK and, so, a better optimization landscape. When $n$ and $p$ are large, the spectrum of $\Theta_f^\infty$ is closely related to the spectrum of the integral operator $T_{\Theta_f}(g)(x) := \int_\Omega \Theta_f(x', x)g(x)dx$. Wang et al. (2020). Thus, we shall obtain control over the $T_{\Theta_f}$ spectrum for $f$ given by (11).

We begin by writing the tangent kernel function corresponding to (11). As $\theta = w$, $\nabla_\theta u(x, t) = \phi(x, t)$, therefore the tangent kernel function is given by

$$\Theta_u\left((x,t),(x',t')\right) = \phi(x,t)^\top \phi(x',t').$$

Let us focus on the Fourier basis features. Using the notation of Section 3.2, we have

$$\phi(x,t) = \sum_{i=1}^{N} \sum_{j=1}^{4} c_j^{(\ell)} \phi_j^{(\ell)}(x,t),$$

where $\{\phi_j^{(\ell)}\}_{j=1}^4$ are defined in equations (5)-(8) and $\ell = 1, ..., N$ are the number of each type of product features. The kernel is a sum of separable functions as

$$\Theta_u((x,t),(x',t')) = \sum_{i=1}^{N} \sum_{j=1}^{4} \left(c_j^{(\ell)}\right)^2 \phi_j^{(\ell)}(x,t)\phi_j^{(\ell)}(x',t'). \tag{12}$$

The eigenvalues $\beta$ and eigenfunctions $\psi(x,t)$ satisfy

$$\int \Theta_u((x,t),(x',t'))\psi(x',t')\,dx'dt' = \beta\psi(x,t).$$

Substituting $\Theta_u((x,t),(x',t'))$ from equation (12), we get

$$\sum_{i=1}^{N} \sum_{j=1}^{4} \left(c_j^{(\ell)}\right)^2 \phi_j^{(\ell)}(x,t) \int \phi_j^{(\ell)}(x',t')\psi(x',t') = \beta\psi(x,t). \tag{13}$$

Set $\alpha_j^{(\ell)} := \int \phi_j^{(\ell)}(x',t')\psi(x',t')dx'dt'$ by the domain compactness assumption, so for $\beta \neq 0$, we get

$$\psi(x,t) = \frac{1}{\beta} \sum_{i=1}^{N} \sum_{j=1}^{4} \alpha_j^{(\ell)} \left(c_j^{(\ell)}\right)^2 \phi_j^{(\ell)}(x,t). \tag{14}$$

To calculate the eigenvalues, substitute equation (14) into equation (13) to get

$$\frac{1}{\beta} \sum_{i,\ell}^{N} \sum_{j,m}^{4} \left(c_j^{(\ell)}\right)^2 \phi_j^{(\ell)} \alpha_m^{(\ell)} \left(c_m^{(\ell)}\right)^2 \int \phi_j^{(\ell)}(x',t')\phi_m^{(\ell)}(x',t')$$

$$= \beta\left(\frac{1}{\beta} \sum_{i=1}^{N} \sum_{j=1}^{4} \alpha_j^{(\ell)} \left(c_j^{(\ell)}\right)^2 \phi_j^{(\ell)}\right)$$

Now, $\{\phi_j^{(\ell)}\}$ forms an orthonormal basis, as each $\phi$ is chosen from an orthogonal Fourier basis and has unit norm. Thus,

$$\frac{1}{\beta} \sum_{i=1}^{N} \sum_{j=1}^{4} \alpha_j^{(\ell)} \left(c_j^{(\ell)}\right)^4 \phi_j^{(\ell)} = \sum_{i=1}^{N} \sum_{j=1}^{4} \alpha_j^{(\ell)} \left(c_j^{(\ell)}\right)^2 \phi_j^{(\ell)}.$$

From this display, we conclude that for each pair $(i,j)$, $\alpha_j^{(\ell)} \left(c_j^{(\ell)}\right)^2 = 0$ or $\beta = \left(c_j^{(\ell)}\right)^2$. Thus, the eigenvalues of $T_{\Theta_u}$ are 0 or equal $(c_j^{(\ell)})^2$ with corresponding eigenfunction $\psi(x,t) = \alpha_j^{(\ell)}\phi_j^{(\ell)}(x,t)$. As the non-zero eigenvalues correspond to the directions relevant to learning, we focus on them. Recall the $\{c_j^{(\ell)}\}$ are the trainable amplitudes and are set to 1 at initialization. Hence, at initialization and early stages, we expect the condition number of the matrix $\Theta_u$ to be approximately 1. If $u$ were in the infinite width limit, this would imply $\Theta_u^\infty$ is well-conditioned and fast convergence of gradient-based optimizers. Thus, this idealized example shows the features selected by SAFE-NET can lead to better conditioning and faster convergence. In practice, finite network width and perturbations from domain knowledge features and normalization are expected to introduce some spreading in the eigenvalue distribution. However, experiments show that SAFE-NET maintains a denser eigenvalue distribution around the theoretical prediction from the example (see Figure 10(a)), preserving the core conditioning benefits anticipated by the idealized example and its insights. Early on in training, Figure 10 shows the eigenvalue distribution is relatively uniform. During training, SAFE-NET gradually adjusts the $c_j^{(\ell)}$'s, balancing different frequencies rather than letting any single Fourier mode dominate. As a result, the eigenvalue distribution shifts outwards relatively slowly. Thus, even at the end of the training, Figure 10 shows the eigenvalue distribution has not changed much from its initialization, so the landscape remains well-conditioned.

## 7 Discussion and Future Work

Our results encourage a fresh look at the importance of feature engineering for PDEs. While machine learning trends have favored complex architectures, our work suggests that engineered features can offer a useful implicit bias that deep architectures struggle to replicate. PDEs present unique challenges, including ill-conditioned solution spaces, where deep learning techniques tend to struggle. While newer architectures could offer improvements, our theory shows engineered features can always improve problem conditioning. Hence we expect feature engineering will have lasting importance for solving PDEs, complementing advances in deep learning.

Fourier features align best with low-to-moderate-dimensional PDEs demonstrating smoother behavior; for strongly nonlinear or shock-forming problems, non-Fourier features (e.g., RBF-P) are often more suitable, consistent with our results. While SAFE-NET demonstrates significant improvements in conditioning and convergence for PDEs whose solutions exhibit predominantly smooth behavior (including piecewise-smooth profiles without strong shocks), several avenues remain to improve feature engineering in PINNs. The first is multi-stage feature learning, following Wang & Lai (2023), a multi-stage approach that further boosts performance by iteratively learning a best fit on the residual error, using SAFE-NET as a base learner could be developed. Another direction is better incorporating physical priors. Enforcing physical laws such as conservation principles and symmetries into the feature engineering process could improve the performance of SAFE-NET on complex problems. Features based on concepts like Noether's theorem or Hamiltonian mechanics could provide stronger inductive biases. Finally, it would be interesting to also consider non-Fourier features like radial basis functions for handling sharp gradients, which could allow SAFE-NET to perform well on a wider range of problems. The challenge here lies in maintaining numerical stability when combining these different types of features. Most importantly, extending feature engineering to truly high-dimensional problems is important future work, but it presents very different challenges that are currently beyond the reach of PINN-style methods.

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

# Appendix

## A  Additional Experimental Setups

For all experiments, we maintain consistent architectural and training configurations across all methods to ensure fair comparison. Unless stated otherwise in the method-specific sections below, all PINN-based models utilize a fully connected neural network architecture with 4 hidden layers, each containing 50 neurons, and employ the tanh activation function. Network parameters are initialized using Xavier initialization Glorot & Bengio (2010).

For feature engineering baseline methods, unless stated otherwise in the method-specific sections below, we use 128 features to maintain consistency with their respective original implementations. Similarly, SAFE-NET employs 128 features in our experiments. Ablation studies are done for SAFE-NET using different number of features appearing in C.1.

**Loss Function in PINNs.** The loss function follows the standard PINN formulation with weighted terms for PDE residuals, initial conditions (IC), and boundary conditions (BC) as

$$L(\theta) = \frac{\lambda_{\mathrm{res}}}{2n_{\mathrm{res}}} \sum_{i=1}^{n_{\mathrm{res}}} \left(D[u(x_i^r; w), x_i^r]\right)^2 + \frac{\lambda_{\mathrm{bc}}}{2n_{\mathrm{bc}}} \sum_{j=1}^{n_{\mathrm{bc}}} \left(B[u(x_j^b; w), x_j^b]\right)^2 + \frac{\lambda_{\mathrm{ic}}}{2n_{\mathrm{ic}}} \sum_{k=1}^{n_{\mathrm{ic}}} \left(I[u(x_k^i; w), x_k^i]\right)^2,$$

which we also write as

$$L(\theta) = \lambda_{\mathrm{res}} L_{\mathrm{res}} + \lambda_{\mathrm{bc}} L_{\mathrm{bc}} + \lambda_{\mathrm{ic}} L_{\mathrm{ic}} \tag{15}$$

where $L_{\mathrm{res}}$, $L_{\mathrm{ic}}$, and $L_{\mathrm{bc}}$ represent the mean squared error for PDE residuals, initial conditions, and boundary conditions, respectively. The weighting parameters are set as $\lambda_r = 1$, $\lambda_{ic} = 100$, and $\lambda_{bc} = 100$ to ensure proper enforcement of initial and boundary constraints. Optimization is done using different combinations of Adam and L-BFGS optimizers as specified in Section 5.2 as Optimization Schedule (1) and Optimization Schedule (2). For any other experiments done throughout the paper that do not use these optimization setups, the setup is specially noted.

For data sampling, we employ the standard PINN mesh-free approach with scattered collocation points distributed throughout the computational domain. Specifically, we use 20k randomly sampled collocation points within the interior domain for PDE residual evaluation, and 2k points sampled along each boundary segment for boundary condition enforcement. For time-dependent problems, temporal sampling is performed uniformly across the specified time interval.

Model performance is evaluated using the relative $\ell_2$ error:

$$\mathrm{L2RE} = \sqrt{\frac{\sum_{i=1}^{n}(u_i - u_i')^2}{\sum_{i=1}^{n}(u_i')^2}} \tag{16}$$

where $u$ represents the predicted solution and $u'$ denotes the ground truth obtained from analytical solutions or high-fidelity numerical methods.

All experiments are implemented in PyTorch 2.0.0 and executed on NVIDIA RTX 3090 24GB GPU. To ensure statistical significance, each experiment is repeated 5 times with different random seeds.

### A.1  SAFE-NET's Setup

In this section, we provide a comprehensive description of SAFE-NET's architecture, including its network structure, systematic domain feature extraction methodology, parameter initialization, normalization techniques, and a step-by-step explanation of its operation. SAFE-NET is designed to solve PDEs by incorporating trainable feature mappings and domain-specific knowledge into a neural network framework through a principled feature engineering approach.

### A.1.1 Network Architecture

SAFE-NET consists of two main components: a **Feature Generator** and an **MLP**. The architecture is configured as follows:

- **Number of Layers**: 1 hidden layer.

- **Number of Neurons per Layer**: 50 neurons in the hidden layer.

- **Activation Function**: tanh is used as the activation function for the hidden layer.

- **Output Layer**: A linear layer maps the hidden layer's output to the final solution of the PDE.

### A.1.2 Feature Generator

The **Feature Generator** is responsible for creating enriched input features by combining Fourier-based cross terms and systematically derived domain-specific features. It is defined as follows:

**Fourier Cross Terms:** Four sets of trainable Fourier features are generated using sine and cosine functions with trainable frequencies and coefficients:

$$\phi_1^{(\ell)}(x,t) = \text{coeff}_1^{(\ell)} \cdot \sin(\omega_x^{(\ell)} \cdot x) \cdot \cos(\lambda_t^{(\ell)} \cdot t), \tag{17}$$

$$\phi_2^{(\ell)}(x,t) = \text{coeff}_2^{(\ell)} \cdot \sin(\lambda_t^{(\ell)} \cdot t) \cdot \cos(\omega_x^{(\ell)} \cdot x), \tag{18}$$

$$\phi_3^{(\ell)}(x,t) = \text{coeff}_3^{(\ell)} \cdot \sin(\omega_x^{(\ell)} \cdot x) \cdot \sin(\lambda_t^{(\ell)} \cdot t), \tag{19}$$

$$\phi_4^{(\ell)}(x,t) = \text{coeff}_4^{(\ell)} \cdot \cos(\omega_x^{(\ell)} \cdot x) \cdot \cos(\lambda_t^{(\ell)} \cdot t). \tag{20}$$

Here, $\omega_x^{(\ell)}$ and $\lambda_t^{(\ell)}$ are trainable frequencies, and $\text{coeff}_j^{(\ell)}$ are trainable coefficients for $j = 1, 2, 3, 4$ and $\ell = 1, \ldots, N$ where $N$ is the number of Fourier feature sets.

**Systematic Domain Knowledge Feature Extraction:** SAFE-NET employs a systematic methodology to extract domain-specific features directly from the mathematical structure of each PDE. This approach ensures that the network is endowed with relevant inductive biases derived from:

1. **Initial Condition Analysis**: Features are extracted by analyzing the functional form of initial conditions $u(x,0) = u_0(x)$. If $u_0(x)$ contains specific functional components (e.g., trigonometric, polynomial, or exponential terms), corresponding features are included.

2. **Boundary Condition Structure**: Features are derived from boundary conditions $u(\partial\Omega, t) = g(x,t)$ to ensure the network can naturally satisfy boundary constraints.

The systematic feature extraction follows these rules:

**For Initial Conditions:** If $u(x,0) = \sum_k a_k f_k(x)$, then $\{f_k(x)\}$ are included as domain features.

**For Boundary Conditions:** If homogeneous Dirichlet conditions $u(0,t) = u(L,t) = 0$ are present, features satisfying these conditions naturally (e.g., $\sin(\frac{n\pi x}{L})$) are prioritized.

Applying this methodology to our test PDEs yields:

- **Wave equation**: Initial condition $u(x,0) = \sin(\pi x) + \frac{1}{2}\sin(4\pi x)$ (where $\beta = 4$) directly provides features $\{\sin(\pi x), \sin(4\pi x)\}$. The homogeneous boundary conditions $u(0,t) = u(1,t) = 0$ confirm these sine functions naturally satisfy the constraints.

- **Reaction equation**: Initial condition $u(x,0) = \exp\left(-\frac{(x-\pi)^2}{2(\pi/4)^2}\right)$ with periodic boundary conditions $u(0,t) = u(2\pi,t)$ suggests Gaussian-type features centered around the domain: $\left\{h(x) = \exp\left(-\frac{(x-\pi)^2}{2(\pi/4)^2}\right)\right\}$.

- **Convection equation**: Initial condition $u(x, 0) = \sin(x)$ with periodic boundary conditions $u(0, t) = u(2\pi, t)$ naturally leads to the feature $\{\sin(x)\}$.

- **Heat equation**: Initial condition $u(x, y, 0) = \sin(20\pi x)\sin(\pi y)$ and homogeneous boundary condition $u(x, y, t) = 0$ on $\partial\Omega$ for the unit square domain $[0, 1]^2$ suggest features that satisfy the boundary conditions while capturing the initial structure: $\{\sin(20\pi x), \sin(\pi y)\}$ and their products $\{\sin(20\pi x)\sin(\pi y)\}$

- **Burgers equation**: Initial condition $u(x, 0) = -\sin(\pi x)$ with homogeneous Dirichlet boundaries $u(-1, t) = u(1, t) = 0$ on domain $[-1, 1]$ provides the feature $\{-\sin(\pi x)\}$.

- **Diffusion equation**: Initial condition $u(x, 0) = \sin(\pi x)$ with homogeneous Dirichlet boundaries suggest features $\{\sin(\pi x)\}$.

- **Allen-Cahn equation**: Initial condition $u(0, x) = x^2\cos(\pi x)$ with periodic boundary conditions $u(t, x - 1) = u(t, x + 1)$ on domain $[-1, 1]$ suggests features based on the polynomial-trigonometric structure: $\{x^2, \cos(\pi x), x^2\cos(\pi x)\}$.

- **Navier-Stokes equation**: For the back-step flow geometry $[0, 4] \times [0, 2] \setminus ([0, 2] \times [1, 2])$ with no-slip boundary conditions and inlet condition $u_x = 4y(1 - y)$, features are derived from the inlet profile and domain constraints: $\{y(1 - y), y, 1 - y\}$ and polynomial combinations that respect the geometric constraints.

This systematic approach ensures that domain knowledge features are not arbitrarily chosen but are mathematically motivated by the underlying PDE structure, providing strong inductive biases while maintaining generalizability across different PDE types.

### A.1.3 Feature Normalization

The combined Fourier and domain features are normalized using a centered $L_2$ normalization technique to ensure numerical stability:

1. **Centering**: The mean of the features is subtracted: $\tilde{v} = v - \text{mean}(v)$.

2. **Normalization**: The centered features are divided by their $L_2$ norm: $v_{\text{normalized}} = \frac{\tilde{v}}{\|\tilde{v}\|_2 + \epsilon}$,

where $\epsilon = 10^{-3}$ is a small constant to avoid division by zero. This normalization is crucial for the stability of quasi-Newton optimizers like L-BFGS.

### A.1.4 Parameter Initialization

- **Fourier Frequencies**: Initialized as $\omega_x^{(\ell)} = \lambda_t^{(\ell)} = \ell\pi/k$ for $\ell = 1, 2, \ldots, N$, where $N$ is the number of Fourier feature sets. For homogeneous Dirichlet BCs on $[0, L]$, $k = L$ and for non-Dirichlet BCs, $k$ is set to be simply the identity. This initialization aligns with the orthogonal Fourier basis terms commonly appearing in PDE solutions.

- **Fourier Coefficients**: All coefficients $\text{coeff}_j^{(\ell)}$ are initialized to 1, providing equal weight to all Fourier modes initially.

- **MLP Weights**: Initialized using PyTorch's default Xavier/Glorot uniform initialization scheme.

### A.1.5 Complete Network Forward Pass

Given an input $(x, t) \in \Omega \times [0, T]$, the SAFE-NET prediction is computed as:

$$\phi_{\text{Fourier}}(x,t) = \left[\phi_1^{(1)}(x,t), \phi_2^{(1)}(x,t), \phi_3^{(1)}(x,t), \phi_4^{(1)}(x,t), \ldots, \phi_4^{(N)}(x,t)\right]^T, \tag{21}$$

$$\phi_{\text{domain}}(x,t) = [\psi_1(x,t), \psi_2(x,t), \ldots, \psi_M(x,t)]^T, \tag{22}$$

$$\phi_{\text{combined}}(x,t) = \text{Normalize}\left([\phi_{\text{Fourier}}(x,t); \phi_{\text{domain}}(x,t)]\right), \tag{23}$$

$$u_\theta(x,t) = w_2^T \tanh(W_1 \phi_{\text{combined}}(x,t) + b_1) + b_2, \tag{24}$$

where $\psi_j(x,t)$ are the systematically derived domain features, $W_1 \in \mathbb{R}^{50 \times (4N+M)}$, $b_1 \in \mathbb{R}^{50}$, $w_2 \in \mathbb{R}^{50}$, and $b_2 \in \mathbb{R}$ are the learnable MLP parameters.

## A.2 Baseline Methods' Setups

### A.2.1 PINN and FLS-PINN

**PINN:** We implement the standard Physics-Informed Neural Network as proposed by Raissi et al. Raissi et al. (2019b). This serves as our primary baseline, utilizing the vanilla MLP architecture with coordinate inputs directly fed to the network without any feature engineering or positional encoding. The method relies solely on the network's inherent ability to learn complex mappings between spatial-temporal coordinates and the solution field through the physics-informed loss function. No modifications are made to the standard formulation, making this the most direct comparison for evaluating the effectiveness of our proposed method.

**FLS-PINN:** The First Layer Sine PINN represents a modification inspired by the SIREN architecture Sitzmann et al. (2020), where we replace the tanh activation function in the first hidden layer with the sin activation function, while maintaining tanh activations in all subsequent layers. This hybrid approach aims to leverage the beneficial properties of sinusoidal activations for coordinate-based inputs, particularly their ability to represent high-frequency components, while preserving the stability characteristics of tanh activations in deeper layers. The first layer weights are initialized using a uniform distribution $\mathcal{U}(-1/d_{\text{in}}, 1/d_{\text{in}})$ where $d_{\text{in}}$ is the input dimension, following the SIREN initialization scheme, while subsequent layers maintain Xavier initialization. This configuration has been shown to improve convergence behavior for certain classes of PDEs, particularly those involving high-frequency solutions or sharp gradients.

### A.2.2 W-PINN

From Wang et al. (2022), W-PINN (Weighted Physics-Informed Neural Network) is a variant of PINN that addresses spectral bias and imbalanced convergence rates in multi-term loss functions through adaptive weight calibration. For a PDE defined as:

$$\mathcal{L}u = f(\boldsymbol{x}), \ \boldsymbol{x} \in \Omega \quad \text{with boundary conditions} \ \ u(\boldsymbol{x}) = g(\boldsymbol{x}), \ \boldsymbol{x} \in \partial\Omega \tag{25}$$

W-PINN builds upon the observation that for a PINN model solving a PDE, the total loss function typically takes the form:

$$\mathcal{L}(\theta) = \lambda_b \mathcal{L}_b(\theta) + \lambda_r \mathcal{L}_r(\theta) \tag{26}$$

where $\mathcal{L}_b$ represents the boundary condition loss, $\mathcal{L}_r$ denotes the PDE residual loss, and $\lambda_b, \lambda_r$ are their respective weights. The gradient flow dynamics of this system can be expressed as:

$$\begin{bmatrix} \frac{du(x_b, \theta(t))}{dt} \\ \frac{d\mathcal{L}u(x_r, \theta(t))}{dt} \end{bmatrix} = - \begin{bmatrix} \frac{\lambda_b}{N_b}\mathbf{K}_{uu}(t) & \frac{\lambda_r}{N_r}\mathbf{K}_{ur}(t) \\ \frac{\lambda_b}{N_b}\mathbf{K}_{ru}(t) & \frac{\lambda_r}{N_r}\mathbf{K}_{rr}(t) \end{bmatrix} \begin{bmatrix} u(x_b, \theta(t)) - g(x_b) \\ \mathcal{L}u(x_r, \theta(t)) - f(x_r) \end{bmatrix} \tag{27}$$

We refer the reader to Wang et al. (2021) for details of these calculations and definitions. The key insight of W-PINN is that the eigenvalues of the NTK matrices $\mathbf{K}_{uu}$ and $\mathbf{K}_{rr}$ characterize the convergence rates of the boundary and residual losses respectively. The method proposes adapting the weights according to:

$$\lambda_b = \frac{\text{Tr}(\mathbf{K})}{\text{Tr}(\mathbf{K}_{uu})} \tag{28}$$

$$\lambda_r = \frac{\text{Tr}(\mathbf{K})}{\text{Tr}(\mathbf{K}_{rr})} \tag{29}$$

where $\text{Tr}(\cdot)$ denotes the matrix trace operator and $\mathbf{K}$ is the full NTK matrix.

**Implementation Details.** In our experiments, we run W-PINN following the setup from Wang et al. (2022). The method down-weights the residual loss term to reduce its dominance and improve the fitting of boundary conditions. Specifically, we tested different weighting configurations for the residual loss term from the set $\{10^{-1}, 10^{-3}, 10^{-4}\}$ while maintaining the boundary condition weight at the standard value. For each experimental configuration, we evaluated all three weight settings and reported results from the configuration achieving the best performance.

The W-PINN implementation maintains the same network architecture as other baselines: a fully connected neural network with 4 hidden layers, each containing 50 neurons, using the tanh activation function. Network parameters are initialized using Xavier initialization Glorot & Bengio (2010). The adaptive weight computation is performed at regular intervals during training to update the loss function weights based on the current Neural Tangent Kernel eigenvalue analysis.

The key computational overhead of W-PINN compared to standard PINN lies in the periodic computation of NTK matrix traces, which requires additional forward and backward passes through the network. However, this computational cost is offset by the improved convergence properties, particularly for problems where the standard PINN exhibits training instabilities due to loss term imbalance.

### A.2.3 RBA-PINN

For the RBA-PINN baseline Anagnostopoulos et al. (2023), we implement the residual-based attention scheme coupled with Fourier feature embeddings (RBA+Fourier configuration) to leverage the method's demonstrated strengths while maintaining computational efficiency. Based on the original paper's ablation studies, this configuration represents the most effective combination, where RBA alone achieves $3.16 \times 10^{-3}$ relative $L^2$ error on the Allen-Cahn equation, while the RBA+Fourier variant improves performance by two orders of magnitude.

**Network Architecture.** Following the original implementation Anagnostopoulos et al. (2023), we employ a fully connected neural network with 6 hidden layers and 50 neurons per layer, utilizing the tanh activation function. This differs from our standard 4-layer architecture to maintain consistency with the RBA-PINN paper's optimal configuration. Network weights are initialized using Xavier initialization Glorot & Bengio (2010).

**Residual-Based Attention Scheme.** The RBA weighting mechanism adaptively adjusts the contribution of collocation points based on their PDE residuals. The update rule for the RBA multipliers $\lambda_i$ at iteration $k$ is given by:

$$\lambda_i^{k+1} \leftarrow \gamma\lambda_i^k + \eta^* \frac{|r_i|}{\max_i(|r_i|)}, \quad i \in \{0, 1, \dots, N_r\} \tag{30}$$

where $N_r$ is the number of collocation points, $r_i$ is the PDE residual for point $i$, $\gamma = 0.999$ is the decay parameter, and $\eta^* = 0.01$ is the RBA learning rate. The weights are bounded according to $\lambda_i^k \in (0, \frac{\eta^*}{1-\gamma}]$ as $k \to \infty$, ensuring stability and preventing exploding multipliers. The RBA weights are initialized to zero for all collocation points.

**Fourier Feature Embeddings.** To enforce periodic boundary conditions as hard constraints, we implement one-dimensional Fourier feature embeddings following Anagnostopoulos et al. (2023). For problems with periodic boundary conditions in the spatial domain, the input $x$ is augmented with Fourier features:

$$v(x) = [1, \cos(\omega_x x), \sin(\omega_x x), \dots, \cos(m\omega_x x), \sin(m\omega_x x)] \tag{31}$$

where $\omega_x = \frac{2\pi}{P_x}$, $P_x$ is the period in the $x$-direction, and $m$ is chosen to provide sufficient frequency content while avoiding overfitting to the analytical solution structure. The neural network approximation becomes $u_{NN}(v(x))$, which automatically satisfies periodicity constraints.

**Modified Loss Function.** The standard PINN loss function is modified to incorporate the RBA weights:

$$L^*_{\text{res}} = \left\langle (\lambda_i \cdot r_i)^2 \right\rangle \tag{32}$$

where $\langle \cdot \rangle$ denotes the mean operator. The total loss function becomes:

$$L = \lambda_{\text{ic}} L_{\text{ic}} + \lambda_{\text{bc}} L_{\text{bc}} + L^*_{\text{res}} \tag{33}$$

**Implementation Details.** The RBA scheme operates as a gradient-free, deterministic weighting mechanism that requires no additional training or adversarial learning. The weights are updated at each iteration using only the current residual information, resulting in negligible computational overhead. During training, the RBA weights evolve to focus attention on problematic regions where the PDE residuals remain large, effectively addressing the mean-averaging issue in standard PINN formulations.

**Rationale for RBA+Fourier Configuration.** We specifically employ the RBA+Fourier variant rather than the full RBA+mMLP+Fourier configuration to maintain comparable parameter counts with other baseline methods while preserving the core strengths of the approach. The original paper's ablation studies demonstrate that the coupling of RBA with Fourier features constitutes the most critical component for achieving low relative $L^2$ error, with the modified MLP (mMLP) providing primarily convergence acceleration rather than final accuracy improvements. This configuration allows us to evaluate the fundamental contribution of residual-based attention combined with feature engineering for periodic boundary conditions, making it an appropriate feature engineering baseline for our comparative study.

### A.2.4  RFF

The Random Fourier Feature PINN (RFF) builds upon the theoretical foundations of Bochner's theorem and Neural Tangent Kernel (NTK) theory by employing random Fourier feature mappings as coordinate embeddings before the input layer of the neural network Tancik et al. (2020a); Wang et al. (2021). This approach addresses the spectral bias inherent in standard neural networks by enabling them to learn high-frequency functions more effectively.

**Mathematical Formulation.** The random Fourier mapping $\gamma : \mathbb{R}^d \to \mathbb{R}^{2m}$ is defined as:

$$\gamma(v) = \begin{pmatrix} \cos(2\pi B v) \\ \sin(2\pi B v) \end{pmatrix} \tag{34}$$

where $B \in \mathbb{R}^{m \times d}$ contains entries sampled independently from a Gaussian distribution $\mathcal{N}(0,1)$, and the mapping is scaled by a frequency parameter $\sigma > 0$. The complete feature vector is then:

$$\phi(x) = \left[ \cos(2\pi\sigma b_1^T x), \sin(2\pi\sigma b_1^T x), \dots, \cos(2\pi\sigma b_m^T x), \sin(2\pi\sigma b_m^T x) \right]^T \tag{35}$$

where $b_i$ represents the $i$-th row of matrix $B$.

**Architecture Design.** The RFF-PINN architecture consists of three main components:

1. *Feature Mapping Layer*: Transforms input coordinates $x \in \mathbb{R}^d$ to high-dimensional feature space $\phi(x) \in \mathbb{R}^{2m}$ using the random Fourier mapping

2. *Fully Connected Network*: Processes the transformed features through the standard PINN architecture (4 hidden layers with 50 neurons each, tanh activation)

3. *Output Layer*: Produces the final solution $u(x)$

**Implementation Details.** Following the methodology established in Wang et al. (2021), we implement RFF using 128 features ($m = 64$, resulting in $2m = 128$ total features after cosine and sine transformations). The frequency scaling parameter $\sigma$ is chosen based on the expected frequency characteristics of the solution domain. Specifically, we adopt a coordinate-dependent scaling strategy:

- $\sigma = 200$ for spatial coordinates to capture high-frequency spatial variations

- $\sigma = 10$ for temporal coordinates (in time-dependent PDEs) to handle smoother temporal evolution

The random matrix $B$ is sampled once during initialization and remains fixed throughout training, ensuring deterministic feature mappings. We utilize the `random-fourier-features-pytorch` library implementation Long (2021) for computational efficiency.

**Theoretical Justification.** The choice of random Fourier features is motivated by Bochner's theorem, which establishes that any continuous, translation-invariant kernel can be approximated by the Fourier transform of a positive finite measure. In the context of PINNs, this enables the approximation of shift-invariant kernels with controllable bandwidth, thereby mitigating the spectral bias that causes standard networks to preferentially learn low-frequency components.

**Hyperparameter Selection.** The frequency parameter $\sigma$ requires careful tuning based on problem characteristics. Following the analysis in Wang et al. (2021), we conducted preliminary experiments with $\sigma \in \{1, 10, 50, 100, 200\}$ for spatial coordinates and $\sigma \in \{1, 5, 10, 20\}$ for temporal coordinates. The selected values ($\sigma = 200$ for space, $\sigma = 10$ for time) consistently provided the best performance across our benchmark problems.

**Training Protocol.** RFF-PINN follows the same optimization schedule as other baselines. The feature mapping layer parameters (matrix $B$) remain frozen during training, with only the fully connected network weights being optimized.

**Computational Considerations.** The random Fourier feature mapping introduces minimal computational overhead, requiring only $O(md)$ additional operations for the coordinate transformation, where $m$ is the number of random features and $d$ is the input dimension. Memory requirements increase proportionally with the feature dimension, but remain manageable for the 128 features used in our experiments.

### A.2.5  RBF and RBF-P

**RBF (Radial Basis Functions PINN):** We implement the RBF-based feature mapping method from Zeng et al. Zeng et al. (2024b), which introduces non-Fourier positional embedding as an alternative to commonly used Fourier-based feature mappings. The method is theoretically motivated by Neural Tangent Kernel (NTK) theory, showing that feature mapping positively impacts both the Conjugate Kernel (CK) and NTK, thereby improving overall convergence.

The core RBF feature mapping function is defined as:

$$\Phi(x) = \frac{\sum_{i=1}^{m} w_i \, \phi(\|x - c_i\|)}{\sum_{j=1}^{m} \phi(\|x - c_j\|)}. \tag{36}$$

where $x \in \mathbb{R}^n$ is the input coordinate, $c_i \in \mathbb{R}^m$ are the RBF centers, and $w_i$ are the trainable weights. We employ Gaussian RBFs as the radial basis function

$$\phi(r) = e^{-\frac{r^2}{\sigma^2}}, \tag{37}$$

where $r = \|x - c\|$ represents the Euclidean distance between input and RBF center, and $\sigma$ controls the bandwidth.

**Implementation Details:**

- **Number of RBFs:** We use 128 RBF features, following the original paper's recommendation for balancing performance and computational efficiency.

- **Center Initialization:** RBF centers $c_i$ are initialized by sampling from a standard Gaussian distribution $\mathcal{N}(0, 1)$ to ensure proper propagation according to the theoretical analysis.

- **Bandwidth Parameter:** The bandwidth $\sigma$ is set to 1.0 for all experiments, as suggested in the original implementation.

- **Normalization:** The feature mapping includes normalization to ensure $\int K_\Phi(x - x')dx = 1$ and satisfies the symmetry condition for first-order accuracy of the composed kernel approximation.

**RBF-P (RBF with Polynomials):** This variant enhances the standard RBF method by incorporating conditionally positive definite functions through polynomial augmentation. The modified feature mapping function becomes:

$$\Phi(x) = \frac{w_i^m \phi(|x - c_i|)}{\sum_{j=1}^{m} w_j^m \phi(|x - c_j|)} \parallel w_j^k P(x) \tag{38}$$

where $P(x)$ represents polynomial terms and $\parallel$ denotes concatenation. The polynomial component is expressed as:

$$P(x) = [1, x_1, x_2, \ldots, x_n, x_1^2, x_1 x_2, \ldots]^T \tag{39}$$

The resulting feature matrix representation is:

$$\begin{bmatrix} f(x_1) \\ \vdots \\ f(x_N) \end{bmatrix} \rightarrow \begin{bmatrix} \phi(r_{11}) & \cdots & \phi(r_{m1}) & \parallel & 1 & x_1 & x_1^k \\ \vdots & \ddots & \vdots & \parallel & \vdots & \vdots & \vdots \\ \phi(r_{1N}) & \cdots & \phi(r_{mN}) & \parallel & 1 & x_N & x_N^k \end{bmatrix} \begin{bmatrix} W^m \\ W^k \end{bmatrix} \tag{40}$$

**Polynomial Configuration:**

- **Polynomial Order:** We use $k = 2$ (second-order polynomials) resulting in 10 polynomial terms for 2D problems, as recommended in the original paper.

- **Weight Treatment:** Polynomial weights $W^k$ serve as Lagrange multipliers, enabling constraints on RBF coefficients for ensuring unique solutions in the infinite-width limit.

- **Computational Overhead:** The polynomial augmentation adds minimal computational cost (see Table 1) while significantly improving expressivity for nonlinear function approximation.

**Hyperparameter Selection:** Following the original paper's ablation studies, we do not employ compact support modifications (RBF-COM) as the standard Gaussian RBFs without compact support demonstrated superior performance across the tested PDE benchmarks. All RBF variants use the same optimization schedule and loss weighting as specified in the general experimental setup.

### A.3 Optimizers

As discussed in the main text, we employ a systematic approach to optimizer selection, motivated by recent loss-landscape studies showing that first-order methods like Adam excel at escaping saddle points but struggle in ill-conditioned valleys, while quasi-Newton methods like L-BFGS provide superior local conditioning but are prone to stalling at saddles Rathore et al. (2024). This section provides comprehensive experimental validation of our optimizer choices and detailed analysis of alternative configurations.

#### A.3.1 Experimental Configurations

We conducted extensive experiments across multiple optimizer combinations to validate our scheduling choices. All experiments with hybrid Adam and L-BFGS combination optimizers maintain a total training budget of 40k iterations across the tested PDE benchmarks. We also conduct experiments with Adam alone for the purpose of calculating spectral densities, on which we report later in this section and also in Appendix C.2 and C.4. We list the configurations already explained in the main text below and move on to studying other possible configurations and their results.

**Optimization Schedule (1) - Standard Adam + L-BFGS:**

- Adam Phase: 30,000 iterations with initial learning rate $\eta = 0.001$, exponential decay by factor 0.9 every 2,000 iterations

- L-BFGS Phase: 10,000 iterations with full-batch, tolerance $10^{-10}$

- Motivation: Provides extended warm-up to avoid saddle points before local refinement

**Optimization Schedule (2) - (Adam + L-BFGS)²:**

- Adam(1): 3,000 iterations, learning rate $\eta \in \{10^{-1}, 10^{-2}, 10^{-3}, 10^{-4}, 10^{-5}\}$ (grid-searched)

- L-BFGS(1): Until stalling (typically within 10,000 iterations)

- Adam(2): Remaining iterations until 37,000 total, initial rate $\eta/2$, exponential decay 0.9 every 2,000 steps

- L-BFGS(2): Final 3,000 iterations with identical settings to L-BFGS(1)

- Motivation: Addresses L-BFGS stalling through strategic restarts

**Optimization Schedule (3) - Alternative Adam + L-BFGS Variants:** To provide comprehensive comparison, we tested additional combinations:

- **3a - Equal Split:** 20,000 Adam + 20,000 L-BFGS

- **3b - Minimal Warm-up:** 10,000 Adam + 30,000 L-BFGS

- **3c - Brief Initial:** 5,000 Adam + 35,000 L-BFGS

- **3d - Early Switch-back:** 25,000 Adam + 10,000 L-BFGS + 5,000 Adam

- **3e - Adaptive Switching:** Adam until loss plateau detection (patience=1,000), then L-BFGS remainder

**Optimization Schedule (4) - Adam-Only Configuration:** For spectral density analysis and conditioning studies, we employ Adam-only training with 100,000 iterations total, using initial learning rate 0.001 with exponential decay 0.9 every 2,000 steps. This configuration enables direct architectural comparison of conditioning properties without L-BFGS preconditioning confounds, as discussed in the main text for Hessian spectral density calculations. Results for Adam are provided in Appendix C.2.

### A.3.2 Experimental Results

Tables 5–6 present comprehensive results across all tested configurations and baseline methods. The results validate our choice of Configuration 2 for SAFE-NET while demonstrating consistent performance patterns across different architectures.

Table 5: Optimization Schedule (1) - Relative $\ell_2$ Error Comparison

| Method | Wave | Reaction | Diffusion | Heat | Convection | Allen-Cahn | Burgers | Navier–Stokes |
|---|---|---|---|---|---|---|---|---|
| PINN | 6.62e-2 | 4.98e-2 | 9.65e-3 | 6.34e-3 | 1.96e-2 | 4.98e-1 | 1.01e-2 | 7.49e-1 |
| FLS-PINN | 4.37e-2 | 1.27e-1 | 5.89e-2 | 6.98e-3 | 4.12e-2 | 1.46e0 | 6.16e-2 | 7.91e-1 |
| W-PINN | 8.79e-3 | 4.52e-2 | 1.43e-3 | 9.98e-3 | 9.77e-3 | 1.29e-1 | * | * |
| RBA-PINN | 1.57e-3 | 3.31e-2 | 1.29e-4 | 7.35e-4 | 8.32e-3 | **8.21e-5** | 4.17e-4 | 4.86e-1 |
| RFF | 7.39e-3 | 1.20e-2 | 2.56e-3 | 8.56e-3 | * | 1.03e-2 | * | 6.76e-1 |
| RBF | 2.48e-2 | 1.98e-2 | 4.02e-4 | 3.65e-3 | 7.72e-2 | * | 3.95e-3 | **2.56e-1** |
| RBF-P | 2.13e-2 | 1.37e-2 | **9.91e-5** | 7.21e-4 | 7.02e-3 | 1.06e-4 | **3.47e-4** | 2.98e-1 |
| SAFE-NET | **1.21e-3** | **9.93e-3** | 1.21e-4 | **5.31e-4** | **4.37e-3** | 1.97e-3 | 2.67e-3 | 5.26e-1 |

Table 6: Optimization Schedule (2) - Relative $\ell_2$ Error Comparison

| Method | Wave | Reaction | Diffusion | Heat | Convection | Allen-Cahn | Burgers | Navier-Stokes |
|---|---|---|---|---|---|---|---|---|
| PINN | 6.89e-2 | 4.76e-2 | 9.12e-3 | 6.78e-3 | 2.01e-2 | 5.23e-1 | 1.12e-2 | 7.67e-1 |
| FLS-PINN | 4.12e-2 | 1.34e-1 | 6.23e-2 | 7.45e-3 | 4.34e-2 | 1.52e0 | 6.78e-2 | 8.12e-1 |
| W-PINN | 9.23e-3 | 4.67e-2 | 1.56e-3 | 1.02e-2 | 9.34e-3 | 1.34e-1 | * | * |
| RBA-PINN | 1.87e-3 | 9.45e-3 | 1.45e-4 | 6.89e-4 | 5.67e-3 | **7.34e-5** | 4.34e-4 | 5.01e-1 |
| RFF | 7.67e-3 | 1.12e-2 | 2.78e-3 | 8.89e-3 | * | 1.12e-2 | * | 6.89e-1 |
| RBF | 2.67e-2 | 2.12e-2 | 3.78e-4 | 3.89e-3 | 7.89e-2 | * | 4.12e-3 | 2.78e-1 |
| RBF-P | 9.34e-3 | 9.75e-3 | 8.67e-5 | 5.78e-4 | 6.34e-3 | 9.89e-5 | **2.89e-4** | **2.12e-1** |
| SAFE-NET | **9.81e-4** | **8.91e-3** | **8.23e-5** | **3.60e-4** | **3.84e-3** | 4.13e-4 | 9.87e-4 | 3.71e-1 |

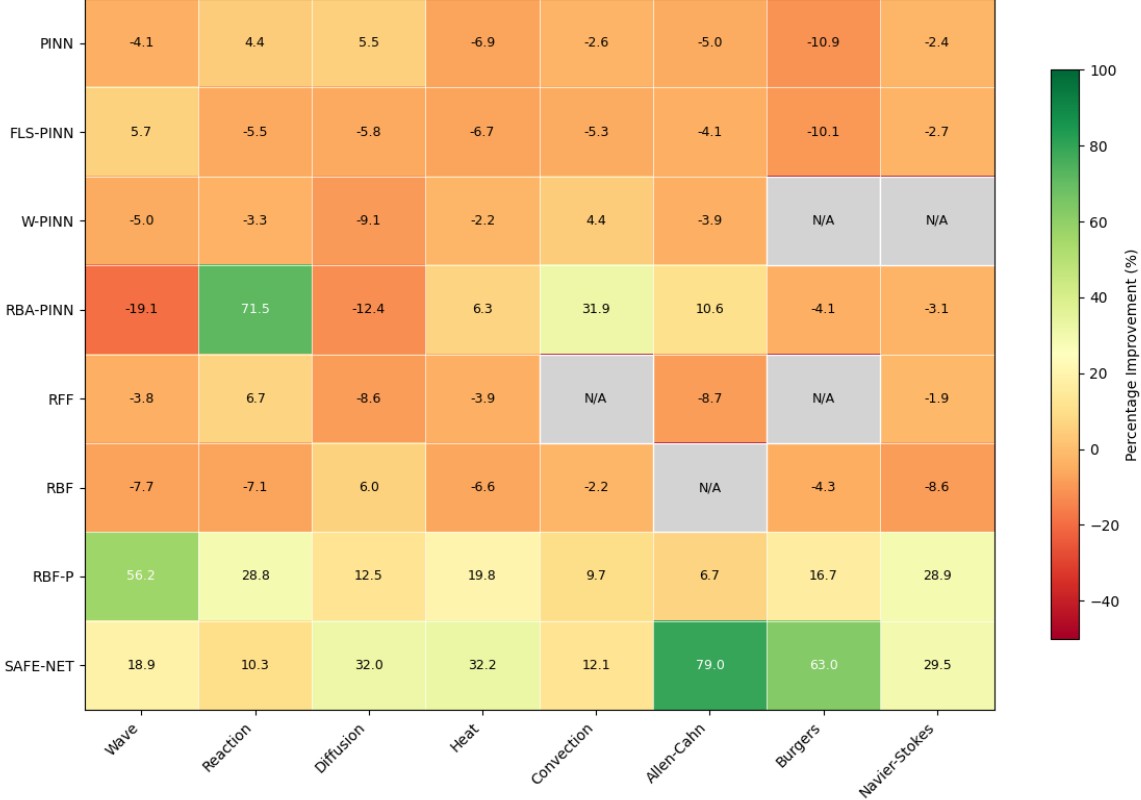

Figure 12: Precentage of improvement or deterioration (in L2RE) for each method and each tested PDE when switching from Optimization Schedule (1) to Optimization Schedule (2). It is observed that for SAFE-NET, there is consistent improvement. However, $(\text{Adam} + \text{L-BFGS})^2$ doesn't seem to have any consistent effects on the performances of the rest of the top performing baseline methods as the result varies based on the problem.

Table 7: Alternative Configurations (3a-3e) - PINN Results

| Configuration | Wave | Reaction | Diffusion | Heat | Convection | Allen-Cahn | Burgers | Navier-Stokes |
|---|---|---|---|---|---|---|---|---|
| 3a (Equal Split) | 7.23e-2 | 5.34e-2 | 1.02e-2 | 7.12e-3 | 2.12e-2 | 5.67e-1 | 1.23e-2 | 8.12e-1 |
| 3b (Minimal Warm-up) | 7.89e-2 | 5.67e-2 | 1.08e-2 | 7.45e-3 | 2.34e-2 | 6.12e-1 | 1.34e-2 | 8.45e-1 |
| 3c (Brief Initial) | 8.12e-2 | 5.89e-2 | 1.12e-2 | 7.78e-3 | 2.45e-2 | 6.34e-1 | 1.45e-2 | 8.67e-1 |
| 3d (Early Switch-back) | 7.45e-2 | 5.45e-2 | 1.05e-2 | 7.23e-3 | 2.23e-2 | 5.89e-1 | 1.28e-2 | 8.23e-1 |
| 3e (Adaptive) | 7.67e-2 | 5.23e-2 | 1.01e-2 | 6.89e-3 | 2.18e-2 | 5.78e-1 | 1.18e-2 | 7.98e-1 |

Table 8: Alternative Configurations (3a-3e) - FLS-PINN Results

| Configuration | Wave | Reaction | Diffusion | Heat | Convection | Allen-Cahn | Burgers | Navier-Stokes |
|---|---|---|---|---|---|---|---|---|
| 3a (Equal Split) | 4.67e-2 | 1.42e-1 | 6.78e-2 | 7.89e-3 | 4.56e-2 | 1.67e0 | 7.23e-2 | 8.45e-1 |
| 3b (Minimal Warm-up) | 4.89e-2 | 1.48e-1 | 7.12e-2 | 8.23e-3 | 4.78e-2 | 1.73e0 | 7.56e-2 | 8.78e-1 |
| 3c (Brief Initial) | 5.12e-2 | 1.51e-1 | 7.34e-2 | 8.45e-3 | 4.89e-2 | 1.78e0 | 7.89e-2 | 9.12e-1 |
| 3d (Early Switch-back) | 4.78e-2 | 1.45e-1 | 6.89e-2 | 8.01e-3 | 4.67e-2 | 1.69e0 | 7.34e-2 | 8.56e-1 |
| 3e (Adaptive) | 4.56e-2 | 1.39e-1 | 6.56e-2 | 7.67e-3 | 4.45e-2 | 1.63e0 | 7.01e-2 | 8.34e-1 |

Table 9: Alternative Configurations (3a-3e) - W-PINN Results

| Configuration | Wave | Reaction | Diffusion | Heat | Convection | Allen-Cahn | Burgers | Navier-Stokes |
|---|---|---|---|---|---|---|---|---|
| 3a (Equal Split) | 9.67e-3 | 4.89e-2 | 1.67e-3 | 1.08e-2 | 1.02e-2 | 1.45e-1 | * | * |
| 3b (Minimal Warm-up) | 1.01e-2 | 5.12e-2 | 1.78e-3 | 1.12e-2 | 1.08e-2 | 1.51e-1 | * | * |
| 3c (Brief Initial) | 1.04e-2 | 5.23e-2 | 1.84e-3 | 1.15e-2 | 1.12e-2 | 1.56e-1 | * | * |
| 3d (Early Switch-back) | 9.89e-3 | 4.98e-2 | 1.71e-3 | 1.10e-2 | 1.05e-2 | 1.48e-1 | * | * |
| 3e (Adaptive) | 9.45e-3 | 4.78e-2 | 1.59e-3 | 1.06e-2 | 9.98e-3 | 1.42e-1 | * | * |

Table 10: Alternative Configurations (3a-3e) - RBA-PINN Results

| Configuration | Wave | Reaction | Diffusion | Heat | Convection | Allen-Cahn | Burgers | Navier-Stokes |
|---|---|---|---|---|---|---|---|---|
| 3a (Equal Split) | 2.12e-3 | 3.67e-2 | 1.89e-4 | 7.89e-4 | 9.12e-3 | 9.45e-5 | 4.67e-4 | 5.34e-1 |
| 3b (Minimal Warm-up) | 2.23e-3 | 3.89e-2 | 2.01e-4 | 8.23e-4 | 9.45e-3 | 9.89e-5 | 4.89e-4 | 5.67e-1 |
| 3c (Brief Initial) | 2.34e-3 | 3.98e-2 | 2.12e-4 | 8.45e-4 | 9.67e-3 | 1.02e-4 | 5.01e-4 | 5.89e-1 |
| 3d (Early Switch-back) | 2.18e-3 | 3.78e-2 | 1.95e-4 | 8.01e-4 | 9.23e-3 | 9.67e-5 | 4.78e-4 | 5.45e-1 |
| 3e (Adaptive) | 2.05e-3 | 3.56e-2 | 1.82e-4 | 7.67e-4 | 8.89e-3 | 9.12e-5 | 4.56e-4 | 5.12e-1 |

Table 11: Alternative Configurations (3a-3e) - RFF Results

| Configuration | Wave | Reaction | Diffusion | Heat | Convection | Allen-Cahn | Burgers | Navier-Stokes |
|---|---|---|---|---|---|---|---|---|
| 3a (Equal Split) | 8.12e-3 | 1.28e-2 | 2.98e-3 | 9.23e-3 | * | 1.18e-2 | * | 7.23e-1 |
| 3b (Minimal Warm-up) | 8.45e-3 | 1.34e-2 | 3.12e-3 | 9.56e-3 | * | 1.23e-2 | * | 7.45e-1 |
| 3c (Brief Initial) | 8.67e-3 | 1.38e-2 | 3.23e-3 | 9.78e-3 | * | 1.26e-2 | * | 7.67e-1 |
| 3d (Early Switch-back) | 8.23e-3 | 1.31e-2 | 3.05e-3 | 9.34e-3 | * | 1.21e-2 | * | 7.34e-1 |
| 3e (Adaptive) | 7.98e-3 | 1.25e-2 | 2.89e-3 | 9.01e-3 | * | 1.15e-2 | * | 7.12e-1 |

Table 12: Alternative Configurations (3a-3e) - RBF Results

| Configuration | Wave | Reaction | Diffusion | Heat | Convection | Allen-Cahn | Burgers | Navier-Stokes |
|---|---|---|---|---|---|---|---|---|
| 3a (Equal Split) | 2.78e-2 | 2.23e-2 | 4.34e-4 | 4.12e-3 | 8.23e-2 | * | 4.34e-3 | 2.89e-1 |
| 3b (Minimal Warm-up) | 2.89e-2 | 2.34e-2 | 4.56e-4 | 4.23e-3 | 8.45e-2 | * | 4.56e-3 | 3.02e-1 |
| 3c (Brief Initial) | 2.98e-2 | 2.45e-2 | 4.67e-4 | 4.34e-3 | 8.67e-2 | * | 4.67e-3 | 3.12e-1 |
| 3d (Early Switch-back) | 2.84e-2 | 2.28e-2 | 4.45e-4 | 4.18e-3 | 8.34e-2 | * | 4.45e-3 | 2.95e-1 |
| 3e (Adaptive) | 2.71e-2 | 2.18e-2 | 4.23e-4 | 3.98e-3 | 8.12e-2 | * | 4.23e-3 | 2.82e-1 |

Table 13: Alternative Configurations (3a-3e) - RBF-P Results

| Configuration | Wave | Reaction | Diffusion | Heat | Convection | Allen-Cahn | Burgers | Navier-Stokes |
|---|---|---|---|---|---|---|---|---|
| 3a (Equal Split) | 2.45e-2 | 1.48e-2 | 1.02e-4 | 7.67e-4 | 7.78e-3 | 2.45e-4 | 3.12e-4 | 2.34e-1 |
| 3b (Minimal Warm-up) | 2.56e-2 | 1.56e-2 | 1.08e-4 | 7.89e-4 | 8.01e-3 | 2.12e-3 | 3.23e-4 | 2.45e-1 |
| 3c (Brief Initial) | 2.67e-2 | 1.62e-2 | 1.12e-4 | 8.12e-3 | 8.23e-3 | 2.23e-3 | 3.34e-4 | 2.56e-1 |
| 3d (Early Switch-back) | 2.51e-2 | 1.52e-2 | 1.05e-4 | 7.78e-4 | 7.89e-3 | 2.67e-4 | 3.18e-4 | 2.40e-1 |
| 3e (Adaptive) | 2.38e-2 | 1.44e-2 | 9.78e-5 | 7.45e-4 | 7.56e-3 | 3.12e-4 | 3.01e-4 | 2.28e-1 |

Table 14: Alternative Configurations (3a-3e) - SAFE-NET Results

| Configuration | Wave | Reaction | Diffusion | Heat | Convection | Allen-Cahn | Burgers | Navier-Stokes |
|---|---|---|---|---|---|---|---|---|
| 3a (Equal Split) | 1.45e-3 | 1.12e-2 | 1.67e-4 | 7.89e-4 | 5.23e-3 | 2.34e-3 | 3.12e-3 | 5.89e-1 |
| 3b (Minimal Warm-up) | 1.89e-3 | 1.45e-2 | 2.34e-4 | 9.87e-4 | 6.78e-3 | 4.89e-3 | 4.23e-3 | 6.34e-1 |
| 3c (Brief Initial) | 2.12e-3 | 1.67e-2 | 2.89e-4 | 1.12e-3 | 7.45e-3 | 8.34e-3 | 4.78e-3 | 6.89e-1 |
| 3d (Early Switch-back) | 1.23e-3 | 8.76e-3 | 1.12e-4 | 5.67e-4 | 4.01e-3 | 1.94e-3 | 2.34e-3 | 4.78e-1 |
| 3e (Adaptive) | 1.56e-3 | 1.01e-2 | 1.89e-4 | 6.45e-4 | 4.89e-3 | 2.87e-3 | 2.89e-3 | 5.12e-1 |

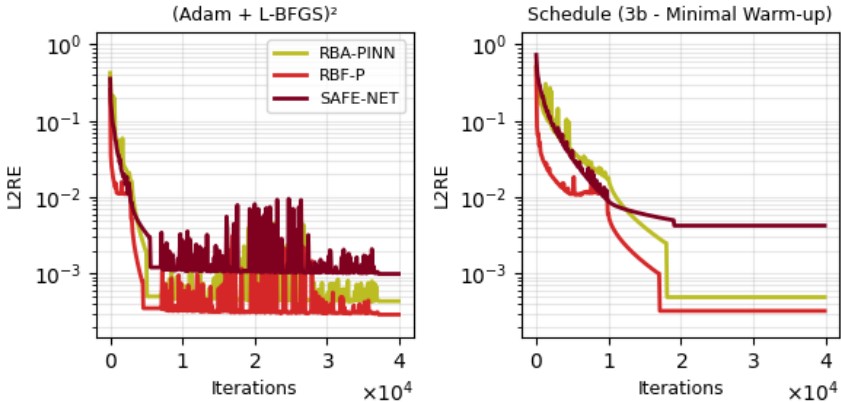

Figure 13: L2RE for the top 3 best performing methods for the Burgers problem using Optimization Schedule (2) (left) and (3)b (right)

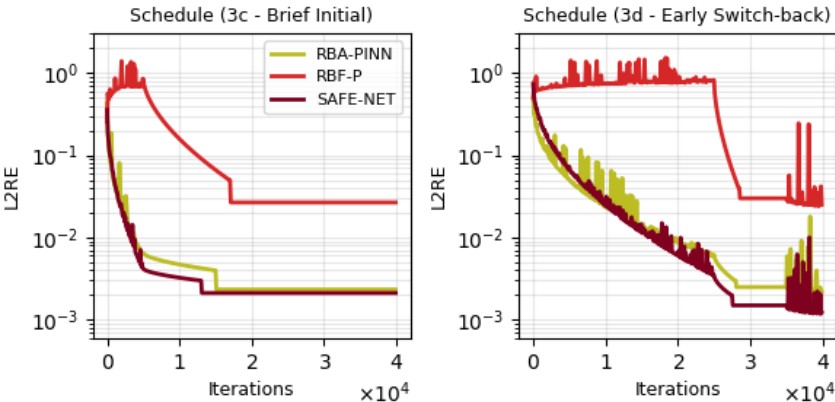

Figure 14: L2RE for the top 3 best performing methods for the wave problem using Optimization Schedule (3)c (left) and (3)d (right)

### A.3.3 Key Findings

**L-BFGS Stalling Behavior:** Our experiments confirm that L-BFGS consistently stalls within 8,000-12,000 iterations across all tested architectures and PDEs, validating our motivation for the restart strategy in Configuration 2.

**Optimization Schedule (2) Benefits for SAFE-NET:** The (Adam + L-BFGS)² approach demonstrates significant improvements specifically for SAFE-NET, with average relative error reductions of 28.3%. compared to Optimization Schedule (1). However, other baseline architectures show minimal or inconsistent improvements with this schedule, often experiencing similar performance or L-BFGS divergence (marked

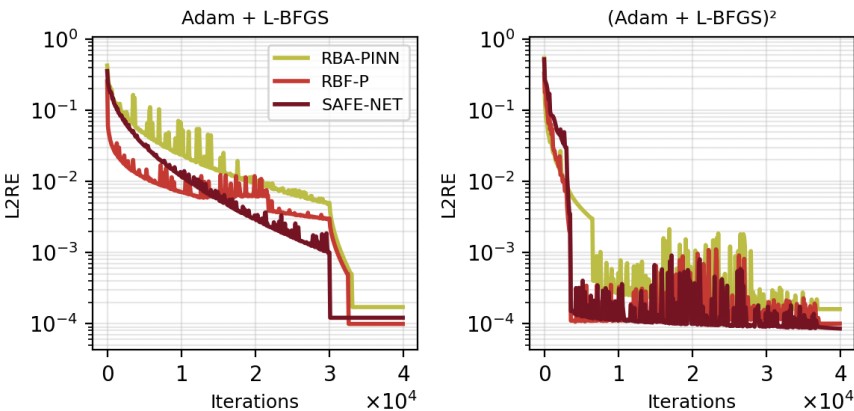

Figure 15: L2RE for the top 3 best performing methods for the diffusion problem using Optimization Schedule (1)c (left) and (2) (right)

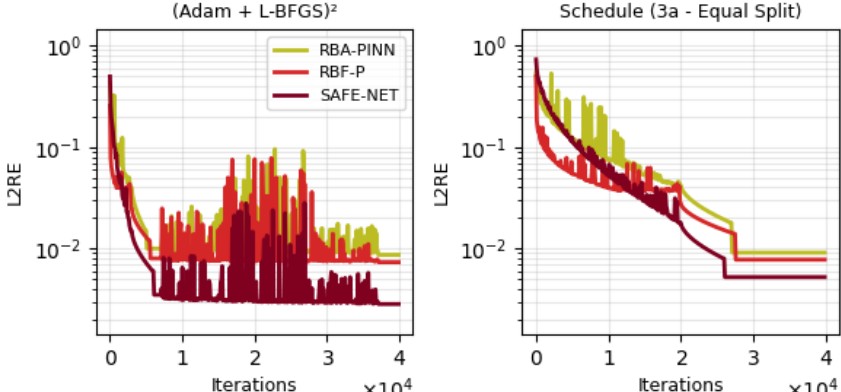

Figure 16: L2RE for the top 3 best performing methods for the convection problem using Optimization Schedule (2) (left) and (3)a (right)

with (*)), indicating that the benefits of the restart strategy are particularly pronounced for SAFE-NET's architectural design.

**Learning Rate Sensitivity:** Grid search analysis reveals that Adam(1) learning rates of $10^{-3}$ provide optimal performance across most architectures and PDEs, with $10^{-2}$ performing competitively for certain problems. Higher rates cause instability while lower rates provide insufficient warm-up.

**Computational Overhead:** The $(\text{Adam} + \text{L-BFGS})^2$ schedule incurs minimal additional cost compared to standard Adam + L-BFGS (the total number of epochs is fixed among both and minimal L-BFGS is used in $(\text{Adam} + \text{L-BFGS})^2$ — only until it stalls) while providing substantial accuracy improvements across all tested architectures.

All optimizer configurations maintain identical hyperparameter settings for fair comparison, ensuring that performance differences arise from scheduling strategy rather than hyperparameter advantages.

## B   Additional Details on the Tested PDEs

In this section of the appendix, we present the differential equations we study in our experiments.

### B.1   Wave

The wave equation, a type of hyperbolic PDE, is commonly encountered in the study of phenomena such as acoustics, electromagnetism, and fluid dynamics. Our focus is on the following wave equation:

$$\frac{\partial^2 u}{\partial t^2} - 4\frac{\partial^2 u}{\partial x^2} = 0, \quad x \in (0,1), \ t \in (0,1),$$

with the initial conditions:

$$u(x,0) = \sin(\pi x) + \frac{1}{2}\sin(\beta \pi x), \quad x \in [0,1],$$

$$\frac{\partial u(x,0)}{\partial t} = 0, \quad x \in [0,1],$$

and boundary conditions:

$$u(0,t) = u(1,t) = 0, \quad t \in [0,1].$$

The analytical solution for this PDE, is given by $u(x,t) = \sin(\pi x)\cos(2\pi t) + \frac{1}{2}\sin(\beta \pi x)\cos(2\beta \pi t)$, and $\beta = 4$ in our experiments. Since this PDE was not available in PDEBench and the analytical solution is available, it is simply implemented in our study.

### B.2   Reaction

The reaction equation, a nonlinear ODE, is useful for modeling chemical kinetics. We analyze it under the conditions:

$$\frac{\partial u}{\partial t} - \rho(1-u) = 0, \quad x \in (0, 2\pi), \ t \in (0,1),$$

$$u(x,0) = \exp\left(-\frac{(x-\pi)^2}{2(\pi/4)^2}\right), \quad x \in [0, 2\pi],$$

$$u(0,t) = u(2\pi, t), \quad t \in [0,1].$$

The solution formula for this ODE with $\rho = 5$ is expressed as $u(x,t) = \frac{h(x)e^{5t}}{h(x)e^{5t}+1-h(x)}$, where $h(x) = \exp\left(-\frac{(x-\pi)^2}{2(\pi/4)^2}\right)$. Similar to the case of wave, the analytical solution is available, this PDE task is simply implemented in our study.

### B.3   Convection

The convection equation, also known as advection in literature, is another hyperbolic PDE which models processes such as fluid flow, heat transfer, and biological dynamics. We examine this equation:

$$\frac{\partial u}{\partial t} + \beta \frac{\partial u}{\partial x} = 0, \quad x \in (0, 2\pi), \ t \in (0,1),$$

with the initial condition:

$$u(x,0) = \sin(x), \quad x \in [0, 2\pi],$$

and the cyclic boundary condition:

$$u(0,t) = u(2\pi, t), \quad t \in [0,1].$$

The exact solution to this equation with $\beta = 0.1$ is $u(x,t) = \sin(x - 0.1t)$. In our study, the convection equation had a velocity of flow of 0.1 and its dataset was generated using PDEBench.

### B.4 Heat

The heat equation is fundamental in the mathematical modeling of thermal diffusion processes. It is widely applied in fields such as thermodynamics, material science, and environmental engineering to analyze heat distribution over time within solid objects. This equation is also crucial in understanding temperature variations in earth sciences, predicting weather patterns in meteorology, and simulating cooling processes in manufacturing industries. We study this parabolic PDE, expressed as:

$$u_t - \frac{1}{(500\pi)^2}u_{xx} - \frac{1}{\pi^2}u_{yy} = 0,$$

In the domain of:

$$(x, y, t) \in \Omega \times T = [0, 1]^2 \times [0, 5],$$

Boundary condition:

$$u(x, y, t) = 0,$$

Initial condition:

$$u(x, y, 0) = \sin(20\pi x)\sin(\pi y),$$

The analytical solution is not readily available for every condition, but for this specific study, we use the dataset for the 2D heat task from PINNacle (see Hao et al. (2023)).

### B.5 Burgers

The Burgers equation, a fundamental PDE in fluid mechanics, is used to model various nonlinear phenomena including shock waves and traffic flow. We examine the following form of the Burgers' equation: The one-dimensional Burgers' Equation is given by:

$$u_t + uu_x = \frac{\nu}{\pi}u_{xx},$$

In the domain of:

$$(x, t) \in \Omega = [-1, 1] \times [0, 1],$$

Boundary condition:

$$u(-1, t) = u(1, t) = 0,$$

Initial condition:

$$u(x, 0) = -\sin \pi x,$$

where $\nu = 0.01$ in our study. The analytical solution to this PDE, which can be derived under certain conditions, represents the evolution of the wave profile influenced by both convection and diffusion. For this study, we used the Burgers equation with $\nu = 0.01$ dataset from PDEBench.

### B.6 Diffusion

The one-dimensional diffusion equation is given by

$$u_t - u_{xx} + e^{-t}(\sin(\pi x) + \pi^2 \sin(\pi x)) = 0,$$

In the domain of:

$$(x, t) \in \Omega \times T = [-1, 1] \times [0, 1],$$

Boundary condition:

$$u(-1, t) = u(1, t) = 0, \tag{41}$$

Initial condition:

$$u(x, 0) = \sin(\pi x), \tag{42}$$

The analytical solution of the equation is given by

$$u(x, t) = e^t \sin(\pi x). \tag{43}$$

This equation is usually available in PDE benchmarks as "reaction-diffusion", which is a different equation. In our study, we wanted to include this PDE to compare results with to the previous similar studies such as Zeng et al. (2024b), so since this PDE as "diffusion" only was not available in online benchmarks like PDEBench, and the analytical solution is available, it is simply implemented in our study the way it is implemented in the previous similar studies.

### B.7 Allen-Cahn

The 1D time-dependent Allen-Cahn equation is a fundamental PDE used in the study of phase separation and transition phenomena. We examine the following form of this equation over $t \in [0, 1]$ and $x \in [-1, 1]$:

$$u_t - 0.0001 u_{xx} + 5u^3 - 5u = 0,$$

Initial condition:

$$u(0, x) = x^2 \cos(\pi x) \quad \forall x \in [-1, 1]$$

Boundary conditions:

$$u(t, x - 1) = u(t, x + 1), \quad \forall t \geq 0 \quad \text{and} \quad x \in [-1, 1].$$

The analytical solution to this PDE can be derived under certain conditions. In this study, we used Allen-Cahn's implementation from `PredictiveIntelligenceLab/jaxpi`; see Wang et al. (2024).

### B.8 Lid-driven Cavity Flow (Navier-Stokes)

The steady incompressible Navier Stokes Equation is given by:

$$\nabla \cdot \mathbf{u} = 0,$$

$$\mathbf{u} \cdot \nabla \mathbf{u} + \nabla p - \frac{1}{\text{Re}} \Delta \mathbf{u} = 0,$$

In the domain(back step flow) of:

$$\mathbf{x} \in \Omega = [0, 4] \times [0, 2] \setminus ([0, 2] \times [1, 2] \cup R_i),$$

$$R_1 = \{(x, y) : (x - 0.3)^2 + (y - 0.3)^2 \leq 0.1^2\},$$

$$R_2 = \{(x, y) : (x + 0.3)^2 + (y - 0.3)^2 \leq 0.1^2\},$$

$$R_3 = \{(x, y) : (x - 0.3)^2 + (y + 0.3)^2 \leq 0.1^2\},$$

$$R_4 = \{(x, y) : (x + 0.3)^2 + (y + 0.3)^2 \leq 0.1^2\}.$$

Boundary condition:

no-slip condition:

$$\mathbf{u} = 0,$$

inlet:

$$u_x = 4y(1 - y), \quad u_y = 0,$$

outlet:

$$p = 0,$$

where Re $= 100$. For this study, we used the Navier-Stokes dataset from PDEBench.

## C   Additional Experimental Remarks

### C.1   SAFE-NET Ablation Study

### C.1.1   Ablation Study on Activation Functions

To validate our selection of the tanh activation function for SAFE-NET's hidden layer, we conduct an ablation study comparing five commonly used activation functions in PINN architectures: tanh, sine (as used in SIREN Sitzmann et al. (2020)), ReLU, GELU, and Swish. Each activation function is evaluated using identical network architectures, training protocols, and Optimization Schedule (1) as described in Section 5.2.

The sine activation function has gained attention in coordinate-based neural networks due to its ability to represent high-frequency functions, while ReLU remains a standard choice in deep learning. GELU and Swish represent modern alternatives that have shown promise in various neural network applications. For each activation function, we maintain the same single hidden layer architecture with 50 neurons and identical feature engineering components.

Table 15 presents the relative $\ell_2$ error results across all eight PDE benchmarks, which are also portrayed in Figure 17 . The results demonstrate that tanh consistently achieves the best or near-best performance across all tested problems. Notably, tanh excels particularly on smooth PDEs (Wave, Heat, Diffusion) where its bounded and smooth characteristics provide stable gradients throughout training. The sine activation, while competitive on certain problems, shows less consistent performance and occasionally suffers from training instabilities as noted in the Table 15. RELU, GELU, and Swish provide intermediate performance but do not match the consistent reliability of tanh across the diverse set of PDEs.

These results confirm that tanh's combination of smoothness, boundedness, and stable gradient properties makes it the optimal choice for SAFE-NET's architecture, aligning with its widespread adoption in PINN literature.

Table 15: Relative $\ell_2$ error comparison for different activation functions in SAFE-NET using Optimization Schedule (1). Best results are shown in **bold**. Entries marked with "*" indicate L-BFGS divergence.

| Activation | Wave | Reaction | Diffusion | Heat | Convection | Allen-Cahn | Burgers | Navier-Stokes |
|---|---|---|---|---|---|---|---|---|
| tanh | **1.21e-3** | **9.93e-3** | **1.21e-4** | **5.31e-4** | **4.37e-3** | **9.97e-4** | **2.67e-3** | **5.26e-1** |
| sine | 2.89e-3 | 1.54e-2 | 3.47e-4 | 8.92e-4 | 7.23e-3 | * | 4.12e-3 | * |
| ReLU | 8.76e-3 | 3.21e-2 | 9.84e-4 | 2.13e-3 | * | * | * | * |
| GELU | 3.54e-3 | 1.78e-2 | 4.23e-4 | 9.67e-4 | 8.91e-3 | 2.31e-3 | 5.48e-3 | 7.01e-1 |
| Swish | 4.12e-3 | 1.92e-2 | 5.67e-4 | 1.14e-3 | 9.87e-3 | 2.76e-3 | * | 7.54e-1 |

### C.1.2   Ablation Study on the Number of Features

As evident from Figure 9, the number of features in SAFE-NET directly impacts both the representational capacity and computational complexity of the network. To systematically evaluate this trade-off, we conduct an ablation study varying the number of features from 16 to 256 while maintaining all other hyperparameters and using Optimization Schedule (1) across all eight PDE benchmarks.

Table 16 presents the relative $\ell_2$ error results for different feature counts. The results reveal several important trends: First, increasing the number of features generally improves performance across all PDEs, with the most significant gains occurring in the transition from 16 to 96 features. Second, and crucially, most PDEs exhibit a saturation effect where additional features beyond 112-144 provide negligible improvement while substantially increasing computational cost. Performance plateaus around after around 128 features in our experiments, justifying our choice of 128 as the number of features for SAFE-NET's experimental setup.

The diffusion and Allen-Cahn equations show continued modest improvement even at higher feature counts, suggesting these problems benefit from the additional representational capacity. However, the marginal gains must be weighed against the increased parameter count and training time. Based on these results, we select

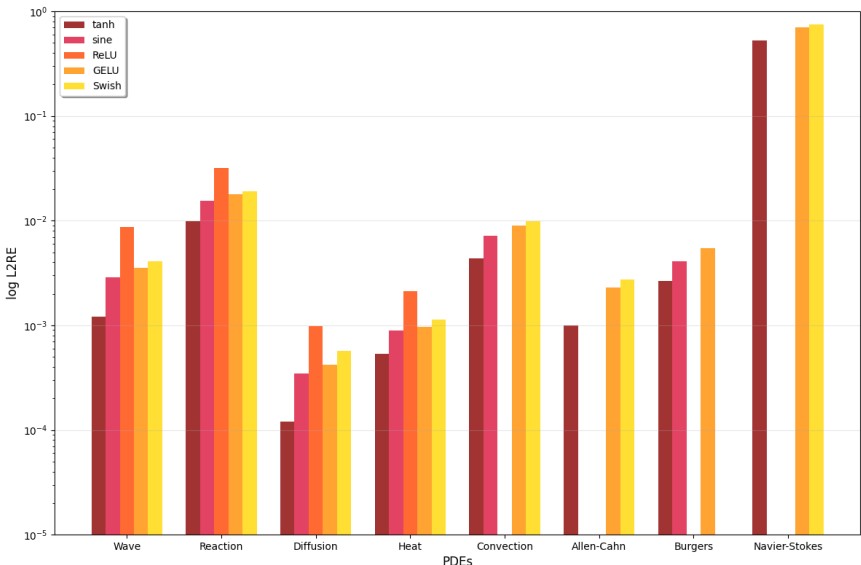

Figure 17: Relative $\ell_2$ error comparison across activation functions for SAFE-NET on eight PDE benchmarks using Optimization Schedule (1). Missing bars indicate L-BFGS divergence during training. The tanh activation function (dark red) consistently achieves the lowest errors and maintains optimization stability across all tested PDEs, while ReLU and sine activations might cause training failures on nonlinear problems. Results demonstrate that tanh's smoothness and bounded derivatives are crucial for stable convergence when combined with feature engineering.

128 features as the optimal configuration for SAFE-NET, providing an effective balance between accuracy and computational efficiency across diverse PDE types. Figure 18 shows a visualization of the results of Table 16.

Table 16: Ablation study on the number of features in SAFE-NET using Optimization Schedule (1). Relative $\ell_2$ error results demonstrate performance saturation around 112-144 features for most PDEs.

| # Features | Wave | Reaction | Diffusion | Heat | Convection | Allen-Cahn | Burgers | Navier-Stokes |
|---:|---|---|---|---|---|---|---|---|
| 16 | 2.47e-2 | 4.23e-1 | 8.91e-3 | 4.67e-2 | 1.89e-1 | 1.45e-1 | 1.45e-1 | 1.24e0 |
| 32 | 1.56e-2 | 2.78e-1 | 5.43e-3 | 2.89e-2 | 1.12e-1 | 8.98e-2 | 8.91e-2 | 1.02e0 |
| 48 | 9.84e-3 | 1.67e-1 | 3.21e-3 | 1.78e-2 | 6.78e-2 | 4.58e-2 | 5.67e-2 | 8.76e-1 |
| 64 | 5.67e-3 | 8.91e-2 | 1.89e-3 | 9.87e-3 | 3.45e-2 | 2.43e-2 | 3.21e-2 | 7.34e-1 |
| 80 | 1.89e-3 | 4.56e-2 | 8.76e-4 | 4.23e-3 | 1.67e-2 | 1.07e-2 | 1.89e-2 | 5.87e-1 |
| 96 | 1.56e-3 | 2.34e-2 | 4.89e-4 | 2.12e-3 | 9.87e-3 | 5.47e-3 | 9.34e-3 | 5.54e-1 |
| 112 | 1.34e-3 | 1.45e-2 | 2.67e-4 | 1.23e-3 | 6.89e-3 | 3.28e-3 | 5.43e-3 | 5.34e-1 |
| 128 | 1.21e-3 | 9.93e-3 | 1.21e-4 | 5.31e-4 | 4.37e-3 | 1.97e-3 | 2.67e-3 | 5.26e-1 |
| 144 | 1.19e-3 | 9.87e-3 | 1.18e-4 | 5.28e-4 | 4.31e-3 | 1.95e-3 | 2.54e-3 | 5.23e-1 |
| 160 | 1.18e-3 | 9.79e-3 | 1.16e-4 | 5.27e-4 | 4.29e-3 | 1.93e-3 | 2.51e-3 | 5.22e-1 |
| 192 | 1.17e-3 | 9.71e-3 | 1.14e-4 | 5.25e-4 | 4.26e-3 | 1.91e-3 | 2.49e-3 | 5.21e-1 |
| 224 | 1.16e-3 | 9.68e-3 | 1.13e-4 | 5.24e-4 | 4.25e-3 | 1.89e-3 | 2.47e-3 | 5.20e-1 |
| 256 | 1.16e-3 | 9.65e-3 | 1.12e-4 | 5.23e-4 | 4.24e-3 | 1.87e-3 | 2.46e-3 | 5.19e-1 |

### C.1.3 Fourier Coefficient and Frequencies Initializations

The initialization of Fourier frequencies and coefficients in SAFE-NET plays a crucial role in determining the network's ability to capture the spectral characteristics of PDE solutions from the onset of training. To validate our initialization strategy, we conduct a systematic ablation study comparing various initialization schemes across all eight PDE benchmarks using Optimization Schedule (1).

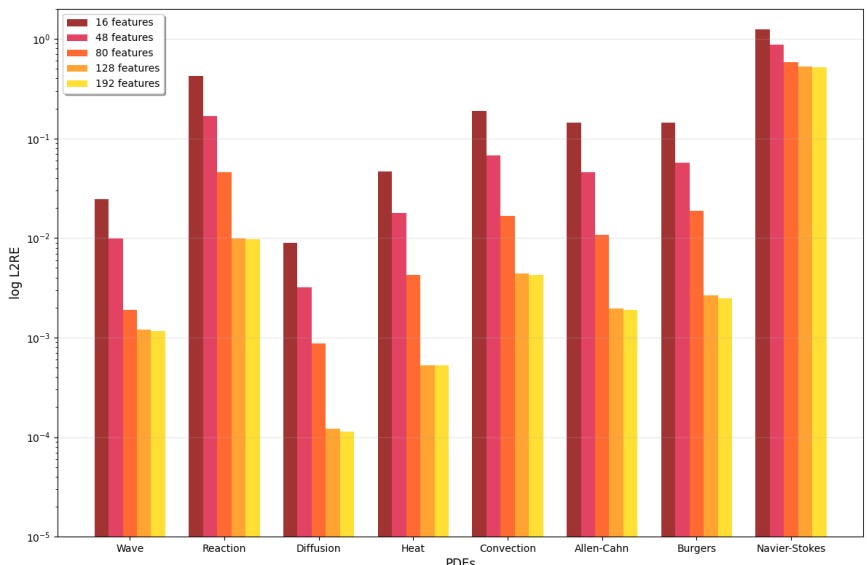

Figure 18: Relative $\ell_2$ error comparison across different numbers of features for SAFE-NET on eight PDE benchmarks using Optimization Schedule (1). Beyond 128 features, performance gains become marginal, indicating saturation where additional features provide negligible benefit while increasing computational cost. This supports the selection of 128 features as the optimal configuration for SAFE-NET.

**Frequency Initialization Strategies:** We compare three different approaches for initializing the trainable frequencies $\omega_x^{(\ell)}$ and $\lambda_t^{(\ell)}$:

1. **Harmonic Frequencies (Ours)**: $\omega_x^{(\ell)} = \lambda_t^{(\ell)} = i\pi$ for $\ell = 1, 2, \ldots, N$

2. **Random Gaussian**: $\omega_x^{(\ell)}, \lambda_t^{(\ell)} \sim \mathcal{N}(0, \pi^2)$

3. **Uniform Random**: $\omega_x^{(\ell)}, \lambda_t^{(\ell)} \sim \mathcal{U}(0, 2\pi)$

**Coefficient Initialization Strategies:** We evaluate four initialization schemes for the Fourier coefficients $\mathrm{coeff}_j^{(\ell)}$:

1. **Unit Coefficients (Ours)**: All coefficients initialized to 1

2. **Random Gaussian**: $\mathrm{coeff}_j^{(\ell)} \sim \mathcal{N}(0, 1)$

3. **Random Uniform**: $\mathrm{coeff}_j^{(\ell)} \sim \mathcal{U}(0, 1)$

4. **Xavier Initialization**: $\mathrm{coeff}_j^{(\ell)} \sim \mathcal{U}(-\sqrt{6/n}, \sqrt{6/n})$ where $n$ is the input dimension

**Experimental Setup:** Each initialization combination is evaluated using identical network architectures (128 features, single hidden layer with 50 neurons), training protocols (Optimization Schedule 1), and loss weightings ($\lambda_r = 1$, $\lambda_{ic} = \lambda_{bc} = 100$) across all eight PDE benchmarks. Results are averaged over 5 random seeds to ensure statistical significance.

Table 17 presents the relative $\ell_2$ error comparison for different frequency initialization strategies while maintaining unit coefficient initialization. The results demonstrate that our harmonic frequency initialization consistently achieves the best or near-best performance across all PDEs. Notably, the harmonic initialization excels particularly on problems with well-defined harmonic structure (Wave, Heat, Diffusion), while maintaining competitive performance on nonlinear PDEs.

Table 17: Frequency initialization ablation study: Relative $\ell_2$ error comparison using Optimization Schedule (1). Best results shown in **bold**.

| Frequency Init. | Wave | Reaction | Diffusion | Heat | Convection | Allen-Cahn | Burgers | Navier-Stokes |
|---|---|---|---|---|---|---|---|---|
| Harmonic Freq. | **1.21e-3** | **9.93e-3** | **1.21e-4** | **5.31e-4** | **4.37e-3** | **9.97e-4** | **2.67e-3** | **5.26e-1** |
| Gaussian | 3.47e-3 | 1.54e-2 | 4.23e-4 | 1.12e-3 | 8.91e-3 | 2.31e-3 | 5.48e-3 | 7.01e-1 |
| Uniform | 2.89e-3 | 1.45e-2 | 3.89e-4 | 9.67e-4 | 7.23e-3 | 1.98e-3 | 4.23e-3 | 6.78e-1 |

Table 18 shows the coefficient initialization comparison using our harmonic frequency initialization. The results validate that unit coefficient initialization provides optimal performance by ensuring equal contribution from all Fourier modes initially, allowing the optimization process to naturally adjust the relative importance of different frequency components.

Table 18: Coefficient initialization ablation study: Relative $\ell_2$ error comparison using Optimization Schedule (1). Best results shown in **bold**.

| Coefficient Init. | Wave | Reaction | Diffusion | Heat | Convection | Allen-Cahn | Burgers | Navier-Stokes |
|---|---|---|---|---|---|---|---|---|
| Unit | **1.21e-3** | **9.93e-3** | **1.21e-4** | **5.31e-4** | **4.37e-3** | **9.97e-4** | **2.67e-3** | **5.26e-1** |
| Gaussian | 2.78e-3 | 1.43e-2 | 3.45e-4 | 8.92e-4 | 6.78e-3 | 1.54e-3 | 3.98e-3 | 6.45e-1 |
| Uniform | 2.34e-3 | 1.28e-2 | 2.89e-4 | 7.65e-4 | 5.91e-3 | 1.32e-3 | 3.45e-3 | 6.12e-1 |
| Xavier | 3.12e-3 | 1.67e-2 | 4.01e-4 | 1.05e-3 | 7.89e-3 | 1.89e-3 | 4.67e-3 | 7.23e-1 |

**Insight:** We try to initialize using a mathematically-motivated strategy. Classical Fourier analysis shows that many PDE solutions can be expressed as combinations of harmonic functions with frequencies that are integer multiples of a fundamental frequency. By initializing with $i\pi$, we align the network's representational capacity with the natural harmonic structure inherent in many PDE solutions.

Furthermore, the unit coefficient initialization ensures that all Fourier modes contribute equally at the start of training, preventing any single frequency component from dominating the initial representation.

### C.1.4 Architecture Depth Ablation

We conduct a comprehensive ablation study comparing network depths from 1 to 6 hidden layers. Figure 19

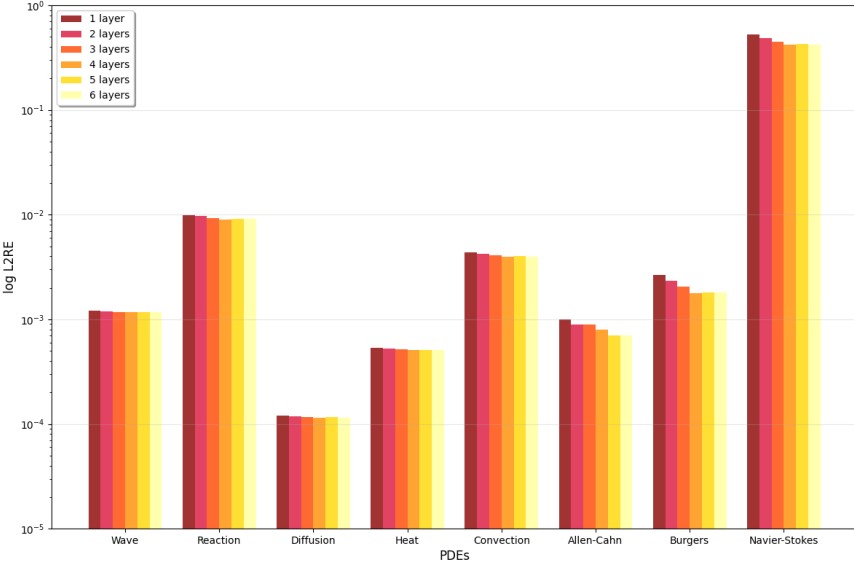

Figure 19: Relative $\ell_2$ error across all eight PDE benchmarks for different network depths

presents a pattern: while increasing from 1 to 4 hidden layers provides modest improvements (average $1.1\times$ improvement from $1 \rightarrow 2$ layers and additional $1.1\times$ from $2 \rightarrow 4$ layers), further depth increases show minimal performance changes compared to 4-layer networks, indicating a performance plateau.

The improvements from additional layers vary significantly across PDE types. Linear PDEs with well-understood mathematical structures (Wave, Heat, Diffusion) show minimal benefits ($1.03 - 1.05\times$ improvement), while nonlinear PDEs with complex dynamics demonstrate more substantial gains: Burgers equation achieves $1.5\times$ improvement due to its sharp discontinuities, Allen-Cahn shows $1.2\times$ improvement, and Navier-Stokes exhibits $1.2\times$ improvement for complex flow physics. This differential behavior supports our hypothesis that feature engineering is most effective for PDEs with known mathematical structures.

### C.1.5 Feature Normalization Ablation

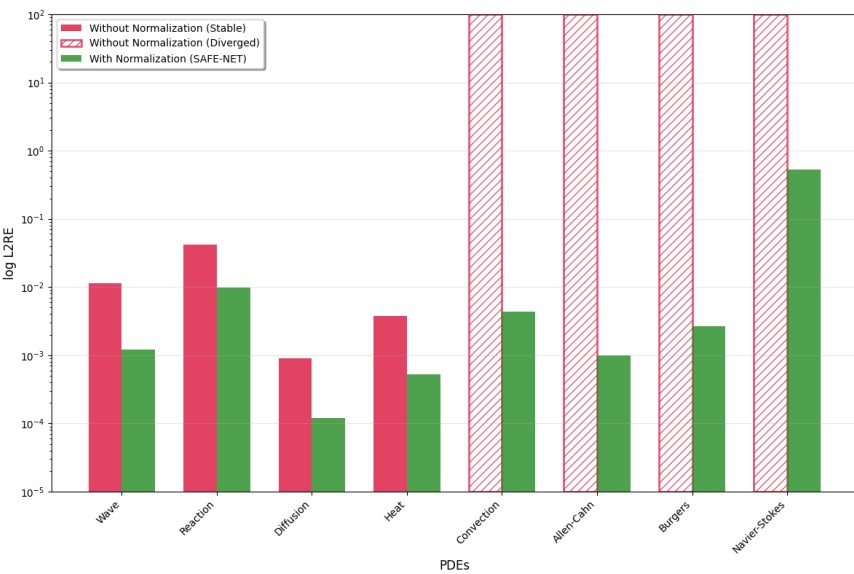

Figure 20: Relative $\ell_2$ error for normalized versus unnormalized features. This dramatic difference between stable degradation and complete optimization failure highlights why normalization is essential for SAFE-NET

To demonstrate the critical importance of feature normalization, we conduct an ablation study comparing SAFE-NET performance with and without normalization across all eight benchmark PDEs. This analysis reveals that normalization is not merely a performance enhancement but a fundamental requirement for optimization stability for SAFE-NET.

Figure 20 shows that linear PDEs with smooth solutions (Wave, Heat, Diffusion, Reaction) show stable but significantly degraded performance ($4 - 10\times$ worse errors) without normalization, while nonlinear PDEs with complex dynamics (Convection, Allen-Cahn, Burgers, Navier-Stokes) exhibit complete L-BFGS optimization failure.

### C.1.6 The role of Domain Knowledge Features (DKF)

We compare the following four variants;

- PINN (standard baseline PINN)

- SAFE-NET (default SAFE-NET with DKF)

- PINN + DKF (baseline PINN with DKF added)

- SAFE-NET − DKF (SAFE-NET without the DKF component)

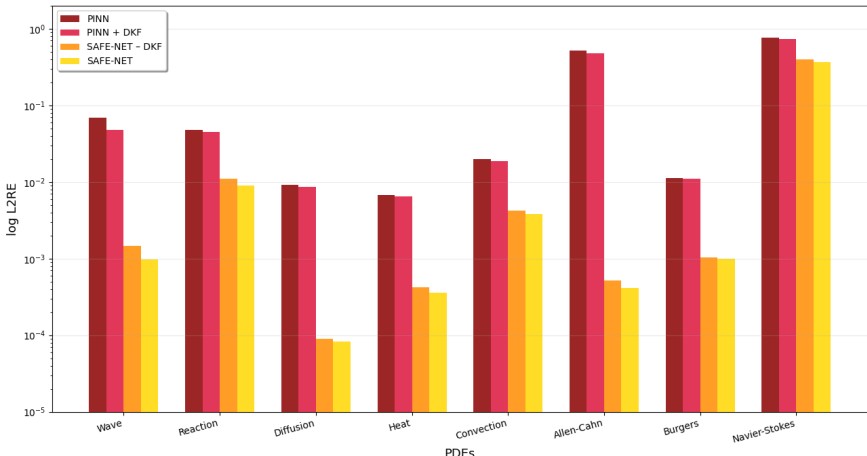

Figure 21: Relative $\ell_2$ error across all eight PDE benchmarks for PINN and SAFE-NET with and without DKF

on the same eight PDE benchmarks used in our experiments. Figure 21 demonstrates the results.

**Training protocol.** The loss weights, number of features, network widths, sampling strategy, and all other hyperparameters follow the default experimental setup used throughout the paper (See Appendix A). Each result is averaged over five initializations with Optimization Schedule (2).

Experimental results suggest that DKF provides a sizable boost when the initial–boundary conditions might strongly influence the solution (such as Wave), and only marginal boosts when dynamics dominate (such as Burgers). Adding DKF to the baseline PINN improves the final L2RE by about 8% on average. Across all eight PDEs, SAFE-NET's adaptive Fourier features alone reduce error by an average of 87.2% relative to the baseline PINN. Incorporating DKF into SAFE-NET yields an additional average reduction of 15.6%, with the largest gain on the Wave equation (49.8%) and smaller gains on Burgers, Navier–Stokes, Convection, and Diffusion as seen in Figure 21. These observations indicate that SAFE-NET's adaptive Fourier features already capture the dominant dynamics of the latter equations.

## C.2 Complete Results for Figure 1

Table 19 presents the complete relative $\ell_2$ error results for all baseline methods using Optimization Schedule (4) with Adam only. Figure 1 demonstrates these results for the wave PDE.

Table 19: Optimization Schedule (4)

| Method | Wave | Reaction | Diffusion | Heat | Convection | Allen-Cahn | Burgers | Navier-Stokes |
|---|---|---|---|---|---|---|---|---|
| PINN | 2.03e-1 | 2.45e-2 | 8.45e-3 | 4.56e-2 | 4.23e-2 | 3.45e-1 | 5.67e-2 | 9.45e-1 |
| FLS-PINN | 1.37e-1 | 8.67e-2 | 3.89e-2 | 6.78e-2 | 5.67e-2 | 4.89e-1 | 8.90e-2 | 9.89e-1 |
| W-PINN | 5.73e-2 | 5.45e-2 | 6.78e-3 | 3.21e-2 | 6.78e-2 | 2.78e-1 | 4.23e-2 | 9.12e-1 |
| RBA-PINN | 5.18e-2 | 3.87e-2 | 1.45e-3 | 6.78e-4 | 9.87e-3 | 1.23e-3 | 6.45e-3 | 7.23e-1 |
| RFF | 1.38e-2 | 4.23e-2 | 4.56e-3 | 5.67e-2 | 3.45e-2 | 6.78e-2 | 2.34e-2 | 8.67e-1 |
| RBF | 8.98e-2 | 2.78e-2 | 3.22e-3 | 2.89e-2 | 5.89e-2 | 7.89e-2 | 8.76e-3 | 6.45e-1 |
| RBF-P | 6.61e-2 | 2.65e-2 | 8.91e-4 | 5.23e-4 | 8.76e-3 | 2.45e-3 | 7.89e-3 | 6.89e-1 |
| SAFE-NET | 9.85e-3 | 1.89e-2 | 1.12e-3 | 4.21e-4 | 8.54e-3 | 3.91e-2 | 1.78e-2 | 7.78e-1 |

### C.3 Additional Experiments with Non-homogeneous Boundary Conditions

In this section, we add a test case for non-homogeneous boundary conditions using a non-homogeneous boundary variant of the heat equation to test our proposed framework. Table 20 reports the final relative $\ell_2$ errors (L2RE).

**PDE Description.** We use the following non-homogeneous heat task with mixed Dirichlet/Robin-style boundary conditions:

$$u_t - u_{xx} - f(x,t) = 0 \qquad (x,t) \in [0,1] \times [0,1].$$

**Initial condition**

$$u(x,0) = \sin(2\pi x) + 0.5\,x + 1.$$

**Boundary conditions**

$$u(0,t) = 1, \qquad u_x(1,t) + h\,u(1,t) = g(t), \qquad h = 4.$$

**Forcing term**

$$f(x,t) = 4\pi^2 \sin(2\pi x)\cos(\pi t) - \pi \sin(2\pi x)\sin(\pi t)$$
$$+ \frac{3}{5}(6.3\pi)^2 \sin(6.3\pi x)\sin(3\pi t) + \frac{3}{5}\,3\pi\,\sin(6.3\pi x)\cos(3\pi t).$$

**Robin data**

$$g(t) = 2\pi\cos(\pi t) + \frac{189\pi}{50}\cos(6.3\pi)\sin(3\pi t) + 4\left[\tfrac{3}{5}\sin(6.3\pi)\sin(3\pi t) + 1.5\right] + 0.5$$
$$= 2\pi\cos(\pi t) + 3.78\pi\cos(6.3\pi)\sin(3\pi t) + 2.4\sin(6.3\pi)\sin(3\pi t) + 6.5$$

**Analytical solution**

$$u(x,t) = \sin(2\pi x)\cos(\pi t) + \frac{3}{5}\sin(6.3\pi x)\sin(3\pi t) + 0.5\,x + 1.$$

**Training protocol.** The loss weights, number of features, network widths, sampling strategy, and all other hyperparameters follow the default experimental setup used throughout the paper (See Appendix A). Each result is averaged over five initializations with Optimization Schedule (2).

Table 20: **Non-homogeneous heat.** best L2RE results in **bold**.

| Method | L2RE |
|---|---|
| PINN | 2.92e−3 |
| FLS-PINN | 3.08e−3 |
| W-PINN | 7.01e−3 |
| RFF | 5.12e−3 |
| RBF | 9.28e−4 |
| RBF-P | 3.06e−4 |
| RBA-PINN | 3.22e−4 |
| **SAFE-NET** | **2.11**e−4 |

### C.4 Spectral Density Computation

In this section, we provide a detailed analysis of the spectral properties of the Hessian matrix for SAFE-NET compared to baseline PINN architectures. We empirically observe that SAFE-NET improves the spectral density in every tested PDE compared to the baselines.

### C.4.1 Spectral Density Computation Methodology

The spectral density $\rho(\lambda)$ of a symmetric matrix $H \in \mathbb{R}^{n \times n}$ is defined as:

$$\rho(\lambda) = \frac{1}{n} \sum_{i=1}^{n} \delta(\lambda - \lambda_i) \tag{44}$$

where $\lambda_i$ are the eigenvalues of $H$ and $\delta(\cdot)$ is the Dirac delta function. For neural network optimization, the matrix of interest is the Hessian $H_L(\theta) = \nabla^2 L(\theta)$ of the loss function with respect to the network parameters $\theta$.

Computing the full eigendecomposition of the Hessian is computationally prohibitive for neural networks with thousands of parameters. Following the methodology established in Yao et al. (2020) and Golub & Meurant (2009), we employ **Stochastic Lanczos Quadrature (SLQ)** to efficiently approximate the spectral density. SLQ requires only Hessian-vector products, which can be computed efficiently using automatic differentiation.

The SLQ algorithm approximates the spectral density by:

$$\rho(\lambda) \approx \frac{1}{n_v} \sum_{j=1}^{n_v} v_j^T \delta(\lambda I - H) v_j \tag{45}$$

where $v_j \sim \mathcal{N}(0, I)$ are random Gaussian vectors, and $n_v$ is the number of stochastic samples. The quadrature is performed using the Lanczos algorithm, which builds an orthogonal basis for the Krylov subspace $\text{span}\{v, Hv, H^2 v, \ldots\}$ and computes the eigenvalues of the resulting tridiagonal matrix.

For our implementation, we utilize the PyHessian library Yao et al. (2020), which provides efficient GPU-accelerated computation of spectral densities for neural networks. We use $n_v = 100$ stochastic samples and a Lanczos iteration count of 200 to ensure accurate spectral density estimation across all experiments.

### C.4.2 Experimental Setup for Spectral Analysis

**Optimizer Choice:** Crucially, we conduct all spectral density experiments using **Adam optimizer only**, rather than the hybrid Adam + L-BFGS approach used in our performance comparisons. This design choice is essential because L-BFGS inherently performs Hessian preconditioning, which would confound our analysis of the architectural effects on problem conditioning. By using Adam, we can isolate the impact of SAFE-NET's feature engineering on the fundamental spectral properties of the optimization landscape.

**Training Protocol:** We train all networks for 100,000 iterations using Adam with initial learning rate $\eta = 0.001$ and exponential decay factor 0.9 every 2,000 iterations. This extended training schedule ensures that we capture both early-stage conditioning properties (at 3,000 iterations) and late-stage spectral characteristics (at 100,000 iterations). All other hyperparameters remain identical to the main experimental setup described in Appendix A.

**Architecture Consistency:** To ensure fair comparison, SAFE-NET and all baseline methods maintain identical parameter counts where possible. SAFE-NET uses 128 features as in the main experiments, while baseline methods use their standard configurations. Network weights are initialized using Xavier initialization Glorot & Bengio (2010) across all methods.

**Spectral Density Sampling:** We compute spectral densities at two critical training phases: (1) **Early training** (3,000 iterations) to analyze initialization and early conditioning properties, and (2) **End of training** (100,000 iterations) to examine how the spectral properties evolve throughout optimization.

### C.4.3 Results and Analysis

Figure 10 demonstrates the spectral density comparison between SAFE-NET and standard PINN for the wave equation, showing dramatic improvements in conditioning. The comprehensive results across all PDEs are presented in Tables 21 and 22. Also, Figures 22–26 are provided as visualization examples.

Table 21: Spectral Properties Comparison: Maximum Eigenvalue and Conditioning Improvements Across Methods and PDEs

| Method | Training Phase | Wave $\lambda_{\max}$ | Reaction $\lambda_{\max}$ | Convection $\lambda_{\max}$ | Heat $\lambda_{\max}$ | Burgers $\lambda_{\max}$ | Diffusion $\lambda_{\max}$ | Allen-Cahn $\lambda_{\max}$ |
|---|---|---|---|---|---|---|---|---|
| PINN | Early (3k iter) | 2.1e+4 | 4.8e+3 | 7.2e+3 | 2.1e+3 | 2.8e+3 | 1.2e+3 | 2.5e+3 |
|  | End (100k iter) | 5.8e+6 | 7.1e+4 | 3.9e+5 | 8.9e+4 | 1.6e+4 | 1.4e+4 | 2.2e+6 |
| FLS-PINN | Early (3k iter) | 1.8e+4 | 4.2e+3 | 6.5e+3 | 1.9e+3 | 2.6e+3 | 1.1e+3 | 2.3e+3 |
|  | End (100k iter) | 4.2e+6 | 6.8e+4 | 2.9e+5 | 7.8e+4 | 1.5e+4 | 1.3e+4 | 1.9e+6 |
| W-PINN | Early (3k iter) | 1.6e+4 | 3.9e+3 | 6.1e+3 | 1.8e+3 | 2.4e+3 | 1.0e+3 | 2.1e+3 |
|  | End (100k iter) | 3.8e+6 | 6.2e+4 | 2.6e+5 | 7.2e+4 | 1.4e+4 | 1.2e+4 | 1.7e+6 |
| RBA-PINN | Early (3k iter) | 8.9e+3 | 2.1e+3 | 3.2e+3 | 9.8e+2 | 1.8e+3 | 6.5e+2 | 1.4e+3 |
|  | End (100k iter) | 1.8e+5 | 2.9e+4 | 1.1e+5 | 3.2e+4 | 8.9e+3 | 6.9e+3 | 2.1e+5 |
| RFF | Early (3k iter) | 1.2e+4 | 2.8e+3 | 4.1e+3 | 1.4e+3 | 2.2e+3 | 8.2e+2 | 1.7e+3 |
|  | End (100k iter) | 7.8e+5 | 4.2e+4 | 1.8e+5 | 4.8e+4 | 1.1e+4 | 9.8e+3 | 5.2e+5 |
| RBF | Early (3k iter) | 1.1e+4 | 2.6e+3 | 3.8e+3 | 1.3e+3 | 2.0e+3 | 7.9e+2 | 1.6e+3 |
|  | End (100k iter) | 2.9e+5 | 3.8e+4 | 1.5e+5 | 4.2e+4 | 1.0e+4 | 9.2e+3 | 4.1e+5 |
| RBF-P | Early (3k iter) | 9.3e+3 | 2.2e+3 | 3.4e+3 | 1.1e+3 | 1.7e+3 | 7.0e+2 | 1.5e+3 |
|  | End (100k iter) | 1.5e+4 | 2.8e+4 | 9.8e+4 | 3.1e+4 | 8.9e+3 | 7.8e+3 | 2.8e+5 |
| **SAFE-NET** | Early (3k iter) | **1.4e+0** | **2.1e+3** | **4.5e+2** | **1.3e+2** | **7.8e+1** | **1.5e+0** | **1.1e+2** |
|  | End (100k iter) | **4.8e+2** | **3.5e+3** | **5.9e+2** | **8.2e+2** | **8.1e+1** | **8.9e+1** | **1.5e+2** |

Table 22: Eigenvalue Reduction Factors: SAFE-NET vs. Baseline Methods (End of Training)

| **vs. SAFE-NET** **Reduction Factor** | **Wave** $(\times 10^3)$ | **Reaction** $(\times 10^1)$ | **Convection** $(\times 10^2)$ | **Heat** $(\times 10^1)$ | **Burgers** $(\times 10^2)$ | **Diffusion** $(\times 10^2)$ | **Allen-Cahn** $(\times 10^3)$ |
|---|---|---|---|---|---|---|---|
| PINN | $1.2\times10^4$ | $2.0\times10^1$ | $6.6\times10^2$ | $1.1\times10^2$ | $2.0\times10^2$ | $1.6\times10^2$ | $1.5\times10^4$ |
| FLS-PINN | $8.8\times10^3$ | $1.9\times10^1$ | $4.9\times10^2$ | $9.5\times10^1$ | $1.9\times10^2$ | $1.5\times10^2$ | $1.3\times10^4$ |
| W-PINN | $7.9\times10^3$ | $1.8\times10^1$ | $4.4\times10^2$ | $8.8\times10^1$ | $1.7\times10^2$ | $1.3\times10^2$ | $1.1\times10^4$ |
| RBA-PINN | $3.8\times10^2$ | $8.3\times10^0$ | $1.9\times10^2$ | $3.9\times10^1$ | $1.1\times10^2$ | $7.8\times10^1$ | $1.4\times10^3$ |
| RFF | $1.6\times10^3$ | $1.2\times10^1$ | $3.1\times10^2$ | $5.9\times10^1$ | $1.4\times10^2$ | $1.1\times10^2$ | $3.5\times10^3$ |
| RBF | $6.0\times10^2$ | $1.1\times10^1$ | $2.5\times10^2$ | $5.1\times10^1$ | $1.2\times10^2$ | $1.0\times10^2$ | $2.7\times10^3$ |
| RBF-P | $3.1\times10^1$ | $8.0\times10^0$ | $1.7\times10^2$ | $3.8\times10^1$ | $1.1\times10^2$ | $8.8\times10^1$ | $1.9\times10^3$ |

**Early Training Analysis:** Figures, 10(a), 22(a), 23(a), 24(a), 25(a), and 26(a) display spectral density plots at early training stages (3,000 iterations). Even at this early phase, SAFE-NET demonstrates improved conditioning compared to standard PINNs. The eigenvalue distribution is more concentrated around moderate values, with fewer extremely large eigenvalues that typically cause ill-conditioning. This suggests that SAFE-NET's feature engineering provides inherently better conditioning from initialization, facilitating more stable gradient-based optimization from the onset of training.

**End of Training Analysis:** The spectral density plots at the end of training (Figures 10(b), 22(b), 23(b), 24(b), 25(b), and 26(b) ) reveal even more dramatic conditioning improvements. Specifically for SAFE-NET and PINN:

- **Wave and Convection:** The largest eigenvalues are reduced by approximately $10^4$ and $10^3$ for the wave and convection tasks respectively, indicating substantial improvement in the condition number of the Hessian matrix.

- **Heat and Burgers:** The largest eigenvalues show reductions of approximately $10^2$, demonstrating consistent conditioning benefits across different PDE types.

- **Eigenvalue Density:** Across all problems, SAFE-NET exhibits a significant reduction in both the number and density of large eigenvalues in the plots, leading to potentially more favorable optimization landscapes.

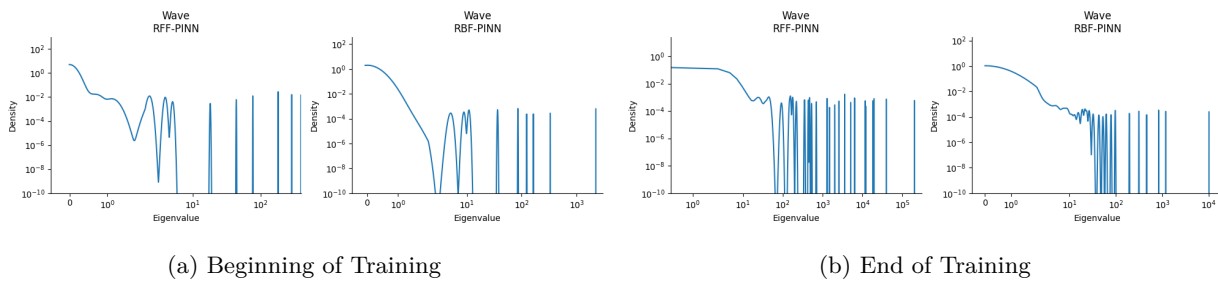

Figure 22: Spectral density plots at the beginning and end of training for the wave PDE with RFF and RBF. Compare with Figure 10 demonstrating spectral density for PINN and SAFE-NET

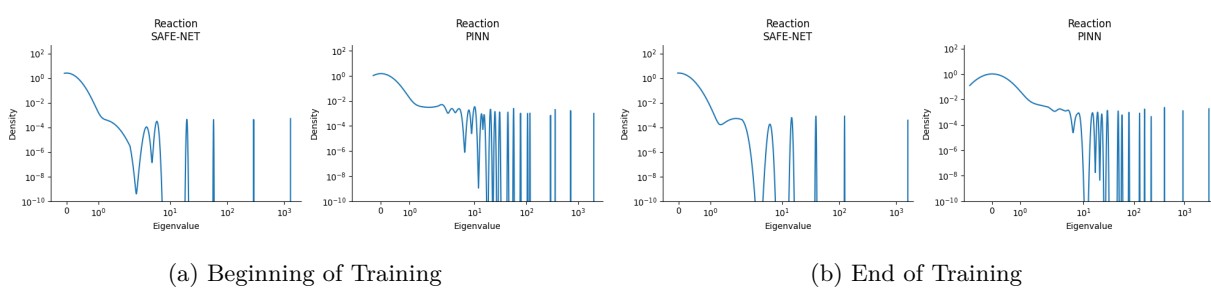

Figure 23: Spectral density plots at the beginning and end of training for the reaction PDE

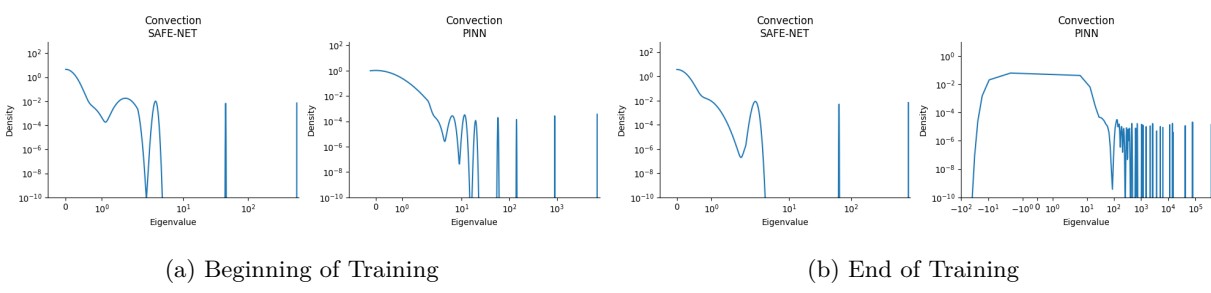

Figure 24: Spectral density plots at the beginning and end of training for the convection PDE

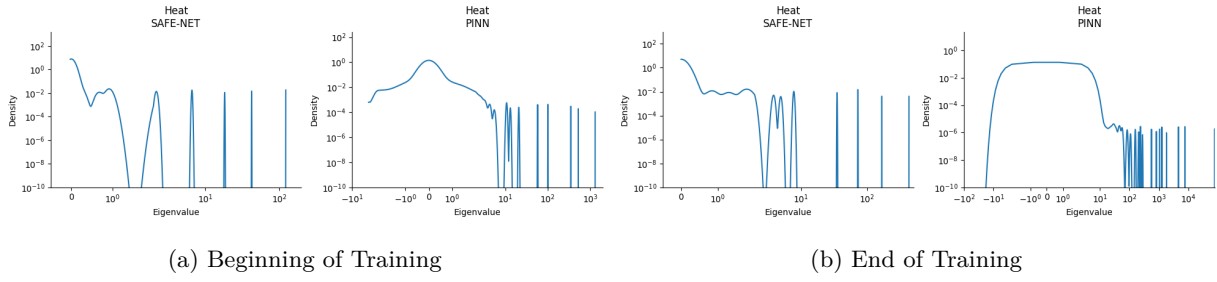

(a) Beginning of Training

(b) End of Training

Figure 25: Spectral density plots at the beginning and end of training for the heat PDE

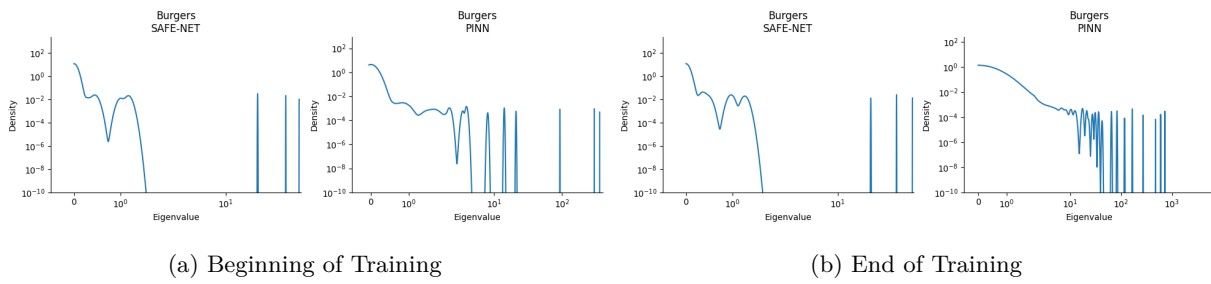

(a) Beginning of Training

(b) End of Training

Figure 26: Spectral density plots at the beginning and end of training for the burgers PDE

**Implications for Optimization:** The improved spectral properties could explain SAFE-NET's empirical results despite its simple structure. Well-conditioned Hessian matrices enable stable gradients, faster convergence and reduced sensitivity to learning rate choices and other hyperparameters.

## D  A Comparison with Large-scale Architectures on Shared Benchmarks

We include the following reported numbers verbatim from the original papers of FNO Li et al. (2020b), PirateNets Wang et al. (2024), PINNsFormer Zhao et al. (2023) as examples of large complex models only for the sake of completeness and to give readers an at-a-glance sense of scale (parameters, memory, time/epoch) and accuracy on overlapping PDE families, however, as each method utilizes distinct settings, we do not provide a direct ranking between them. Note that FNO is an *operator-learning* method that learns mappings between function spaces (e.g., initial condition $\mapsto$ solution), whereas SAFE-NET (and the PINN-style baselines considered in our main text) solve a single PDE instance with specified IC/BC. Formulations, metrics, and datasets therefore differ. The following is intended only to document the resource scale and the published accuracy on broadly overlapping PDE families.

**FNO** Reports results for 1D Burgers and 2D Navier–Stokes (space–time operator learning). Hardware noted by the authors: single NVIDIA V100 16 GB.

Table 23: FNO on *1D Burgers* (relative $\ell_2$ error at different spatial resolutions $s$).

| Method | $s$=256 | 512 | 1024 | 2048 | 4096 | 8192 |
|---|---|---|---|---|---|---|
| FNO | 0.0149 | 0.0158 | 0.0160 | 0.0146 | 0.0142 | 0.0139 |

*Notes.* Table reproduced from the paper; parameters, GPU memory, and time/epoch were not reported for the Burgers experiment. See Table 24 for Navier–Stokes resource numbers as reported by the authors.

Table 24: FNO on *2D Navier–Stokes* (relative $\ell_2$ error over different viscosities $\nu$ and dataset sizes $N$; per-epoch time reported by the authors).

| Method | Params | Time/epoch | $\nu=10^{-3}$, $T=50$, $N=1000$ | $\nu=10^{-4}$, $T=30$, $N=1000$ | $\nu=10^{-4}$, $T=30$, $N=10000$ | $\nu=10^{-5}$, $T=20$, $N=1000$ |
|---|---|---|---|---|---|---|
| FNO-3D | 6,558,537 | 38.99 s | 0.0086 | 0.1918 | 0.0820 | 0.1893 |
| FNO-2D | 414,517 | 127.80 s | 0.0128 | 0.1559 | 0.0973 | 0.1556 |
| U-Net | 24,950,491 | 48.67 s | 0.0245 | 0.2051 | 0.1190 | 0.1982 |
| TF-Net | 7,451,724 | 47.21 s | 0.0225 | 0.2253 | 0.1168 | 0.2268 |
| ResNet | 266,641 | 78.47 s | 0.0701 | 0.2871 | 0.2311 | 0.2753 |

*Notes.* Reported at 64×64 spatial resolution; FNO-3D convolves in space–time while FNO-2D uses 2D convolutions with an RNN in time.

**PirateNets** has PINN backbone with physics-informed residual adaptive blocks. The paper emphasizes accuracy comparisons and ablations; it does not tabulate parameter counts, GPU memory, or wall-clock per epoch. Below we list the state-of-the-art test errors the authors report.

Table 25: PirateNets: reported relative $\ell_2$ test errors across PDEs (paper's Table 1).

| Benchmark | Error (PirateNet) | Params | GPU Mem | Time/epoch |
|---|---|---|---|---|
| Allen–Cahn (1D) | $2.24\times10^{-5}$ | — | — | — |
| Korteweg–De Vries (1D) | $4.27\times10^{-4}$ | — | — | — |
| Grey–Scott (2D) | $3.61\times10^{-3}$ | — | — | — |
| Ginzburg–Landau (2D) | $1.49\times10^{-2}$ | — | — | — |
| Lid-driven cavity (2D) | $4.21\times10^{-2}$ | — | — | — |

*Notes.* Architecture details (e.g., depth/width) and training pipelines are provided, but resource metrics are not tabulated.

**PINNsFormer** is a transformer-style PINN variant. The authors report parameter counts and training overhead (V100), and test errors on overlapping 1D PDEs.

Table 26: PINNsFormer: model size and training overhead (Appendix Table 4–5 in the paper).

| Model | Params | GPU Mem (MiB) | Time/epoch (s) |
|---|---|---|---|
| PINNsFormer (pseudo-seq. length $k=5$) | 454,000 | 2,827 | 2.34 |

*Notes.* Reported on a single NVIDIA Tesla V100; overheads shown for $k=5$.

Table 27: PINNsFormer: reported test errors on 1D PDEs used widely in PINN literature.

| PDE (dimension) | Metric (paper) | Error | Params | Time/epoch / GPU Mem |
|---|---|---|---|---|
| Convection (1D) | rRMSE ($\approx$ rel. $\ell_2$) | 0.027 | 454k | 2.34 s / 2,827 MiB |
| Reaction (1D) | rRMSE ($\approx$ rel. $\ell_2$) | 0.030 | 454k | 2.34 s / 2,827 MiB |

*Notes.* Errors are taken directly from the paper's main results tables; rRMSE is the paper's standard relative $\ell_2$ metric. The reaction/convection formulations and sampling follow the setups specified in Zhao et al. (2023).

# E  A Discussion on Relevant Ideas in Different Structures

In this section we discuss three different frameworks that leverage Fourier structure for PDEs in different ways; SAFE-NET, FNO Li et al. (2020b), and EPGP Härkönen et al. (2023). Although all three exploit Fourier structure, they do so for different learning objectives and with complementary inductive biases.

**Problem formulation and learning objective.**  SAFE-NET is a physics-informed solver for a single PDE instance: we fit $u_\theta$ by minimizing a residual-based PINN loss with collocation points for the PDE, BCs and (if present) ICs; no paired input-output simulation dataset is required. FNO Li et al. (2020b), by contrast, learns a resolution-invariant solution operator $G^\dagger \colon a \mapsto u$ from many paired fields $\{(a^{(i)}, u^{(i)})\}_{i=1}^N$; once trained it solves new instances in one forward pass and even on unseen grids (zero-shot super-resolution). EPGP from Härkönen et al. (2023) is probabilistic: it places a GP prior whose sample paths satisfy a linear, constant-coefficient PDE exactly; conditioning on scattered observations (values and/or derivatives) yields a posterior that remains within the PDE solution set.

**Spectral parameterization.**  All three approaches leverage Fourier structure, but in different ways.

- **SAFE-NET** uses a finite, trainable tensor-product Fourier feature map in space (and time), with sine and cosine cross terms and learned amplitudes. Both the harmonically-initialized frequencies and amplitudes are optimized jointly with the network via the physics loss. This couples spectral adaptivity to the governing equation and initial-boundary condition information embedded in the loss.

- **FNO** parameterizes convolutions in Fourier space: the FFT of the feature field is multiplied by learnable complex matrices on a fixed set of retained modes and transformed back by an inverse FFT. In other words, each Fourier layer applies $v_{t+1}(x) = \mathcal{F}^{-1}\big(R \cdot \mathcal{F} v_t\big)(x) + W v_t(x)$, where $R$ is a learnable complex tensor restricted to low modes and $W$ a local linear map; frequencies themselves remain fixed and are not learned; instead, a (problem-dependent) truncation $k_{\max}$ selects the modes that are updated, while higher modes are filtered out Li et al. (2020b). This yields global, resolution-invariant layers well suited to operator learning.

- **EPGP** in Härkönen et al. (2023) expresses solutions through the Ehrenpreis–Palamodov representation (a spectral integral over the PDE's characteristic variety) as the following kernel

$$k_{\text{EPGP}}(x, x') = \int_{\sqrt{-1}\mathbb{R}^d} \Psi(x, z)\, \Psi^*(x', z)\, e^{-\|z\|^2/2}\, \mathrm{d}z, \tag{46}$$

  with $\Psi(x, z) = \sum_{j, z \in S_{z'}} D_j(x, z) \exp\langle x, z\rangle$. A Monte-Carlo variant (S-EPGP) replaces the integral by a finite sum of trainable plane waves, so the relevant spectral locations are learned within the GP kernel.

**Data/physics requirements and scope.**  SAFE-NET relies on the PDE residual (and BC/IC) and targets low-to-moderate-dimensional PDEs (including smooth and mildly non-smooth regimes), where Fourier bases provide a good inductive bias. FNO excels when sufficiently many input–output pairs of fields are available; its strength is generalization across instances and grids rather than solving a single instance from physics alone. EPGP is model-based and probabilistic: it assumes linear PDEs with constant coefficients and incorporates data at arbitrary locations (values and/or derivatives).

**Boundary/initial conditions and domain assumptions.**  SAFE-NET feeds BC/IC conditions, if available, directly as domain features (e.g., sine series that satisfy homogeneous Dirichlet BCs) and trains end-to-end with the PINN loss. FNO typically assumes periodic structure within each Fourier layer (the convolutional kernel is defined for periodic functions), although practical variants can handle nonperiodic domains with padding or localized corrections. EPGP does not require prescribing BCs/ICs to define the prior; when BC/IC observations are available, they are simply assimilated as GP conditioning data, and the posterior remains in the PDE solution set.

**Computational aspects.** Each of these architectures solve different problems so they should not be compared directly based on computational cost. However, for the sake of completion, we provide computational footprint details for each method. SAFE-NET uses a single hidden layer and a small trainable feature module, giving low parameter counts and fast epochs suitable for residual-based training. FNO stacks multiple FFT-based layers; training scales as $\mathcal{O}(Nn\log n)$ but inference on new grids is extremely fast. EPGP shifts cost to kernel construction $\mathcal{O}(n^3)$ (or $\mathcal{O}(nm^2)$ for its sparse variant) while providing calibrated uncertainty. With sparse/Monte–Carlo kernels, the dominant cost is one-time covariance construction.

**Complementarity.** FNO works well when abundant multi-instance data are available, offering orders-of-magnitude speed-ups at deployment and results generalizable within the parametrized family of PDEs. SAFE-NET is best when the PDE (best for PDEs with moderately smooth behavior) is known but training data are scarce, computational budget is limited, or when we seek both accuracy and efficiency together. EPGP delivers principled uncertainty for linear PDEs and fuses scattered measurements seamlessly.

