# OpenReview forum: "Enhancing Physics-Informed Neural Networks Through Feature Engineering"
_TMLR — Accepted by TMLR_

### Review · Reviewer_LL46 · 2025-08-02

**Summary Of Contributions:**

The paper constructs specific features suitable for differential equations, such that PINNs are significantly easier to train using these features. These feature use Fourier transforms and Fourier series, amongst others. Careful experiments are conducted on small neural networks, which show that the method SAFE-NET, introduced in this paper, is superior in quality to baseline models, in particular for linear equations.

**Audience:**

Yes

**Claims And Evidence:**

Yes

**Requested Changes:**

1. You might want to check your placement of figures. Sometimes they appear on seemingly random positions of the page.
2. The text in the images used in the figures is much too small to be readable. Please enlarge it at least to the size of footnotes.
3. Introduction: "PDE solutions have well-understood mathematical structures rooted in Fourier analysis, differential geometry, and physics principles" I think this sentence might profit from references into the literature, or making an entire paragraph out of it.
4. On page 6, the systematic feature extraction method is described really vague. How does this work?
5. Page 6 comments on "prioritizing" certain frequencies for homogeneous Dirichlet boundary conditions. How does this work in detail? Can you also comment on the connection of this to Fourier series?
6. While I get that comparisons against mall models is reasonale here, can you comment on accuracy, traning times and memory for larger models, e.g. by citing their papers?
7. In the first, unnumbered, equation on page 15, you have a limit $p\\to\\infty$, but no $p$ appears afterwards in the equation (ok, it is the number of parameters, but this is not obvious locally.).
8. In appendix A.1.4, using i as a running variable is somewhat confusing, since $i=\\sqrt{-1}$ also makes sense.
9. In appenxis B.3, there is a typi $x in ...$ instad of $x\\in ...$.
10. Some relations hold to various neural operator approaches, e.g. FNO, which you are citing. Can you comment more on similarities and differences?
11. A similar approach called EPGP also trained Fourier frequencies and amplitudes for Gaussian processes, but not in a grid. Can you comment on similarities/differences?
12. The paper "Sifan Wang, Xinling Yu, and Paris Perdikaris. When and why pinns fail to train: A neural tangent kernel perspective. Journal of Computational Physics" appears twice in the bibliography.
13. The paper "Attention is all you need" is not from 2023, and it is not just a preprint.

**Strengths And Weaknesses:**

Strength:

1. The experiments are very fully described.
2. Making many ablations w.r.t.optimization schedules and an NTK-based theoretical discussion about loss landscapes, which also give additional insights into problems with PINN-training

Weaknesses:

1. While some justification is given as to why no large scale experiments are being performed, I would be intereed in at least 1-5 training runs.
2. While some intuition is given about the features, I cannot say that the deep connection between choosing the features precisely this way and the theory of differential equations.

Questions

1. I am confused about the notation in your first paragraph. I a potentially existing time part of the domain $\\Omega$? If so, the initial condition should probably just be evaluated at $\\partial\\Omega$. Can you clarify?
2. You are considering PINNS "just" for solving PDEs. They are also a regression tool in the context of PDE, in case data is given. Perhaps you might want to clarify your stance on this distinction. Does your approach also work in this regression setting? In particular, you seem to assume knowledge about the Fourier transform of initial and boundary conditions. This might or might not be a reasonable assumption, depending on circumstances. Can you comment here?
3.  You are learning a (non-regular) grid of Fourier frequencies. How does this scale to high input dimensions? Did you try experiments for higher dimensions? (Usually, an advantage of PINNs over methods from scientific computing is, that PINNs are less vulnerable against the curse of dimensionality. Here, it seems, you get a strong curse of dimensionality.) The question holds particularly since your experiments use at most 2 input dimensions (Table 2, wven though if I understand correctly, having "time dependent" here means you use an additional input dimension?). Does your approach scale to 3 or 4 dimensions? How long would computations take? (Yes, I read your comment at the end of Section 3, but can you say more here?)
4. Fourier based theory for differential equations is most relevant for linear differential equations. Can you comment on differences in the performance of your approach between linear, close-to-linear, and highly non-linear PDEs? See also Figure 3, where Safe-Net performes particularly well for the linear equations and at best mediocre for non-linear equations.
5. Are the learned frequencies in any way related to the frequencies in the characteristic varieties of the linear PDEs or to the Fourier frequencies of the initial conditions?

---

> ### Author Response · Authors · 2025-08-23
> **Response to Reviewer LL46 (1/2)**
>
> Thank you for your time and your constructive feedback! We have carefully considered your comments and questions to revise our work.
>
> Large-scale experiments:
>
> Table 2 lists spatial dimensions – typical in PINNs papers. Our experiments are on 2D and 3D PDEs (temporal and spatial dimensions combined), so our largest is already 3D.  We follow the standard PINN benchmark suite to have a fair comparison with existing methods, especially with other feature engineering methods. Other PINN architectures e.g. [1], [2], [3] and the baselines we use run on these PDEs with similar IC/BC specifications. Other PINN architectures, even the larger more complex ones in [4] [5] [6], use these common benchmarks with at most 3 dimensions to indicate performance too. We also did ablation studies on deeper networks (Appendix C.1.4), which led to minimal performance changes overall. Besides these experiments, what other different large scale experiments would you like to see? We would be glad to report results on those and include them in the revised version of SAFE-NET!
>
> **Questions (in order)**
>
> 1. Thanks for catching the confusing formulation of the general PDE in the first paragraph. In the revision, I is removed and B is left as a general IC/BC-condition operator evaluated at $\partial \Omega$
>
> 2. You bring up an interesting point! SAFE‑NET is aimed at forward PDE solves where the differential operator and the IC/BC functions are known. If only scattered solution data are available (regression), SAFE‑NET still works, but the domain knowledge features requiring IC/BC forms would not be available.
>
> 3. We added a small note in Table 2’s caption for further clarity about spatial dimensions. As previously noted, we use 2D and 3D PDEs (temporal + spatial dimensions), so our largest experiment is already 3D. In 4D, $2^4 = 16$ combinations of sine/cosine products per frequency tuple $(n_1, n_2, n_3, n_4)$ exist. For a modest count of frequency tuples and trainable amplitudes for each term, the computational load becomes excessive. Timing estimates for a hypothetical 4D heat problem will be added to the revision’s appendix.
>
> 4. SAFE-NET excels on smooth, relatively well-behaved shock-free PDEs (including all the linear PDEs in Table 2), outperforming other methods by an order of magnitude in some cases (Fig 6). It also outperforms others on mild nonlinearities (e.g. the reaction PDE). It is less suited to highly non-linear PDEs with sharp discontinuities or shocks. We observe this for the Burgers’ equation; SAFE-NET’s result is good in the sense of “accuracy + effort/efficiency” trade-off (L2RE 2.67e-03 with more than 50% fewer parameters than the top performing baselines), but it is not the top choice for such PDEs. RBF-based features can handle local sharp gradients more effectively, which we acknowledge and discuss in the paper.
>
> 5. SAFE‑NET’s domain‑knowledge features come from the IC/BCs (e.g. in the wave and heat problems with initial conditions $u(x, 0) = \sin(\pi x) + \frac{1}{2} \sin(\beta \pi x)$ and $u(x, y, 0) = \sin(20\pi x) \sin(\pi y)$ respectively; Section 3.2) so characteristic frequencies within ICs inform SAFE-NET directly. We recorded the learned adaptive frequencies for the wave task and found they remain close to the analytic solution as expected! A plot and discussion will be included in the revision’s appendix.
>
> **Requested Changes**
>
> comments 1-3, 7-9, 12-13: We appreciate your careful eye on the presentation! We fixed all of these typos and added more references on the suggested areas.
>
> Comments 4-5: SAFE-NET feature selection details are in Appendix  A.1.2 under “Systematic Domain Knowledge Feature Extraction”. To make the process more clear in the revised version, we added examples of two PDEs and their IC/BC feature extraction steps to the main text on page 6 and a detailed flowchart diagram of the whole feature module.
>
> Comment 6: Sure! We added Appendix D in the revision to list  FNO [4], Piratenets [5], PINNsformer [6]: parameter count, GPU memory, wall‑clock per epoch, and L2RE on the shared benchmarks. As an example, [4] reports test L2RE 0.0149 for 1D Burgers (same formulation as us, only the numerical value of the parameter $\nu$ is different). PINNsformer [6] reports test L2RE 0.030 for the our exact reaction PDE using 454k parameters, training time 2.34 (sec/epoch) on overage, and GPU memory 2827 (MiB).
>
> Comments 10-11: Thank you for suggesting EPGP [7] as it is an interesting approach! Operator nets aim for family generalization across PDE parameter spaces, whereas SAFE‑NET targets single‑instance solves efficiently. EPGP embeds the Ehrenpreis-Palamodov spectrum inside a GP kernel, marginalizing all frequencies and solving linear constant-coefficient PDEs. SAFE‑NET keeps the Fourier modes explicit and trainable into a lightweight PINN. A full detailed comparison to FNO‑type operator learners and to EPGP/Gaussian‑process latent‑force models is added as Appendix E to the revision.

---

> > ### Author Response · Authors · 2025-08-23
> > **Response to Reviewer LL46 (2/2)**
> >
> > References
> >
> > [1] C. Zeng, T. Burghardt, and A. M Gambaruto. Feature mapping in physics-informed neural networks (pinns), 2024. arXiv:2402.06955
> >
> > [2] S. Wang, X. Yu, and P. Perdikaris. When and why pinns fail to train: A neural tangent kernel perspective, 2022.  arXiv:449:110768
> >
> > [3] S. J. Anagnostopoulos, J. Toscano, N. Stergiopulos, and G. Karniadakis. Residual-based attention and connection to information bottleneck theory in pinns, 2023. arXiv:2307.00379.
> >
> > [4] Z. Li, N. Kovachki, K.Azizzadenesheli, B. Liu, K. Bhattacharya, A. Stuart, and A. Anandkumar. Fourier neural operator for parametric partial differential equations, 2020. arXiv:2010.08895
> >
> > [5]  S. Wang, B. Li, Y. Chen, and P. Perdikaris. Piratenets: Physics-informed deep learning with residual adaptive networks. arXiv:2402.00326
> >
> > [6] Z. Zhao, X. Ding, and B. Aditya Prakash. Pinnsformer: A transformer-based framework for physics-informed neural networks, 2024. arXiv: 2307.11833.
> >
> > [7] M. Härkönen, M. Lange‑Hegermann, and B. Raiţă. Gaussian Process Priors for Systems of Linear Partial Differential Equations with Constant Coefficients, 2022. arXiv: 2212.14319

---

> > > ### Comment · Reviewer_LL46 · 2025-08-29
> > >
> > > Thx for your answers!
> > >
> > > The concerns from my own review are addressed, as long as the final version of the paper contains the restrictions (dimension <=3, smoothness) prominently in the introduction.
> > >
> > > The concerns from Reviewer 1 (9qG6) are also adequately addressed from my point of view.
> > >
> > > The concerns from Reviewer 2 (Pn2M) are only partly addressed, from my point of view. I will sit that discussion out, since Reviewer 2 seems more knowledgeable in this area.

---

> > > > ### Author Response · Authors · 2025-09-01
> > > > **Thank you very much for your response!**
> > > >
> > > > Hi, hope you are having a wonderful day! We have now replaced our original submission with a revised version of our paper. In it, we have made the corrections you suggested such as clarifying dim $\leq 3$ and best performance with smoothness in the introduction, other clarifications as you listed in your review, fixed typos and formulations you pointed out, and added new sections in the appendix; Appendix D includes reports of the experimental setups and results on shared baselines for large-scale frameworks directly from their papers per your suggestion. Appendix E now contains a discussion on approaches leveraging Fourier structure such as EPGP, citing their papers. We would be very happy if you let us know of further additions you would like. Thank you again for your constructive comments and for helping us improve this work!

---

### Review · Reviewer_Pn2M · 2025-08-14

**Summary Of Contributions:**

The key idea of the work is train PINNs using simple model which hand-crafts a functional transform of the inputs and then passes it on to a single layer of activation. The hand-crafted features are a linear combination of Fourier-like modes in space and temporal variables with learnable frequencies.

The experimental evidence is very thin in terms of covering conventional hard cases although for the simple exemplars chosen the authors have given extensive ablation studies for the algorithmic and the architectural choices.

And even among the chosen simple tests, the given data is ambiguous --- for 50\% of the PDEs i.e Diffusion, Allen-Cahn, Burgers and Navier-Stokes the data in Table-3 says that SAFE-NET is not the one with the lowest error. So it seems that the author's own data undermines their proposed model!

But then again these could be just strange artefacts of the unexplained capping of the models to just one layer of activation - this choice is possibly fundamentally affecting the results but that has never been highlighted or explained!

There is some attempt to justify the idea in terms of saying that "SAFE-NET enjoys an implicit preconditioning effect that is global" - but there is nothing rigorous about this idea visible in the draft - not even a formally defined conjecture.

As it stands, this paper needs a significant and almost total overhaul to be accepted.

**Audience:**

Yes

**Claims And Evidence:**

No

**Requested Changes:**

- The paper needs to be entirely rewritten/recalibrated to include non-trivial PDE examples. The demonstration choices made (Table 2) are not at all well-chosen to serve as an indicator of performance. Of the 8 PDEs chosen, none have 3 or more variables and 6 are having only one or 2 variables and these are not even known to be classically hard PDEs.

  So, firstly we need to see evidence of performance in high number of variables. PDEs like Burger, anisotropic linear Fokker-Planck, HJB and heat are some non-trivial examples where exactly solvable cases are available in every dimension. Hence these can be good test cases to benchmark against --- particularly the non-linear ones.

- One can see the kind of examples that were chosen for demonstration in this work, https://arxiv.org/abs/2407.20741. The examples here not only consist of 4 variable PDEs but also consider classically hard PDEs for even 2 variable PDEs like the 3-solitonic solution for KdV and the methods therein were shown to supersede the erstwhile best known solution for that in terms of needing smaller models. Such a demonstration is needed to make SAFE-NET a credible competitor and for that these same benchmarks can be tested against.

- Also for the simple one and 2 variable PDEs chosen in this work, the boundary and initial conditions are simple enough that they could be incorporated into the model as shown in the demonstrations in the above paper. So, with SAFE-NET, we definitely need this study whether SAFE-NET with inclusion of boundary and initial conditions in the model supersedes a standard PINN with the same inclusions in the model.

- The point about the ill-conditioning needs to be much more rigorously said. If the authors want to claim that "SAFE-NET enjoys an implicit preconditioning effect that is global" then this needs to be said as a mathematical conjecture - currently even the notion of preconditioning as in this context has not been defined. In Appendix C.3.1 there is a discussion on the spectral density, but its not clear as to what is the formal conjecture about the spectral density of the Hessian of SAFE-NET that they want to make. There is a need for a lot more clear conjectures and showing how the evidence aligns with it.

- Lastly, its entirely unclear why the authors are stuck to using only one layer of activation. Maybe SAFE-NET performs great at high-depth and its not clear why these obvious experiments were not done. Or, if the authors want to claim that among all depth-2 models for PDE solving, SAFE-NET is the optimal choice, then that needs to be said that way and the whole paper needs to be positioned that way. Currently this capping of the depth to 2 looks entirely artificial.



Minor Issues,

- the definition of the tangent kernel in Definition 6.1 seems to have a typo wih its evaluation points.

- Section 5.1 in its opening line attempts to classify the PINN baselines in 2 categories but that organization is not maintained when the actual review is done in Appendix A.2. This is confusing writing and needs looking into. When defining RBF-PINN, the variable $c_i$ has not been defined properly.

- For the one variable setup, a classic hard example to test would be the energy eigenfunctions of the quantum harmonic oscillator. At quantum numbers $3$ or more, these are strong targets for contest between vanilla PINNs  and any other modification.

**Strengths And Weaknesses:**

The strength of the paper is the ablation study done on the methods of training as can be seen in Appendices C and A.3.

The key weaknesses of the paper are the misplaced choices of demonstration and lack of rigour in making the point about ill-conditioning.

---

> ### Author Response · Authors · 2025-08-28
> **Response to Reviewer Pn2M (1/2)**
>
> Thank you for your time and your feedback on our submission. Our responses to your comments are as follows.
>
> 1. > “Of the 8 PDEs chosen, none have 3 or more variables and 6 are having only one or 2 variables … not even known to be classically hard PDEs.”
>
>   > “... very thin in terms of covering conventional hard cases”
>
>   > “The demonstration choices made (Table 2) are not at all well-chosen to serve as an indicator of performance. “
>
> **Dimensionality:** We use **2-variable and 3-variable** PDEs. Table 2 lists spatial dimensions – typical in PINNs papers. Every time‑dependent task adds one temporal coordinate. The heat task is a **3-variable** PDE. The temporal coordinate counts for SAFE-NET as it uses Fourier basis terms with all spatial and temporal variables. Details of each PDE, their variables, etc. appear in Appendix B.
>
> **Well-chosen to indicate performance:** The eight PDEs we report already cover **(i) linear, mildly nonlinear, and highly nonlinear dynamics**, **(ii) time‑dependent and steady‑state problems**, and **(iii) smooth and discontinuous PDEs with shocks** (Table 2). We follow the standard PINN benchmark suite to have a fair comparison with existing methods, especially with other feature engineering methods. Other PINN architectures e.g. [1]-[3] and the other baselines we use run tests on these PDEs with similar IC/BC specifications. Other PINN architectures, even the larger more complex ones in [4]-[6], use these common benchmarks to indicate performance. To show robustness, we added a non-homogeneous BC variant of the heat equation to the revision, on which SAFE-NET still outperformed the competitors with an L2RE of 2.11e-04.
>
> **Difficulty:** These PDEs remain challenging even for large complex architectures. [4] reports L2RE 0.0149 for 1D Burgers (same formulation, only the numerical value of the parameter $\nu$ is different, but it still develops a shock like in our case). [6] reports L2RE 0.030 for the exact reaction PDE we test. These models use hundreds of thousands of parameters to learn. We achieve comparable results (Fig 6) using $70-100 \times$ fewer parameters (Table 1) more efficiently (Fig 4).
>
> 2. > “PDEs like Burger …. and heat are some non-trivial examples … these can be good test cases to benchmark against”
>
> Thank you for highlighting that Burgers and heat are considered good non-trivial test cases. This view appears to contrast with the earlier comment suggesting that the PDEs we tested on were not considered hard, given that both Burgers and heat are among those we did test. We use the 3-variable heat PDE (2 spatial + 1 temporal) and the 2-variable Burgers (1 spatial + 1 temporal). These, and the other PDE benchmarks we used, are widely adopted in the PINN literature, including in works employing larger or more complex architectures [1]–[6].
>
> 3. > “for 50% of the PDEs i.e Diffusion, Allen-Cahn, Burgers and Navier-Stokes the data in Table-3 says that SAFE-NET is not the one with the lowest error.”
>
> Table 3 uses optimization schedule (1) (line 12 of Section 5.2) that works better for some baselines. With that, SAFE-NET consistently ranks 1st on wave, reaction, heat, and convection tasks. For diffusion it’s a close second, and on PDEs with shocks and discontinuities (e.g. Burgers, Allen-Cahn) it comes in third even though it is not designed to handle sharp discontinuities well unlike RBF kernels, which we acknowledge several times in the paper. When we switch to optimization schedule (2) (again Section 5.2), which is demonstrably better aligned with SAFE‑NET as seen in Table 4, SAFE-NET continuously ranks first on the first 5 tasks and improved performance on PDEs with discontinuities and shocks. SAFE-NET’s goal is the “accuracy + effort/efficiency” trade-off. We attain $95\%$ faster epochs than other feature‑engineering methods (Fig. 4) with 6.8 k parameters vs. 14 k -16 k for the competitors.
>
> 4. > “we need to see evidence of performance in high number of variables”
>
> SAFE‑NET is designed for low/moderate-dimensional PDEs. Extending feature engineering to truly high‑dimensional problems as you suggest is important future work, but presents very different challenges that are currently beyond the reach of PINN-style methods.
>
> 5. > “... study whether SAFE-NET with inclusion of boundary and initial conditions in the model supersedes a standard PINN with the same inclusions in the model.”
>
> Good question! We conducted a simple ablation study comparing 4 architectures: SAFE‑NET $\pm$ domain‑knowledge features (DKF) vs. PINN $\pm$ DKF. Adding only DKF to PINN improves the final L2RE by <10% on average. We added this ablation study to Appendix C.1 in the revised manuscript.

---

> ### Author Response · Authors · 2025-08-28
> **Response to Reviewer Pn2M (2/2)**
>
> **Continuation of our responses to your comments**
>
> 6. > “... not clear as to what is the formal conjecture about the spectral density of the Hessian of SAFE-NET”
>
> Definition is added to the main text in the revision for more clarity. Empirical observations (spectral density) show SAFE-NET improves conditioning at the early stages of training and at the end, and we use an early-stage idealized example there in Section 6 only to build intuition of SAFE‑NET’s conditioning benefit, not as a formal conjecture. We have clarified the assumptions, conditions, and goal of these calculations in the revision.
>
> 7. > “... entirely unclear why the authors are stuck to using only one layer of activation”
>
> > “Maybe SAFE-NET performs great at high-depth and its not clear why these obvious experiments were not done”
>
> These experiments are done. Appendix C.1.4 (Fig. 19) already contains a depth ablation from 1 to 6 hidden layers (with activation). Beyond one layer minimal performance change is observed (median L2RE improves by <7 %) while parameter counts and runtime grow, increasing the computational cost significantly, contradicting our goal with SAFE-NET.
>
> 8. > “Minor Issues, …”
>
> We appreciate your careful eye on the presentation. We fixed all of these in the revised text.
>
> References
>
> [1] C. Zeng, T. Burghardt, and A. M Gambaruto. Feature mapping in physics-informed neural networks (pinns), 2024a. arXiv:2402.06955
>
> [2] S. Wang, X. Yu, and P. Perdikaris. When and why pinns fail to train: A neural tangent kernel perspective, 2022a.  arXiv:449:110768
>
> [3] S. J. Anagnostopoulos, J. Toscano, N. Stergiopulos, and G. Karniadakis. Residual-based attention and connection to information bottleneck theory in pinns, 2023. arXiv:2307.00379
>
> [4] Z. Li, N. Kovachki, K.Azizzadenesheli, B. Liu, K. Bhattacharya, A. Stuart, and A. Anandkumar. Fourier neural operator for parametric partial differential equations, 2020b. arXiv:2010.08895
>
> [5]  S. Wang, B. Li, Y. Chen, and P. Perdikaris. Piratenets: Physics-informed deep learning with residual adaptive networks. arXiv:2402.00326
>
> [6] Z. Zhao, X. Ding, and B. Aditya Prakash. Pinnsformer: A transformer-based framework for physics-informed neural networks, 2024. arXiv: 2307.11833.

---

> ### Comment · Reviewer_Pn2M · 2025-08-30
> **Thanks.**
>
> Thanks for trying to write a detailed reply.
>
> Unfortunately, I am not seeing any critical issue having been addressed.
>
> - I don't think there is any confusion about the counting of PDE variables. As per appendix B, of the 8 PDEs tested, *only one* (heat)  has 3 variables. And even with these small variable PDEs, most of these are not classically hard. So this is at best a sanity test of the method and not a publication level demonstration of a new method in 2025. I pointed out using the reference I linked to, as to what kinds of tests constitute classically hard tests.
>
>    Unless most of the demonstrations are with 3 or more variables, I cant see a new PINN method to have a discovery claim, particularly so in a paper which is non-theoretical.
>
>     The entire logic of PINNs is about avoiding curse-of-dimensionality in high-dimensions.
>
>     In low dimensions, PINNs will be easily beat by even standard FEM/V kind of methods for almost all cases (even with shocks!)!
>
> - You are misinterpreting my point about heat and Burger being good test cases. My point is that these PDEs have settings which yield exact solutions in arbitrary number of variables. So for high (3 or more variable) tests, its reasonable to use the exact solution setups of these PDEs as the test cases.
>
> - If the claim is that Table-4 shows that the "Optimization Schedule (2)" for SAFE-NET is the best then it should have been given in the same form as Table-2 showing comparison with baselines. This presentation is very confusing.
>
> - If the point is to show the "“accuracy+ effort/efficiency” trade-off" then that needs a very different kind of plot showing some kind of Pareto-optimility for SAFE-NET w.r.t other competitors in Appendix A.2.
>
> - Where is this study that you call "SAFE‑NET
>  domain‑knowledge features (DKF) vs. PINN
>  DKF."? I dont see any table or Figure in C.1 which corresponds to this. What am I missing?
>
> - I can still see this vague thing being said "SAFE-NET enjoys an implicit preconditioning effect that is global" and there is still no formal conjecture about what is the proposed relationship between spectral density of the Hessian and the efficacy of the PINN method.
>
> - Its not reasonable to compare performances against operator learning benchmarks given that they solve a wholly more complicated target of learning an infinite-dimensional map than just PDE solving.
>
> In summary, it seems like none of the major issues have been fixed - in particular, high-variable demonstrations are still missing. So I am not seeing any reason to change my evaluations. My sincere apologies and I look forward to seeing the required demonstrations.

---

> > ### Author Response · Authors · 2025-09-02
> > **Our response (1/2)**
> >
> > Thank you for your comments! We address them point-by-point in their original order as follows.
> >
> > - We are glad that the matter of the number of variables in our PDE tasks is clarified. As noted in the Introduction and again in Section 4 (Related Work): “Much work has been done to improve the training of PINNs that take different approaches from feature engineering”. Each of them makes different design choices to optimize a different axis of the design space and therefore may perform best on specific PDE categories. Our study focuses more on feature engineering in PINNs, testing a number of Fourier and non-Fourier feature engineering methods on a diverse set of low-to-moderate-dimensional PDE tasks that span (i) linear, mildly nonlinear, and highly nonlinear dynamics, (ii) time‑dependent and steady‑state problems, and (iii) smooth, discontinuous, and shock-forming. To fairly compare with other baseline methods with comparable parameter counts, feature engineering baselines in particular, we chose these standard tasks from our community benchmarks from PINNs literature published in **2024**, such as [1-2] (feature engineering), [3-4] (large scale PINN-style models) or other recent work such as [5-6]).
> >
> > > “... not a publication level demonstration of a new method in 2025”
> >
> > The following contains our baseline Fourier and non-Fourier papers, their publication years, and information about the PDE tasks they use:
> >
> > **Zeng et al. [1] (ICLR 2024)** handle a range of PDEs including wave, diffusion, heat, Poisson, Burgers, and Navier–Stokes equations. Their setting involves 2–3 variables (except for Poisson), and the PDEs include both linear and non-linear types. In terms of regularity, 5 out of 6 are smooth or moderately smooth, and one includes a discontinuity.
> >
> > **Anagnostopoulos et al. [2] (CMAME 2024)** focus on the Allen–Cahn and Helmholtz equations with 2 variables. These also span linear and non-linear types, and include both smooth and non-smooth cases.
> >
> > **Wang et al. [7] (CMAME 2021)** cover Poisson, heat, wave, and reaction–diffusion systems, with 1–3 variables. Among these, approximately 3/4 are linear and 1 is non-linear. Similarly, about 3/4 are smooth and 1 is non-smooth.
> >
> > **Focus of SAFE-NET:** Fourier feature engineering leading to a design that achieves an excellent accuracy + efficiency/effort trade-off compared with other well-performing Fourier and non-Fourier feature engineering baselines for low-to-moderate dimensional PDEs.
> >
> > **Not the focus:** high-variable cases.
> >
> > Our choice of PDEs based on the baselines: Wang et al. is a classic method incorporating fourier features so it has to be among the baselines. The rest of these feature engineering papers are published in 2024. Zeng et al. includes two variants and we test both of them, (RBF and RBF-P). The Poisson task is both linear and smooth, not a strong test case for SAFE-NET in low-to-moderate dimensions (which is our target regime).  We thus follow our PDE mix, drawing them from standard commonly used PDE benchmarks.
> >
> > Now, the paper you refer to in [9] uses a different strategy to learn PDE solutions. They use standard PINN and embed a deterministic algebraic pre-map so the network already satisfies either BCs or ICs, allowing tests on higher dimensional tasks by design. They test on Poisson (with exact analytical solution available for all), Burgers (up to 3D with exact analytical solution available for all), and KdV (2D with exact analytical solutions available for each case).  By contrast, SAFE-NET tackles the representation bottleneck and our scope is low-to-moderate-dimensional tasks by design. Poisson is linear and smooth, it would not be a strong additional test case for SAFE-NET in low-to-moderate dimensions as we already have similar linear and smooth PDEs. We already include Burgers. The remaining task is KdV, and to fully address this matter and be thorough, we will include a variant of the 2D KdV (by citing your reference) in the final revision as well.
> >
> > All in all, we conducted this study of feature engineering in PINNs through SAFE-NET’s design with the goal of improving the accuracy vs. efficiency (as well as observed problem conditioning) for a number of practical PDE tasks, plus extensive ablations to clarify design choices and lead to a more insightful work. We hope this provides an engaging reference point for the PINN community exploring feature engineering and for future paths that improve and combine it with other great strategies; ultimately moving PINNs closer to the dream of strong efficient performance on truly high-dimensional problems.

---

> > > ### Author Response · Authors · 2025-09-02
> > > **Our response (2/2)**
> > >
> > > - Thank you for your suggestion of using PDEs such as heat and Burgers under conditions with exact analytical solutions in arbitrary numbers of variables. We explained our PDE choices above for the scope of the current work. Improving SAFE-NET and extending this idea (by possibly making use of multi-stage training, incorporating physical priors within the network architecture, etc. as discussed in our discussions section) are important feature work, following this path forward. We will definitely use your suggestion for PDE test cases with exact analytical solutions in our feature work as we develop efficient, effective feature engineering for truly high-dimensional setups.
> > >
> > > - Thank you for this suggestion! Optimization Schedule (1) is more common in PINNs training (see the systematic study in [8]), so we introduced it first for orientation. Optimization Schedule (2) is less common, but still well-motivated by Adam and L-BFGS behavior in PINNs training and is demonstrably better aligned with SAFE‑NET as seen in Table 4. If re-ordering the two schedules in the final revision would improve readability, we are happy to do so!
> > >
> > > - Thank you for this suggestion! We already report parameter counts and average runtime comparisons in Table 1 and Fig 4, but adding them into a Pareto-optimility plot would help the readers visualize the trade-off directly. We will add this in the final revision.
> > >
> > > - This comparison is in Appendix C.1.6 of the current revision.
> > >
> > > - The sentence “SAFE-NET enjoys an implicit preconditioning effect…” appears in Related Work (Section 4) and immediately directs the reader to Section 5.3. As mentioned before and in Section 5.3, we provide empirical evidence through spectral density plots (Fig 10 and supplementary material in the Appendix) showing that SAFE-NET consistently reduces both the number and the magnitude of large eigenvalues at early and late training stages, improving conditioning. This is pointed out in a sentence right above Fig 10. The empirical results of Section 5.3 are followed by calculations in an example for the early stages of training under idealized conditions (following prior work in [10]) to build better intuition into this empirical observation, not as a formal conjecture. We will rephrase this sentence to avoid confusion.
> > >
> > > - We do not benchmark against operator-learning models. The parameter counts and other information we cite for architectures such as FNO are given only to illustrate the relative model sizes familiar to the readers of the PINNs community and create insights. Table 3 confines all accuracy comparisons to PINN-style baselines with comparable parameter counts. We will clarify this explicitly in the text to prevent misunderstanding.
> > >
> > >
> > > As the discussion period has already ended, we would like to thank you once again the time you have devoted to our submission and the feedback you have provided both on this submission and on directions for future work. Thank you for helping us improve our work.
> > >
> > > References:
> > >
> > > [1] C. Zeng, T. Burghardt, and A. M Gambaruto. Feature mapping in physics-informed neural networks (pinns), 2024a. arXiv:2402.06955
> > >
> > > [2] S. J. Anagnostopoulos, J. Toscano, N. Stergiopulos, and G. Karniadakis. Residual-based attention and connection to information bottleneck theory in pinns, 2023. arXiv:2307.00379.
> > >
> > > [3]  S. Wang, B. Li, Y. Chen, and P. Perdikaris. Piratenets: Physics-informed deep learning with residual adaptive networks. arXiv:2402.00326
> > >
> > > [4] Z. Zhao, X. Ding, and B. Aditya Prakash. Pinnsformer: A transformer-based framework for physics-informed neural networks, 2024. arXiv: 2307.11833.
> > >
> > > [5] Chen, S.; Liu, Z.; Zhang, W.; Yang, J. A Hard-Constraint Wide-Body Physics-Informed Neural Network Model for Solving Multiple Cases in Forward Problems for Partial Differential Equations. Appl. Sci. 2024, 14, 189.
> > >
> > > [6] F. Jiang, X. Hou, and M. Xia. Densely multiplied physics-informed neural networks. arXiv preprint arXiv:2402.04390, 2024.
> > >
> > > [7] S. Wang, H. Wang, and P. Perdikaris. On the eigenvector bias of Fourier feature networks: From regression to solving multi-scale PDEs with physics-informed neural networks. Computer Methods inApplied Mechanics and Engineering, 384:113938, 2021
> > >
> > > [8] P. Rathore, W. Lei, Z. Frangella, L. Lu, and M. Udell. Challenges in training PINNs: A loss landscape perspective. ICML, 2024.
> > >
> > > [9] M. Ren, Z. Fang, K. Li, and A. Mukherjee. Improving PINNs by algebraic inclusion of boundary and initial conditions. arXiv preprint arXiv:2407.20741, 2024
> > >
> > > [10] S. Wang, H. Wang, and P. Perdikaris. On the eigenvector bias of fourier feature networks: From regression to solving multi-scale pdes with physics-informed neural networks. arXiv:2012.10047, 2020.

---

### Review · Reviewer_9qG6 · 2025-08-15

**Summary Of Contributions:**

This paper proposed a single layer network coupled adaptive fourier feature maps to improve the condition of PINN optimization. It is shown that such small network can attain decent accuracy and speed for most linear PDE problems.

**Audience:**

Yes

**Claims And Evidence:**

Yes

**Requested Changes:**

Some of the testing cases are oversimplified as most PDEs are equipped with initial conditions of only one frequency and homogeneous boundary condition. Appendix A.1.2. Most equations are tested with very simple trigonometric initial conditions and homogeneous boundary conditions. Such oversimplification can not fully demonstrate the effectiveness of the proposed SAFE-NET.

The preconditioning analysis in section 6 may not be suitable as it used NTK as the main tool, where it is assumed that the number of model parameters p goes to infinity. This assumption may not be consistent with small size of SAFE-NET, which is one of the main advatange of SAFE-NET over other feature engineering approaches. Furthermore, the preconditioning analysis did not consider adaptive frequencies w and \lambda as model parameters in (11), but directly assuming that \phi^{(i)}_j are orthonormal. This assumption is questionable as the learned frequcies might not be integer multiples of \pi.


Typos:
below eqn: Where -> where
above (13): \Theta_n -> \Theta_u
Misspelled of Convection in Table 3.

Notations:
The formulation (1) is confusing for time-dependent problems; For example, the 2nd equation should be defined on \partial\Omega \times (temporal domain).
Notations in (3) and (4) are not consistent.
Page 5: Frequency notations are not consistent in Figure 2 and the discussion below it.
Superscript of frequncies are somehow missing in (9).

**Strengths And Weaknesses:**

Strength: The proposed model is very simple and clean, yet it achieved great numerical performance over many PDEs. The author also did a comprehensive comparison study with different feature engineering approaches over several benchmark PDEs.

Weakness:
A major problem of this approach is its scalability in dimension. Even if for a 1D time-dependent PDE, the number of basis N in (9) can easily grow beyond 100 with only ten quadrature points per dimension in the 2D Fourier transformer.

The preconditioning analysis in Section 6 is built up unrealistic assumptions, making its justification of the better conditioning weak. Please see details in requested changes.

There are quite many typos and inconsistent notations throughout the paper, including but not limited to the ones listed below. The authors should make a further screening to avoid them.

---

> ### Author Response · Authors · 2025-08-22
> **Response to Reviewer 9qG6**
>
> Thank you for your time and your constructive feedback! We have carefully considered your comments and questions to revise our work.
>
> Scalability of SAFE-NET:
> Your analysis is correct! As acknowledged in Section 3.2 (the number of “engineered” features increases exponentially with dimension) SAFE‑NET is designed for low/moderate-dimensional PDEs. Our experiments are on 2D and 3D PDEs (temporal and spatial dimensions combined), so our largest is already 3D. Other larger, more complex PINN architectures in [4], [5], [6], also use these common PDE benchmarks in at most 3D. Extending feature engineering to truly high‑dimensional problems as you suggest is important future work, but presents very different challenges that are currently beyond the reach of PINN-style methods.
>
> Requested changes:
>
> 1.
> > “Most equations are tested with very simple trigonometric initial conditions and homogeneous boundary conditions.”
>
> Thank you for bringing up this point! The eight PDEs we report already cover **(i) linear, mildly nonlinear, and highly nonlinear dynamics**, **(ii) time‑dependent and steady‑state problems**, and **(iii) smooth and discontinuous PDEs with shocks** (Table 2). We follow the standard PINN benchmark suite to have a fair comparison with existing methods, especially with other feature engineering methods. Other PINN architectures e.g. [1], [2], [3] and the baselines we use run on these PDEs with the similar IC/BC specifications. Even with homogeneous BCs these PDEs remain challenging even for large complex architectures. For instance, [4] reports L2RE 0.0149 for 1D Burgers (same formulation, only the numerical value of the parameter $\nu$ is different, but it still develops a shock like in our case) while [6] reports L2RE 0.030 for the exact reaction PDE we test. These models use hundreds of thousands of parameters to learn. We achieve comparable results (Fig 6) using $70-100 \times$ fewer parameters (Table 1) more efficiently (Fig 4). To address your comment and show robustness, we added a non-homogeneous BC variant of the heat equation to the revision, on which SAFE-NET still outperformed the competitors with L2RE 2.11e-04.
>
> 2. Preconditioning analysis:
> > “it is assumed that the number of model parameters p goes to infinity.”
> > “the preconditioning analysis did not consider adaptive frequencies … directly assuming that \phi^{(i)}_j are orthonormal.”
>
> Thank you for raising this point to help us achieve a more clear revision! Following prior work on NTK analysis (e.g. [7]), we consider the infinite‑width limit to make the tangent kernel spectrum tractable. We will clarify in the revision that the analysis is an **idealized example** and does not hold exactly for finite‑width SAFE‑NET. Its purpose is to build intuition: Fourier features with balanced amplitudes create a near-orthonormal basis at around initialization (A.1.4), so the calculations show non-zero NTK eigenvalues become nearly equal, indicating good *early-stage* conditioning. Adaptive frequencies might break exact orthonormality later in training, and we will make this remark fully clear in the revised version. Empirical observations (spectral density) show SAFE-NET improves conditioning at the early stages of training and at the end of it, and we use this early-stage example with idealized conditions only to build intuition. We have clarified the assumptions, conditions, and goals of these calculations in the revision.
>
> 3. Typos:
>
> We appreciate your careful eye on the presentation! We fixed all these typos in the revised manuscript. You are right that the formulation of the general PDE in the first paragraph is confusing. In the revision, I is removed and B is left as a IC/BC-condition operator evaluated at $\partial \Omega$. Please let us know if any further detail would be helpful!
>
>
> References
>
> [1] C. Zeng, T. Burghardt, and A. M Gambaruto. Feature mapping in physics-informed neural networks, 2024. arXiv:2402.06955
>
> [2] S. Wang, X. Yu, and P. Perdikaris. When and why pinns fail to train: A neural tangent kernel perspective, 2022.  arXiv:449:110768
>
> [3] S. J. Anagnostopoulos, J. Toscano, N. Stergiopulos, and G. Karniadakis. Residual-based attention and connection to information bottleneck theory in pinns, 2023. arXiv:2307.00379
>
> [4] Z. Li, N. Kovachki, K.Azizzadenesheli, B. Liu, K. Bhattacharya, A. Stuart, and A. Anandkumar. Fourier neural operator for parametric partial differential equations, 2020. arXiv:2010.08895
>
> [5]  S. Wang, B. Li, Y. Chen, and P. Perdikaris. Piratenets: Physics-informed deep learning with residual adaptive networks. arXiv:2402.00326
>
> [6] Z. Zhao, X. Ding, and B. Aditya Prakash. Pinnsformer: A transformer-based framework for physics-informed neural networks, 2024. arXiv: 2307.11833
>
> [7] S. Wang, H. Wang, and P. Perdikaris. On the eigenvector bias of fourier feature networks: From regression to solving multi-scale pdes with physics-informed neural networks, 2020. arXiv:2012.10047

---

> ### Comment · Reviewer_9qG6 · 2025-08-30
> **I still think the IC and BC are oversimplified**
>
> Thanks for addressing most of my concerns. I still think the IC and BC are oversimplified.
>
> You are comparing your numerical results with [1][2][3][4] but [2] is a theoretical paper, [4] is considering operator learning so the comparison is not appropriate. I agree with the author that some of the examples are challenging with homogeneous IC and BC like Burgers eqn but for simple equations e.g. B.1 - B.4, especially equipped with homogeneous or simple trigonometric initial conditions, the solution can be easily obtained with high precision by any classical method.
>
> Given that you are claiming in the introduction taht one main advantage of SAFE-net is encoding the ``domain knowledge", i.e. the IC and BC here, these numerical tests are too weak to support such claim, especially considering that
> * seven out of eight PDEs are equipped with homogenous or periodic BC, the remaining one is no-slip BC so again homegneous. I don't think SAFE-net encodes the domain knowledge of BC.
> * for convection, Burgers and diffusion, the IC is simply sin(\pi x), it is hard to convince people that the domain knowledge of IC is playing a role here. The IC of other PDEs are only slightly more complicated, e.g. sum/product of two trignometric functions.

---

> > ### Author Response · Authors · 2025-08-31
> > **Response to Reviewer 9qG6's comment**
> >
> > Thank you very much for your follow-up comment.
> >
> > Our mention of works like [1]–[6] (and more; such as [8]–[10] as you see below) was only to show that these PDEs and conditions are commonly used across a wide range of PINN literature –  in complex large scale architectures like [5],[6] and simpler PINN architectures like the rest, and that SAFE-NET outperforms or achieves comparable results with these methods on their tasks (Fig 6) using $70-100 \times$ fewer parameters (Table 1) more efficiently (Fig 4).
> >
> > Your suggestion of using non-homogeneous boundary conditions is excellent, and to address it, we have added an experiment on a non-homogeneous BC variant of the heat equation in the revision, where SAFE-NET still outperforms competitors, achieving an L2RE of 2.11e−4. Are there others you'd like us to test as well?
> >
> > Also, we have added an additional ablation study comparing SAFE-NET and PINNs, both with and without the domain knowledge features (DKF). SAFE-NET’s performance varies across PDEs with and without DKF. For example, on the wave task, adding DKF leads to > 47% improvement, while for Burgers for instance, it’s <5%. SAFE-NET’s core mechanism is its adaptive-frequency-and-amplitude framework, and the DKF is a lightweight supplementary component with added benefits when the IC/BC carry more information. We agree that this should be added to the revised text introduction to clarify the role of DKF in SAFE-NET.
> >
> > As there is no separate slot for posting revision PDFs, we must replace the original submission to post a revision at this stage of the review process. We plan to do this in the next 24 hours so you can review the added material. We would be very happy if you let us know of further additions you would like. Thank you again for your constructive comments.
> >
> > References:
> >
> > [8] A. D Jagtap, K. Kawaguchi, and G. Em Karniadakis. Adaptive activation functions accelerate convergence in deep and physics-informed neural networks. Journal of Computational Physics, 2020. 404:109136, 2020
> >
> > [9] Chen, S.; Liu, Z.; Zhang, W.; Yang, J. A Hard-Constraint Wide-Body Physics-Informed Neural Network Model for Solving Multiple Cases in Forward Problems for Partial Differential Equations. Appl. Sci. 2024, 14, 189.
> >
> > [10] F. Jiang, X. Hou, and M. Xia. Densely multiplied physics-informed neural networks. arXiv preprint arXiv:2402.04390, 2024.

---

> > ### Author Response · Authors · 2025-09-01
> > **Hi again! Revision added**
> >
> > Hi again and hope you are having a wonderful day. We have now replaced our original submission with a revised version of our paper. In it, we have made the corrections you suggested such as clarifications regarding the domain knowledge component in the introduction and the idealized example's conditions, fixed typos and formulations as you pointed out, and added new sections in the appendix (ablation study of the role of DKF in Appendix C.1.6 and the heat variant in Appendix C.3 are now in this revision). We would be very happy if you let us know of further additions you would like. Thank you again for your constructive comments and for helping us improve this work.

---

### Author Response · Authors · 2025-08-22
**Thank you all for your constructive comments!**

We thank all the reviewers for their time and feedback! We sincerely appreciate your thorough evaluation of our work. We were glad to see that our attempt at demonstrating SAFE-NET’s efficiency and accuracy through extensive experiments have been positively received, with the reviewers describing the experiments as “very fully described” (Reviewer LL46), citing the extensive ablation studies as one of the strengths (Reviewer Pn2M) “which also give additional insights into problems with PINN-training” (Reviewer LL46), describing our model as “very simple and clean, yet it achieved great numerical performance over many PDEs” (Reviewer 9qG6). We also appreciate their questions, comments, and suggestions. We provide answers to the main points raised by each reviewer below, and outline changes that we plan to address in our revision. Please feel free to follow up with us! We very much welcome any feedback that can further strengthen our paper.

---

### Decision · Action_Editor_KNF1 · 2025-10-13

**Recommendation:** Accept as is

**Audience:**

Yes

**Audience Explanation:**

There is continued interest in PINN training, especially the inclusion of ideas from classical PDE theory (like fourier bases) to improve their training.

**Claims And Evidence:**

Yes

**Claims Explanation:**

The work uses a typical suite of PDE test problems for their  evaluation and compare to a sizeable range of alternative PINN modifications. While I appreciate the thoroughness of reviewer Pn2M in identifying ways in which the manuscript could be strengthened, I believe that the bar of TMLR is met.